# Exploration of diverse solutions for the calibration of imperfect climate models

Saloua Peatier[1], Benjamin M. Sanderson[2], and Laurent Terray[1]

[1]CERFACS/CECI, Toulouse, France
[2]CICERO, Oslo, Norway

**Correspondence:** Saloua Peatier (peatier@cerfacs.fr)

**Abstract.** The calibration of Earth System Model parameters is subject to both data, time and computational constraints. The high dimensionality of this calibration problem, combined with errors arising from model structural assumptions, makes it impossible to find model versions fully consistent with historical observations. Therefore, the potential for multiple plausible configurations presenting different tradeoffs between skills in various variables and spatial regions remains usually untested. In this study, we lay out a formalism for making different assumptions about how ensemble variability in a Perturbed Physics Ensemble relates to model error, proposing an empirical but practical solution for finding diverse near-optimal solutions. We argue that the effective degrees of freedom in model performance response to parameter input (the 'parametric component') is, in fact, relatively small, illustrating why manual calibration is often able to find near-optimal solutions. The results explore the potential for comparably performing parameter configurations making different trade-offs in model errors. These model candidates can inform model development and could potentially lead to significantly different future climate evolution.

## 1 Introduction

General Circulation Models (GCM) and Earth System Models (ESM) are the primary tools for making projections about the future state of the climate system. It is an important goal of climate science to continually improve these models and to better quantify their uncertainties. Constraints on computational resources limit the ability to resolve small-scales mechanisms, and sub-grid parametrizations are used to represent processes such as atmospheric convection or clouds. These parametrizations are based on numerous unconstrained parameters that introduce uncertainty in climate simulations. Therefore, climate models are subject to a challenging calibration (or 'tuning') problem. When used as tools of projection of future climate trajectories, they cannot be calibrated directly on their performance. Instead, assessment of performance and skill arises jointly from confidence in the understood realism of physical parametrizations of relevant climatological processes, along with the fidelity of model representation of historical climate change. Practical approaches to model calibration are subject to both data, time and computational constraints.

For the simplest models (zero or low dimensional representations of the climate system), model simulations are sufficiently cheap with sufficiently few degrees of freedom that Bayesian formalism can be fully applied to estimate model uncertainty

(Ricciuto et al., 2008; Meinshausen et al., 2011; Bodman and Jones, 2016; Nauels et al., 2017; Dorheim et al., 2020). However, more complex models such as GCM present a number of difficulties for objective calibration which have resulted in a status quo in which manual calibration remains the default approach (Mauritsen et al., 2012; Hourdin et al., 2017). Such approaches have not yet been operationally replaced by objective calibration approaches, but leave large intractable uncertainties. In particular, the potential existence of comparably performing alternative configurations with significantly different future climate evolution (Ho et al., 2012; Hourdin et al., 2023). Failing to explore alternative model configurations can result in model ensembles which may not adequately sample projection uncertainty. For example, some of the CMIP6 model projections were 'too hot' when compared with other lines of evidence and using all of these models without statistical adjustment (in a simple 'model democracy' approach) could lead to an overestimate of future temperature change (Hausfather et al., 2022).

Although manual calibration remains by far the most common practice, objective calibration methods have been developed and tested in climate models (Price et al., 2006; Nan et al., 2014; Sellar et al., 2019). Approaches to date with GCMs have mainly relied on Perturbed Parameter Ensembles (PPE) of simulations, allowing an initial stochastic sample of the parametric response of the model. The construction of meta-models is then needed to emulate this parametric response and enhance the number of samples. The meta-models can be quadratic (Neelin et al., 2010), logistic regression (Bellprat et al., 2012), Gaussian process emulators (Salter and Williamson, 2016) or neural networks (Sanderson et al., 2008). Each of these meta-modeling approaches offers different advantages in terms of accuracy, flexibility and speed (Lim and Zhai, 2017), but often require prior assumptions on how smooth the parameter response surface might be, how noisy the samples themselves are. Such approaches allow for the definition of plausible or "not ruled out yet" spaces when using a low dimensional output space (such as global mean quantities, Bellprat et al. (2012); Williamson et al. (2015)), potentially allowing for additional ensemble generations which sample in the "not ruled out yet" space (Williamson et al., 2015). Emulators can be improved in promising sub-regions of the parameter space by running a new PPE in a reduced parameter space to increase the ensemble density (sometimes referred as an "iterative refocussing" approach, Williamson et al. (2017)). However, the choice of which region to initially focus on depends on advice from model developers and is itself subject to error in emulation. Finally, one of the strongest limitation when developing GCM automatic tuning approach is the high computational cost of the PPE, leading to the impossibility of running ensemble large enough for uncertainty quantification. Dunbar et al. (2021) relies on calibrate-emulate-sample methods to generate probability distributions of the parameters at a fraction of the computational cost usually required to obtain them, allowing for climate predictions with quantified parametric uncertainties.

Climate models produce high dimensional output across space, time and variable dimensions. Performance is often addressed by integrated output spanning these dimensions (Gleckler et al., 2008; Sanderson et al., 2017) and so calibration techniques must be able to represent spatial performance in order to be useful to development. In a low dimensional space defined by global mean quantities, it is possible to find one model version which is consistent with observations (Williamson et al., 2015), but this is not true when considering the high dimensionality of climate model outputs. When considering an assessment of model error integrated over a large number of grid points and variables, structural trade-offs may arise between model outputs

which cannot be simultaneously optimized by adjusting model parameters. For example, McNeall et al. (2016) found that land-surface parameters which were optimal for the simulation of the representation of the Amazon forest fraction were not optimal for other regions. In another case, structural errors in an atmospheric model were found to increase significantly with the addition of variables to a spatial metric (Sanderson et al., 2008). As such, the potential structural error component is implicitly related to the dimensionality of the space in which the cost function is constructed. For example, Howland et al. (2022) demonstrated that the use of seasonally averaged climate statistics, rather than annually averaged, could narrow the uncertainty of climate model predictions.

In order to reduce the complexity of the emulation problem, and to preserve the covariance structure of the model output, it is common to reduce the dimensionality of the output through Principal Component Analysis (PCA, e.g. Higdon et al. (2008); Wilkinson (2010); Sexton et al. (2012)). Notably, for some spatial applications, this dimensional reduction may be insufficient to resolve certain important climatological features such as extreme precipitation frequency (Jewson, 2020). This PCA representation, however, has some apparent drawbacks for optimization. An orthogonal space constructed from the dominant modes of variability in a perturbed parameter ensemble may not be able to describe some components of the spatial pattern of model error (O'Lenic and Livezey, 1988). Salter et al. (2019) proposed an approach to global optimization of a model with spatially complex output by a rotation of principal components such that model errors were describable in reduced dimensionality basis set by including some aspects of higher order modes in the rotated, truncated basis set in order to better describe the error patterns of ensemble members. The method, however, makes some significant assumptions about the ability of a statistical model to predict the parametric response of high order modes and does not allow for an exploration of structural trade-offs between different variables, such as those found by McNeall et al. (2016).

In this study, we argue that considering a sub-set of plausible candidate calibrations sampling the diversity of model error spatial patterns can help better understand the model biases. Such approach could also help to better understand model uncertainty in climate projections, as previous studies highlighted the possibility for several calibrations of a single climate model to present very different future climates (Hourdin et al. (2023), Peatier et al. (2022)). In this sense, we are not searching for an optimal parameter configuration, rather for model configurations which perform comparably to a reference model version. We lay out an alternative formalism which makes different assumptions about how ensemble variability in a PPE relates to structural error and how it can thus inform model development. This formalism allows the empirical decomposition of the model error into one component depending on the parameter values, and a component arising from structural inaccuracies. The approach is used as a practical solution for finding diverse near-optimal solutions exploring key model error trade-offs. We start by illustrating the method using a simplified uni-variate case focusing on surface temperature errors (Section 3), before applying it to a more generalised multi-variate tuning case using 5 climatic fields (Section 4). Finally, we discuss and summarize the main results (Section 5).

## 2 Methods

### 2.1 Model and Perturbed Parameter Ensemble (PPE)

The model used in this study is ARPEGE-Climat, the atmospheric component of the CNRM-CM6 climate model, referred to as $f$, the climate model. The reference configuration of this model will be referred to as CNRM-CM6-1 and has been tuned by the model developers for the CMIP6 exercise (Roehrig et al., 2020). A PPE of this model f is created, containing 102 amip simulations Eyring et al. (2016) differing by their parameter values, representing the period 1979-1981 (3 years), with pre-specified Sea Surface Temperatures (Peatier et al., 2022). Thirty model parameters (see Appendix A1) are perturbed with a Latin Hypercube Sampling (LHS) strategy, producing a variety of simulated climate states in the experiment : $F = (f(\theta_1), ..., f(\theta_n))$ based on a space-filling maximin design $\theta = (\theta_1, ..., \theta_n)$ (Peatier et al., 2022), with $n = 102$ and $\theta_i$ a vector of 30 parameter values. For the present study, we consider the annual means averaged over the whole 1979-1981 period. We write the model output $f(\theta_i)$ as a vector of length $l$, such that $F$ has dimension $l \times n$, where $n$ is the number of ensemble members ($n = 102$) and $l$ the number of grid points $l = 32768$.

### 2.2 EOF analysis

In order to build emulators of the simulated spatial climatology, general practice is to reduce the dimensionality of the emulated response, and a common strategy is an EOF (Empirical Orthogonal Function) analysis (Higdon et al., 2008; Wilkinson, 2010; Sexton et al., 2012; Salter et al., 2019), which produces $n$ eigenvectors that can be used as basis vectors. Given $n << l$, the reconstruction of $F$ is exact and reduces the complexity of the emulator required.

Variability in $F$ is explained in descending order of eigenvectors, such that a truncation to the first $q$ modes yields a basis $\Gamma_q = (\gamma_1, ..., \gamma_q)$ which produces an approximate reconstruction of the initial data, thus further reducing the scale of the emulation problem. Truncation length is often chosen such that a given fraction of ensemble variance (often 90-95%) is preserved (Higdon et al., 2008; Chang et al., 2014), but some authors have argued that higher order modes may need to be included to allow resolution of optimal configurations (Salter et al., 2019). We discuss choices of $q$ in the first application (Section 3).

The EOF basis $\Gamma_q$ is based on the centered ensemble $(F - \mu)$, with $\mu$ the ensemble mean. As a result, each anomaly $(f(\theta_i) - \mu)$ is associated with a coefficient $c(\theta_i)$ (or Principal Component, PC) such as:

$$c(\theta_i) = (\Gamma_q^T \Gamma_q)^{-1} \Gamma_q^T (f(\theta_i) - \mu) \tag{1}$$

Given an orthogonal basis, the full spatial field of length $l$ can be approximately reconstructed as a function of the $q$ coefficients:

$$f(\theta_i) - \mu = \Gamma_q c(\theta_i) + r_f, \tag{2}$$

with $r_f$ a residual that depends on the choice of $q$. Considering a variable $j$ (for example, the air surface temperature, as in the first application - section 3.1), such that $z_j$ is the observed field for the variable and $f_j(\theta_i)$ is the model simulated field for that variable, for a given parameter input $\theta_i$. As for $F$, we can subtract the ensemble mean $\mu$ from the observation and project the anomaly of the observation $(z_j - \mu)$ (which is also the error of the ensemble mean $\mu$) onto the basis $\Gamma_q$ using Eq. 1 :

$$z_i - \mu = \Gamma_q c_z + r_z \tag{3}$$

where $r_z$ is a residual representing the part of the observation $z_j$ that can not be projected on the basis $\Gamma_q$. This residual $r_z$ will, as $r_f$, depend on the choice of $q$ but will never (even when $q = n$) equals 0, as the basis $\Gamma_q$ explains the maximum amount of variability in $F$ but does not guarantee to fully represent the spatial pattern of the observation $z_j$ (Salter et al., 2019).

## 2.3 Model error decomposition

The model error pattern of a given parameter sample, $E_j(\theta_i) = z_j - f_j(\theta_i)$, can be expressed in the basis $\Gamma_q$ and becomes the sum of a term that depends on the vector of parameter $\theta_i$ (here called parametric component), and a term unsolvable in the basis $\Gamma_q$ (here called non-parametric component) :

$$E_j(\theta_i) = \underbrace{\Gamma_q[c_z - c(\theta)]}_{Parametric} + \underbrace{r_z - r_f}_{Non-parametric} \tag{4}$$

We could consider a skill score defined by the Mean Square Error (MSE) of the spatial error pattern $E_j(\theta_i)$:

$$e_j(\theta_i) = \frac{1}{l}\Sigma((E_j(\theta_i))^2), \tag{5}$$

$$= \frac{1}{l}\left(\Sigma(\Gamma_q[c_r - c(\theta_i)] + r_z - r_f)^2\right) \tag{6}$$

Furthermore, because $(r_z - r_f)$ is orthogonal by construction to the basis $\Gamma_q$, the interaction terms in Eq.5 are zero. As a result and using Eq. 4, the integrated model error $e_j(\theta_i)$ becomes a linear sum of a parametric component $p_j(\theta_i)$ and a non-parametric component $u_j$:

$$e_j(\theta_i) = \frac{1}{l}\Sigma\left(\Gamma_q[c_z - c(\theta_i)]^2\right) + \frac{1}{l}\Sigma(r_z - r_f)^2, \tag{7}$$

$$= p_j(\theta_i) + u_j \tag{8}$$

## 2.4 The discrepancy term

We consider, following Rougier (2007)and Salter et al. (2019), that an observation $z$ can be represented as a sum of a simulation using the 'best' set of parameter $\theta^*$ of the climate model $f$ and a term (initially unknown) representing discrepancy $\eta$.

$$z = f(\theta^*) + \eta \tag{9}$$

The discrepancy effectively represents the difference between the climate model and the measured climate that cannot be resolved by varying the model parameters (Sexton et al., 2012). Such differences could arise from processes which are entirely

missing from the climate model, or from fundamental deficiencies in the representation of processes which are included : through limited resolution, the adoption of an erroneous assumption in the parameterization scheme or parameters not included in the tuning process. The discrepancy $\eta$ can be defined as the integrated error associated with the optimal calibration $\theta*$. Considering a variable $j$, the discrepancy term $\eta_j$ is defined as :

$$\eta_j = \frac{1}{l}\Sigma((z_j - f_j(\theta^*))^2), \tag{10}$$

$$= e_j(\theta^*) \tag{11}$$

In this case and following Eq. 4, $\eta_j$ can also be expressed as a linear sum of a parametric component $p_j(\theta^*)$ and a non-parametric component $u_j$ :

$$\eta_j = \frac{1}{l}\Sigma\left(\Gamma_q[c_z - c(\theta^*)]^2\right) + \frac{1}{l}\Sigma(r_z - r_f)^2, \tag{12}$$

$$= p_j(\theta^*) + u_j \tag{13}$$

The irreducible error component of the climate model is represented by the $\eta$ term, known as the discrepancy. To make this statement, Sexton et al. (2012) have to assert that the climate model is informative about the real system and the discrepancy term can be seen as a measure of how informative our climate model is about the real world. Sexton et al. (2012) think of the discrepancy by imagining trying to predict what the model output would be if all the inadequacies in the climate model were removed. The result would be uncertain and so discrepancy is often seen as a distribution, assumed Gaussian, and described by a mean and variance (Rougier, 2007; Sexton et al., 2012).

The calibration $\theta^*$ is usually defined as the 'best' input setting, but it is hard to give an operational definition for an imperfect climate model (Rougier, 2007; Salter et al., 2019). In practice, we can only propose an approximated $\theta*$ and multiple 'best analogues' to this approximation exist (Sexton et al., 2012). In this work, we intend to select $m$ near-optimal model candidates $(\hat{\theta}_1, ..., \hat{\theta}_m)$ approximating $\theta^*$ and sampling the discrepancy term distribution $\eta$. We discuss optimization using a simple emulator design in Section 2.5 and candidates selection in Section 2.6.

## 2.5 Statistical model and optimization

Optimization requires the derivation of a relationship between the model input parameters $\theta$ and the PC coefficients $c(\theta)$. In the following illustration and as in Peatier et al. (2022), we consider a multi-linear regression:

$$c_{em}(\theta_i) \approx \beta\theta_i + c_0, \tag{14}$$

where $\beta$ is the least-square regression solution derived from $F$, and $c_0$ is the ensemble mean coefficients. The regression predictions are used in Eq. 12 to predict the model MSE as a function of input parameters $\theta_i$. More details on the choice and performance of the statistical model can be find in the Appendix C.

In this study, the objective of optimization is to look for non-unique solutions $(\hat{\theta}_1, ..., \hat{\theta}_m)$ whose performances are lower or comparable to that of a reference model, yet sampling possible trade-offs in the objective function. The reference model will be the model CNRM-CM6-1, tuned by the model developers for the CMIP6 exercise (Roehrig et al., 2020). This reference model has been validated by the experts and can serve as a threshold to define whether a model calibration is near-optimal.

We can then consider a $10^5$ member Latin Hypercube sample of the model parameter space and produce a distribution of predicted $p_{em}(\theta_i)$ values. The parametric error associated with the reference model, hereafter named $p(\theta_0)$, is considered as a threshold to define the near-optimal candidates. For a given climatic field $j$, we consider the subset of $m$ emulated cases $\hat{\theta} = (\hat{\theta}_1, ..., \hat{\theta}_m)$, where model error is predicted to be lower than the reference model error :

$$p_{em,j}(\hat{\theta}_i) < p_j(\theta_0) \tag{15}$$

For operational use, ESM developers generally attempt to minimize a multi-variate metric (Schmidt et al., 2017; Hourdin et al., 2017), considering $n_j$ different climatic fields. In this case, all the individual errors $e_j(\theta_i)$ and $p_j(\theta_i)$ need to be aggregated in a single score. The simplest way to obtain such multi-variate skill score is to normalize each uni-variate parametric errors $p_j(\theta_i)$ relatively to the reference model error such as :

$$p_{em,tot}(\hat{\theta}_i) = \frac{1}{n_j} \Sigma_{j=1}^{n_j} \frac{p_{em,j}(\hat{\theta}_i)}{p_j(\theta_0)} \tag{16}$$

In this case, the condition for the near-optimal sub-set is :

$$p_{em,tot}(\hat{\theta}_i) < 1 \tag{17}$$

For an application to surface temperature, we can consider, within the optimized subset of emulated cases $\hat{\theta}_{tas} = (\hat{\theta}_1, ..., \hat{\theta}_m)$, a selection of candidate calibrations producing pattern error as diverse as possible while minimizing the aggregated metric $e_{tas}(\hat{\theta}_i)$. The selection of candidate calibrations is detailed in Section 2.6, the results are shown in Section 3. For a multi-variate application, we consider the near-optimal sub-set $\hat{\theta}_{tot} = (\hat{\theta}_1, ..., \hat{\theta}_m)$, which minimize the metric $e_{tot}(\hat{\theta}_i)$. The selection of candidate calibrations is detailed in Section 2.6, the results are shown in Section 4.

## 2.6 Selection of diverse candidate calibrations

Given the subsets of plausible model configuration $\hat{\theta}_{tas}$ or $\hat{\theta}_{tot}$, we aim to identify $k$ solutions which explore different trade-offs. This is obtained through a k-medians clustering analysis, which separate samples in $k$ groups of equal variance, minimizing a criterion known as the inertia, computed as the sum of the minimal distances within the clusters (Hastie et al., 2009; Pedregosa et al., 2011).

As a first step, we apply the k-medians clustering to the surface temperature Principal Components of the plausible model configuration sub-set, the coefficients $c_{tas}(\hat{\theta}_{tas})$. The medians of the samples in each clusters are called the centroids. The centroids are points from the original dataset, therefore we know their associated vector of parameters $\theta$ and can use them to

sample the sub-set of diverse and plausible configurations. These calibration candidates are tested in the climate model and results are presented in the Section 3.

In a multi-variate context, the candidates should reflect the model error diversity among both the different climatic fields $j$ and the different EOF modes of each field. Considering the subset of plausible configurations $\hat{\theta}_{tot}$ , we apply the k-medians clustering analysis to all the data coefficients $c_j(\hat{\theta}_{tot})$, normalized by the reference model coefficients $c_j(\theta_0)$, for $j$ climatic fields. In this case, applying the clustering analysis within the EOF space reduce the computational cost and allows to look for candidates representing the diversity of error patterns in a multi-variate context. The clustering will minimize the distances

between the coefficients of a same cluster, while maximizing the inter-cluster distances. As for the uni-variate application, the $k$ centroids will be retained as candidates to sample the represent the diversity of error patterns in the plausible subset of configurations. In the Section 4, we propose an application considering 5 climatic variables ($n_j = 5$, Table 1).

    For both cases, we tested the sensitivity of the analysis to the number of clusters. Following the Elbow method applied the

the inertia and the maximization of Dunn's index, we have decided to keep $k = 12$ clusters for both applications. More detailed about the sensitivity of the analysis to the cluster number and the choice of $k$ are given in the Appendix B.

## 3   First application : surface temperature error

We consider an example problem where the objective is to propose diverse candidates minimizing the Mean Squared Error of a single climatic field, the surface air temperature, when compared with observational estimates. Here we use the BEST dataset

(Rohde and Hausfather, 2020), over the simulated period (1979-1981). Observations have been interpolated onto the model grid for a better comparison.

    In this example, the first key question will be to select the truncation length of the basis $\Gamma_q$. (Salter et al., 2019) define two main requirements for an optimal basis selection : being able to represent the observations $y_j$ within the chosen basis (a

feature not guaranteed by the EOF analysis of the PPE), and retaining enough signal in the chosen subspace to enable accurate emulators to be built for the basis coefficients. Our objectives here are a bit different, as we want to conserve our ability to identify the trade-offs made by candidates calibrations in the non-parametric components of the model performance. We argue that the original basis $\Gamma_q$ is representative of the spatial degrees of freedom achievable through perturbations of the chosen parameters. As such, the degree to which the observational bias projects onto it is itself meaningful and can be used as a tool to

identify components of model error which are orthogonal to parameter response patterns (and therefore not reducible through parameter tuning).

Furthermore, we want, as (Salter et al., 2019), to be able to build accurate emulators for the basis coefficients. In this sense, the basis should not include variability modes poorly represented by the emulator. Sections 3.2 and 3.3 discuss the choice of $q$, the truncation length.

## 3.1 Assessing meaningful number of degrees of freedom

We first consider how modes of intra-ensemble variability relate to the representation of model integrated Mean Square Error of surface temperature $e_{tas}(\theta_i)$. Following Section 2.2, by projecting the spatial anomalies of models and observations onto the basis defined by the truncated EOF set, the mean-squared error can be partitioned into a parametric component (the projection $p_{tas}(\theta_i)$) and non-parametric component (the residual $u_{tas}$). Figure 1 considers examples of the full model errors associated with the PPE simulations and its decomposition for different numbers of EOF modes retained, $q = 102$ being the perfect reconstruction of the full error $e_{tas}(\theta_i)$.

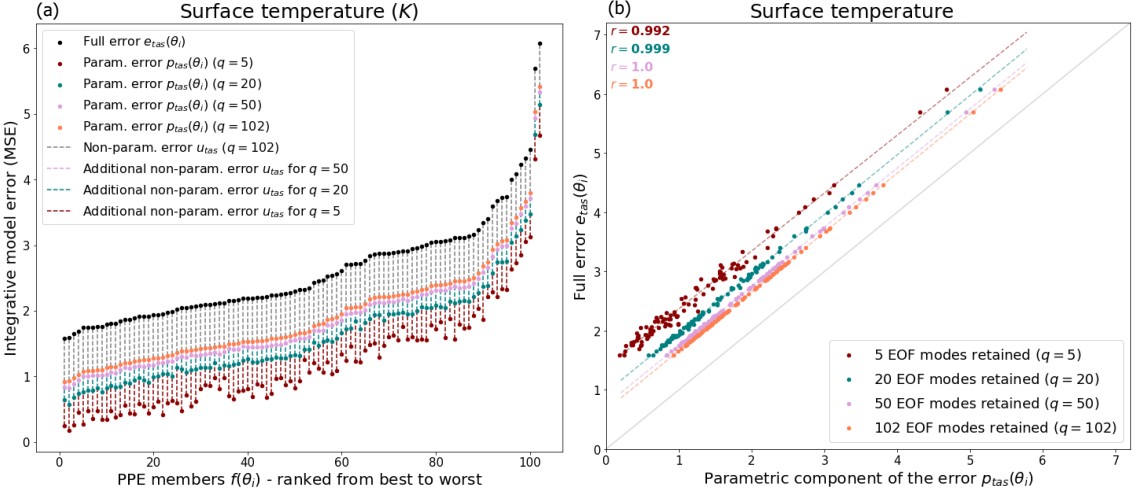

**Figure 1.** Full model error $e_{tas}$ and its parametric component $p_{tas}(\theta_i)$ for different truncation length : $q = 5$ (red dots), $q = 20$ (blue dots), $q = 50$ (pink dots), $q = 102$ (orange dots). (a) Full error partitioning in parametric and non-parametric components in the PPE members $f(\theta_i)$ ranked from lowest to highest error. (b) Correlation between the full error $e_{tas}$ and its parametric component $p_{tas}(\theta_i)$ within the PPE.

While retaining a relatively small number of modes ($q = 5$), the correlation between the full model error and its parametric component is already really strong among the PPE members, with a Pearson correlation coefficient of $r = 0.982$ (Figure 1b). This correlation does not improve a lot when considering higher modes : $r = 0.998$ for $q = 20$ and $r = 0.999$ for $q = 50$. This implies that only a relatively small number of modes is required to reproduce the ensemble variance in $e_{tas}(\theta_i)$. The variance in ensemble spatial error pattern could be described by a small number of degrees of freedom.

However, even for the perfect reconstruction of the model error (when $q = 102$), a non-null non-parametric component exists, and its ratio corresponds to $26\%$ of the full model errors averaged over the PPE members. This ratio increases when retaining less EOF modes, and a large fraction of the model error pattern is not represented within the parametric component. For a truncation of $q = 5$, the non-parametric component of the error $u_{tas}$ is $53\%$ of the total $e_{tas}(\theta_i)$, in average over the PPE. Together, this implies that the variance in model error seen in the PPE can be explained by a small number of modes, but a significant fraction of this error is not represented within the parametric component of the error decomposition.

## 3.2 Truncation and parametric emulation

In section 3.1, we demonstrated that the majority of variance in model MSE can be described as a function of a small number of spatial modes. In an operational model tuning exercise, we want to make sure that we explain most of the ensemble variance within the truncated EOF basis, so we decided to use a subjective minimum of $85\%$ of explained variance when deciding on the truncation length. Now, how does this relate to parametric dependency? We follow Section 14 to build a linear emulator relating the model parameters $\theta$ to the PC coefficients $c(\theta)$. Out of a total of 102 simulations, 92 are randomly selected to form the training set. This training set is used to compute the EOF analysis and to derive least-square regression coefficients of the emulator. The out-of-sample emulator performance is then assessed on the remaining 10 simulations, after projection onto the EOF basis. This process is repeated 10 times with random samples of $F$ used as training sets, to assess the predictive performance of the regression model (i.e. the correlation between out-of-sample predicted $c(\theta)$ and true $c(\theta)$).

Figure 2a shows both in-sample and out-of-sample skill scores, in terms of mean and standard deviation across the 10 repetitions. The average of out-of-sample performance cumulative on modes is also represented by the red curve (ex : when $q = 5$, the red curve is the average of the orange curve over the modes 1 to 5 included). We find that out-of-sample emulation skill declines rapidly when the number of modes increases. This result challenges the utility of including high-order modes in the high order modes in spatial emulator of parametric response (as in Salter et al. (2019)), indicating that high order spatial modes may be too noisy to represent any parametric signal in the ensemble and emulator design considered here. Here we consider an example of truncation at $q = 18$ that will be used in the rest of the study. It corresponds to the point when the average of out-of-sample performance cumulative on modes reach the arbitrary threshold of 0.5 and explains $94\%$ of the ensemble variance (respective our condition of at least $85\%$ of explained variance).

Figure 2b shows the ratios between the PPE parametric (dark blue), non-parametric errors (light blue) and the total error (green), as a function of the number of EOF modes retained. We see that for a EOF basis retaining 1 to 5 modes, each component represents around $50\%$ of the total error on average. For the truncation of $q = 18$, the parametric error represents $63\%$ of the full error on average, and the non-parametric error $37\%$. This ratio evolves slowly when adding higher modes, and for a perfect reconstruction ($q = 102$), $\frac{p(\theta_i)}{e(\theta_i)} = 74\%$ and $\frac{u}{e(\theta_i)} = 26\%$. But we also note that the large variability of $p_{tas}(\theta_i)$ across the PPE (represented by the standard deviation) is constant irrespective of the number of EOF modes retained, highlighting the strong dependency of this error component on the parameter values. On the other hand, the variability of the residual error $u$

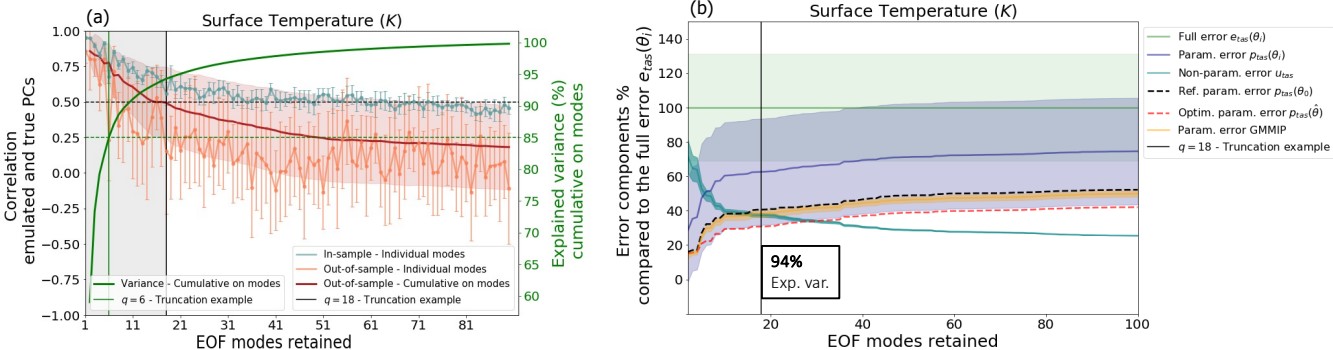

**Figure 2.** Truncation choice based on parametric emulation and error decomposition. (a) Shows the correlation between the emulated and true PCs of the surface temperature EOF, for the different modes of variability. The correlation is showed within the training set (blue curve) and the test set (orange curve). The red curve and shading is the mean correlation averaged over the modes cumulatively. The solid green curve represents the percentage of variance explained when retaining up to $q$ modes of the EOF. The dashed green horizontal line shows a threshold of $85\%$ of explained variance and the solid green vertical line is the truncation length needed to satisfy this threshold. (b) Shows the ratio of the error components compared to the full error $e_{tas}(\theta_i)$ (in green) as function of the number of modes of variability retained. The lines are the ensemble means and the shadings represent the standard deviations. The plot shows the ratios of the PPE parametric error (dark blue), the PPE non-parametric error (light blue), the reference calibration parametric error $p_{tas}(\theta_0)$ (red dotted curve) and the GMMIP parametric error (orange). An example of truncation at $q = 18$ is represented on both plots by the black vertical line.

within the PPE decreases when retaining more EOF modes, and is already very small for our truncation example of $q = 18$.

In the context of the Global Monsoons Model Inter-comparison Project (GMMIP, Zhou et al. (2016)), an ensemble of 10 atmospheric-only simulations of the CNRM-CM6-1 was run. In this ensemble, the reference model calibration was used, the SST was forced with the same observations as the PPE and the members differ by their initial conditions only. This dataset can be used to consider the effect of internal variability on the error decomposition, and will be referred to as the GMMIP dataset.

The GMMIP dataset includes 10 atmospheric simulations of CNRM-CM6-1 with the reference calibration but different initial conditions, that can be projected into the PPE-derived EOF basis to compute their associated parametric errors. The variability of the parametric component of the error across the GMMIP dataset is very small and does not depend on the truncation length. The fact that, for $q = 18$ or higher, the variability of $u$ is even smaller than the internal variability of the parametric component confirms that this part of the error is not dependent on the parametric values anymore.

Another point to note from Figure 2b is that the reference calibration of the model performs well and shows a near-minimal value of parametric error in the ensemble. Following Eq. 14, we use a multi-linear regression that emulates the parametric component of the model error from the calibration values. This emulator is then optimized to find an example of near-optimal calibration $\hat{\theta}$ that minimizes the parametric component of the error. The optimization is done for for all the different truncation

lengths. As shown on Figure 2a, the parametric component of the near-optimal calibration $p(\hat{\theta})$ is a bit lower than the para-
metric error of the reference calibration when retaining 5 or more modes and starts evolving in parallel of the PPE mean when
retaining 7 or more modes. The difference between the PPE mean and this example of optimal calibration becomes constant
when $q = 7$ or more, suggesting that there is no improvements of the optimization when adding modes higher than 7.

These results suggest that the EOF basis $\Gamma_q$ truncated at a relatively small number $q = 18$ is a good representation of the
parametric component of the model error pattern. Therefore, the truncation can be used to identify the residual $u$, that does
not depend on the perturbed parametric values. Adding further modes has limited impact on the representation of ensemble
variation in integrated error and does not improve the ability to find near-optimal candidates because of the poor skill of the
higher modes regression prediction. In the following, we will only use a truncation at rank $q = 18$.

### 3.3   Trade-offs in model candidates

Following the methodology discussed in the Section 2.6, all emulated members with a parametric error lower than the reference
are selected from a 100,000 LHS set of emulations and considered as a sub-set of near-optimal calibrations. From this sub-set,
12 candidates have been identified in order to maximize the diversity of model errors. The 12 calibrated set of parameters were
then used in the ARPEGE-Climat model to produce actual atmospheric simulations. One of the calibrations leads to a crash
in the model and 11 others produced the complete atmospheric simulations. The annual mean surface temperature of these 11
candidates were projected onto the EOF basis computed from the 102 members of the PPE, to obtain the principal components.
Figure 3 presents the representation of the first five EOF modes by the principal components of the projected model candidates
: the closer the candidates are to the observation in the different modes, the lower their parametric error.

    Figure 3 provides some confidence in both the emulation skill and the method used for the selection of near-optimal and
diverse candidates. Although some differences exist between the emulations of the candidates and their actual atmospheric
simulations, all of them show principal components within the near-optimal sub-set of calibrations for the 5 first EOF modes,
thus respecting the condition for near-optimal calibration. Moreover, the candidates seem to explore a range of principal com-
ponent values as wide as the near-optimal sub-set of calibrations, meaning that we achieve the diversity expected in terms of
model errors. In the fifth mode, the projected observations are outside of the emulated ensemble, illustrating that all ensemble
members have a non-zero error for this mode, highlighting the existence of a structural bias preventing us from tuning the
model to match observations on this axis.

Figure 3 also illustrates the constraints due to optimisation on the principal components of the near-optimal sub-set of cal-
ibration. Indeed, the principal components associated with the first EOF mode of the near-optimal sub-set of calibrations (in
dark gray on Figure 3) span a very reduced range of values compared to the full emulated ensemble. This result highlights a
strong constraint on the first mode of the EOF, stronger than on the other modes. In other words, the candidates have to have

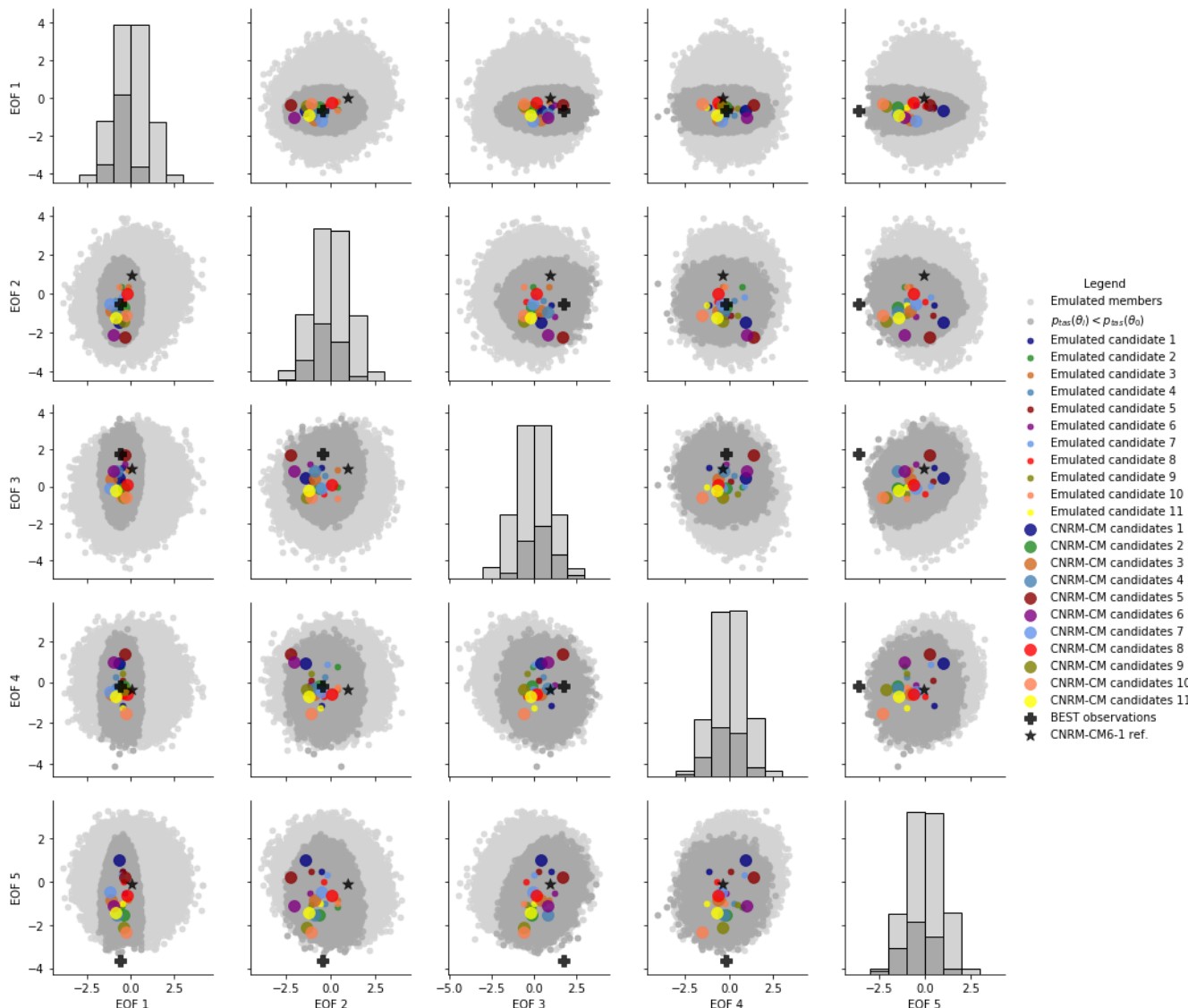

**Figure 3.** Correlation between the different standardised PC (obtained from the 102 memberPPE EOF) for the 100.000 emulated simulations (light gray), the near-optimal emulated members (dark grey, parametric error lower than the reference CNRM-CM6-1, in dark grey), the 11 emulated candidates (colored dots) and the 11 actual CNRM-CM simulations (colored disks), the reference simulation (star) and the observation (cross).

a representation of the first EOF mode close to the projected observations in order to achieve a parametric error below the
reference. This is an expected result, knowing that the first mode explains most of the PPE variance and that the amount of variance explained by each mode individually decreases in higher modes.

Finally, Figure 3 illustrates that it is impossible for the model candidates to perform equally well on all modes and fit observations perfectly. Tradeoffs exist even in this space where the variability is driven by the calibration.

Candidate 5, for example, represents very well the modes 1 and 3, with values of principal components almost equal to those obtained by projecting the observation on the EOF basis, but is further from the observations in the modes 2,4 and 5. In the same way, candidate 10 performs well for the modes 1, 2 and 5 (being the candidate the closest to observation in mode 5, with the observation outside of the emulated ensemble), but not for modes 3 and 4. Candidate 3 is the best candidate, close to the observations on all modes 1 to 4.

All of the 11 candidates have comparable values of their integrated temperature errors (and all lower than the reference values $p(\theta_0)$ and $e(\theta_0)$) and Figure 3 is a good representation of the trade-offs they have to make in order to minimize this metric. This is a good illustration of the main issue of model tuning : the existence of structural error, which is illustrated here by mode 5, makes impossible the perfect fitting to the observation and candidates are making trade-offs to achieve the metric minimization. This is well known when considering a classic model tuning approach, where multiple climatic variables are considered and the near-optimal calibrations are better representing certain fields at the expense of others in order to minimize a multi-variate metric. Figure 3 is illustrating the problem at the scale of a single field (surface temperature, in this case), highlighting the existence of trade-offs within the near-optimal representation of this field : the temperature will be equally well represented in all the candidates when considering an integrated score (like an MSE), but their spatial error patterns will differ.

## 3.4 Examples of temperature discrepancy term decomposition

Considering, as described in the Section 2.4, that the error associated with a near-optimal model is an approximation of the discrepancy term magnitude, the candidates selected here illustrate that near-optimal solutions can be obtained with a diversity of spatial trade-offs that can be made for a minimization problem, even for a single variable output. Moreover, the discrepancy terms can be decomposed in parametric and non-parametric components as seen in Section 2.4. Given the results of Section 3.3, there is a good practical case for choosing a low dimensional basis for calibration - with evidence that it is sufficient to describe the majority of ensemble error variability and that higher modes are not predictable from parameters. The truncation chosen here is $q = 18$ and Figure 4 presents the decomposition of the near-optimal candidates errors, based on this EOF basis $\Gamma_{q=18}$.

For practical reason, only 4 of the 11 candidates are presented in Figure 4, the rest of the candidates can be seen in Appendix (Figure D1). All of the candidates show full temperature MSEs $e(\hat{\theta}_i)$ between $1.62K$ and $1.99K$, so below the MSE of the reference of $e(\theta_0) = 2.01K$. The candidate 7 is the least good, with $e(\hat{\theta}_7) = 1.99K$ and $p(\hat{\theta}_7) = 0.98K$ and the candidate 3 is the best performing model, with $e(\hat{\theta}_3) = 1.62K$ and $p(\hat{\theta}_3) = 0.62K$. The quality of the statistical emulations of the parametric component varies depending on the candidate and the biases over Antarctica are poorly captured by the emulations. We note that the emulation of candidate 3 shows a rather different parametric error than the actual pattern, with an opposite sign of

the biases over Antarctica, Australia, India and Argentina, as well as a strong under-estimate of the positive bias over central Africa. For candidate 10, the statistical emulation of the parametric error's spatial pattern is really closed to the truth. We discuss the uneven performance of the statistical predictions in Appendix C and argue that the emulator skill is mostly limited by size of the training set.

As stated before, near-optimal candidate errors are our best estimate of the discrepancy term diversity. The full errors shown in Figure 4 display features common to 4 candidates and the reference : negative biases over the mountain regions (Himalaya, Andes and North American mountains) and a positive bias over central Africa. However, the magnitude and position of these biases vary from a model to another, with a particularly strong negative bias over north America in candidate 1 and a strong positive bias over central Africa in candidate 3, for example. This diversity is highlighted when looking at the parametric components of the candidate errors, showing a diversity of error signs and patterns over the poles (especially Antarctica), the South of Europe, India, North Africa and Canada.

The non-parametric components of the errors are smaller and qualitatively similar among the candidates, confirming that they are not strongly controlled by the parameter values. In other words (as expected by the method), the first few modes of the EOF analysis are enough to represent the diversity of model error spatial trade-offs among a sub-set of near-optimal candidates. Moreover, the method allows to visualize and compare these trade-offs through the spatial representation of the parametric component (Figure 4).

## 4 Second application : multi-variate error

### 4.1 Variables, EOF analysis and truncations

The uni-variate analysis conducted in Section 3 illustrates qualitatively the potential for trade-offs and multiple near-optimal solutions of the climate model optimization problem. In this Section, we considered a single uni-variate metric, allowing to select 12 near-optimal candidates maximizing the diversity of spatial error patterns and trade-offs among the different EOF modes.

| Observable variables | Symbol | Units | Data product reference | Years |
|---|---|---|---|---|
| Surface Temperature | tas | $K$ | (Rohde and Hausfather, 2020) | 1979-1981 |
| Precipitation | pr | $mm/day$ | (Huffman et al., 2009) | 1979-1981 |
| Sea Level Pressure | psl | $Pa$ | (Saha et al., 2010) | 1979-1981 |
| SW flux, TOA | SW | $W.m^{-2}$ | (Loeb et al., 2018) | 2000-2002 |
| LW flux, TOA | LW | $W.m^{-2}$ | (Loeb et al., 2018) | 2000-2002 |

**Table 1.** Table of observable variables used in this study, plus citations for the data-products used.

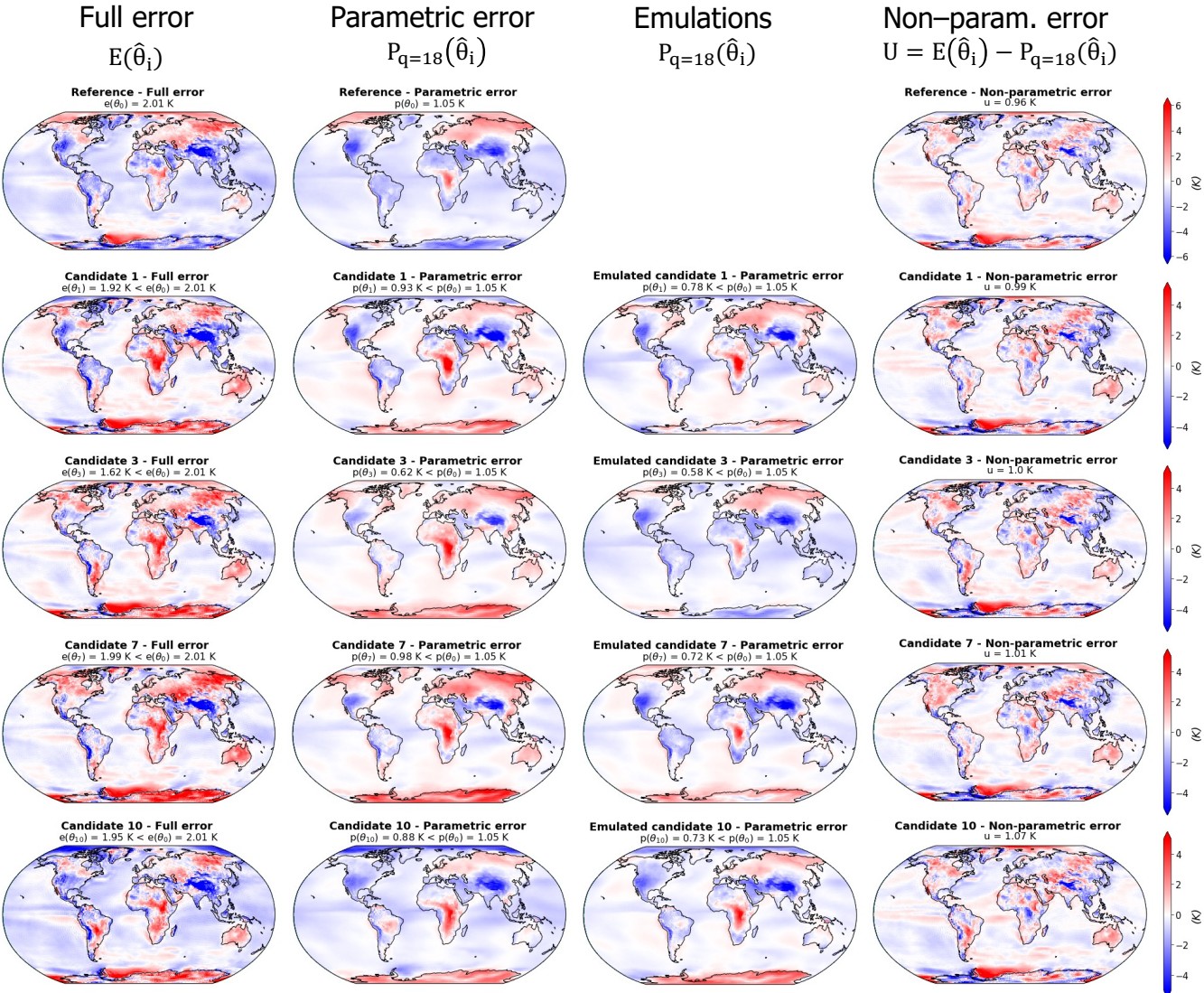

**Figure 4.** Differences between the simulations of temperature and the observations BEST (Rohde and Hausfather, 2020), for the 4 model candidates and the reference. Decomposition of model errors in parametric and non-parametric components using the methodology described in the Section 2, with an EOF basis truncated after the mode 18. The left column shows the full differences between simulations and observations, the second column shows the parametric component of this difference, the third column presents the emulation by the linear regression of this parametric component and the last column is the non-parametric component estimated as the difference between the full error and its parametric component.

In an operational GCM tuning application, the metric considered must encompass multiple climate fields and the optimization results in trade-offs between different uni-variate metrics, with near-optimal models better representing some fields at the

expense of others. The general solution to model calibration for operational use requires consideration of a wide range of climatological fields spanning model components, including both mean state climatologies, assessment of climate variability and

410 historical climate change. This is inherently more qualitative - requiring subjective decisions on variable choices and weighting, which are beyond the scope of this study. However, we can consider an illustration of a multi-variate application, based on 5 climatic fields : the surface temperature (*tas*), the precipitation (*pr*), the short wave (*SW*) and long wave fluxes (*LW*) at the top-of-atmosphere and the surface pressure (*psl*). The model errors will be defined as the MSE between model simulations and the observational datasets lists in Table 1. As for the uni-variate application, EOF analysis of the PPE variance is computed for

the annual means of the different climatic fields and the EOF truncation choices depends on the parametric emulation skill and the error decomposition.

Figure 5 presents the performances of multi-linear regressions in the prediction of the principal components for the 5 fields and we note a strong decrease in out-of-sample prediction skills as we move toward higher EOF modes for all climatic fields.

Based on this result, it is clear that, as for the uni-variate application, the optimization should only retain the first few modes. The truncation lengths should be different from a climatic field to another as the linear regressions perform the best for the SW fluxes but have rather poor out-of-sample skill in terms of sea level pressure, for example. Examples of EOF truncations are given on Figure 5, based on an arbitrary threshold of 0.5 for the averaged correlation coefficient of predicted and true out-of-sample PCs. We also ensured that the truncated basis explained at least $85\%$ of the ensemble variance. These examples suggest

that it is possible to retain up to 28 EOF modes for the TOA SW fluxes uni-variate metric, whereas no mode higher than 8 should be considered for the sea level pressure in order to keep satisfying statistical predictions. Moreover, some variables require more EOF modes than other in order to explain most of the ensemble variance. For precipitation, we need to keep 18 EOF modes in order to explain $85\%$ of variance, whereas for sea level pressure, the first 8 EOF modes explain $92\%$ of the variance. However, for every climatic field considered, the variance of model errors within the PPE is already very-well represented by

the first 5 EOF modes, as suggested by the correlations between reconstructed and full errors (Figure 6). Considering these truncation lengths, the PPE mean parametric component represents $80\%$ of the full PPE mean error for the TOA SW fluxes, but only $66\%$ for the sea level pressure.

The error reconstructions presented on Figure 6 are the sums of the parametric components of the errors $p_j(\theta_i)$ and the PPE

mean non-parametric components $u_{j,mean}$. As expected, the PPE mean non-parametric components decrease as higher EOF modes are retained for the reconstruction but is never equal to 0 (even for a full reconstruction of $q = 102$). This is due to the fact that observations can never be fully captured by their projections into the model EOF basis (Figure 6). As presented before, the parametric component $p_j(\theta_i)$ can be emulated with multi-linear regressions, and the PPE mean non-parametric component $u_{j,mean}$ can be used as an approximation to reconstruct the full error $e_j(\theta_i)$. This method succeeds to produce high correla-

tions between the reconstructions and the actual full model errors among the PPE, with an offset due to the non-parametric component variability across the PPE, which decreases when retaining more EOF modes. Even though higher EOF modes are not well predicted by the emulators (Figure 5), they also explain small fractions of the model error variances. As a result,

the performances of the emulators to predict model errors are much more sensitive to the climatic field considered than to the number of EOF modes retained.

On the other hand, the reference calibration CNRM-CM6-1 remains one of the best models of the PPE for most of the climatic fields and can be considered as near-optimal in the ensemble. Therefore, its model bias can be seen as a representative of the CNRM-CM discrepancy term. Indeed, the reference CNRM-CM6-1 is the best model for surface pressure and one of the best for precipitation and TOA fluxes, but several PPE members outperform it for surface temperature. This is a simple

illustration of a complex tuning problem, and based on the results we obtained in the uni-variate application. It seems likely that comparably performing parameter configurations potentially exist for a multi-variate tuning problem, making different model trade-offs among both climatic fields and EOF modes representations of uni-variate errors (Figure 3). In the next Section, we will attempt to identify some of them, in order to illustrate the different choices that could be made when tuning a climate model.

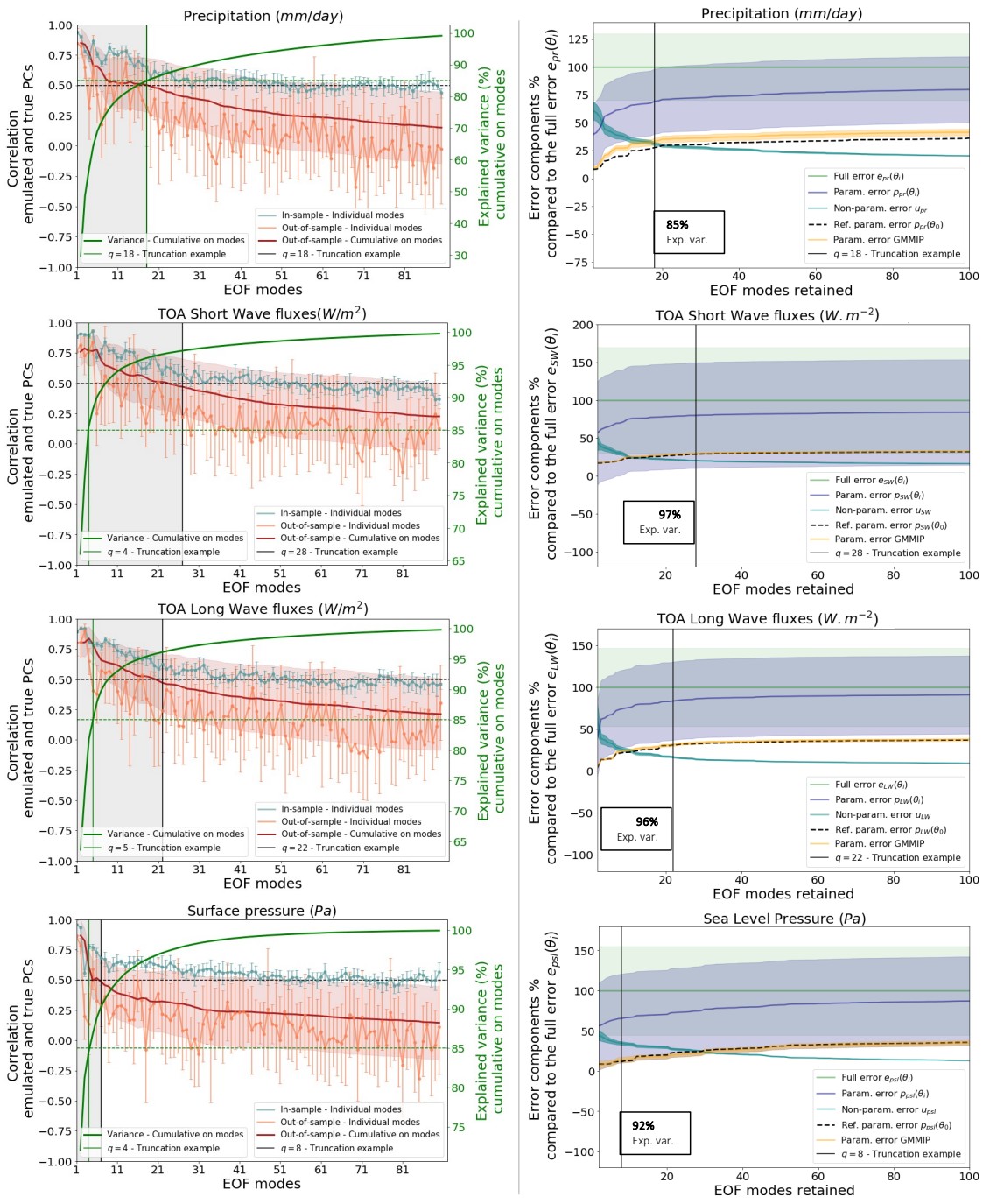

**Figure 5.** Truncation choice based on parametric emulation and error decomposition for 5 climatic fields : surface temperature, precipitations, TOA SW fluxes, TOA LW fluxes and surface pressure. Same legend as Figure 2, the observations used are listed in Table 1.

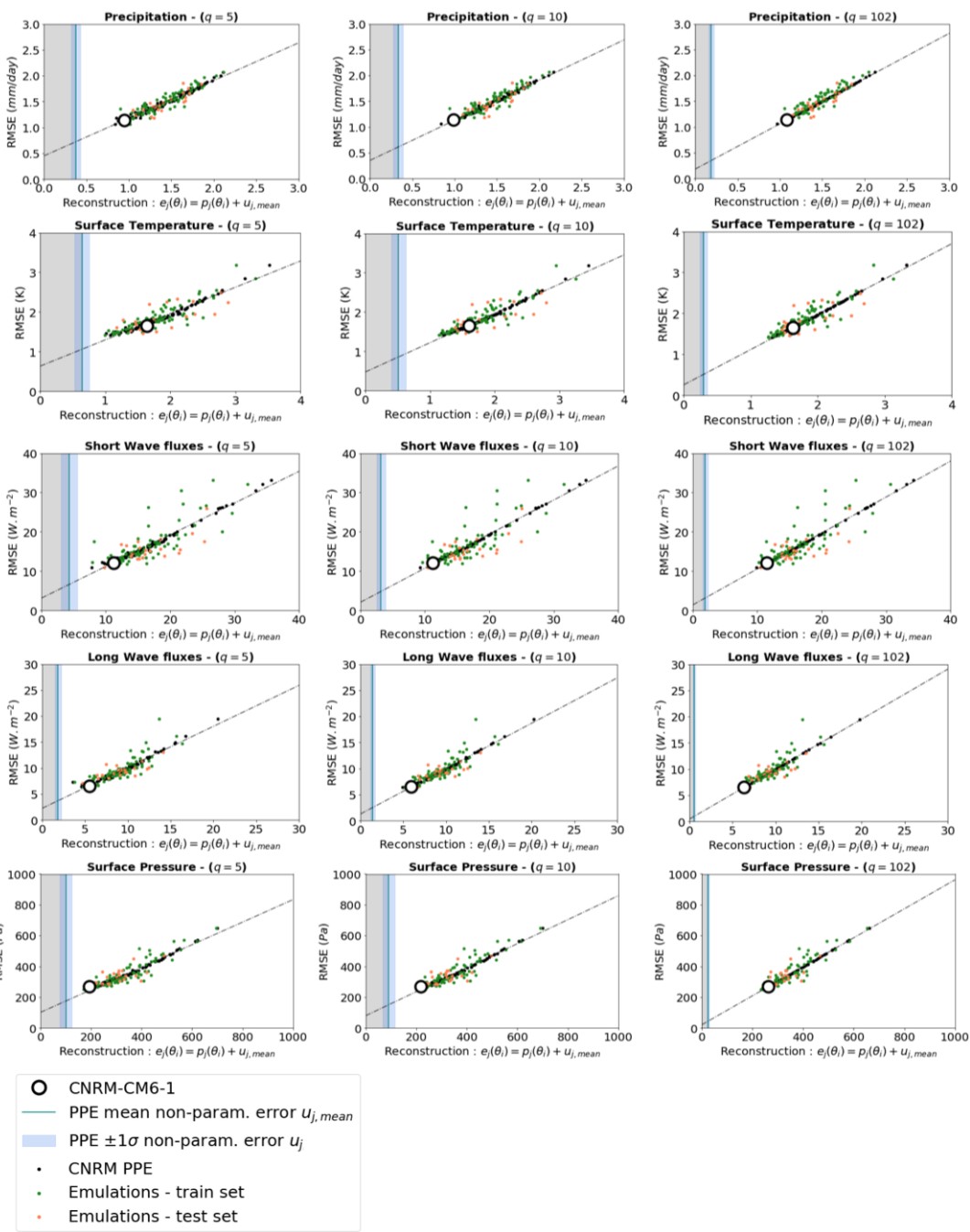

**Figure 6.** Correlations between full errors (coordinate) and EOF-based reconstructions of these errors (abscissa) using different truncation examples : retaining 5 (left column), 10 (center column) and 102 EOF modes (right column). Results are presented for the CNRM PPE (black dots) and for statistical predictions of the PPE using linear regressions trained on 80% of the data (green dots) and tested on the other 20% (orange dots). For each PPE member or emulation, the error reconstruction is the sum of the parametric component of the errors $p_j(\theta_i)$ and the PPE mean non-parametric component $u_{j,mean}$ (blue line). The variability of $u_j$ among the PPE is represented by the standard deviation $\sigma$ and the range $\pm 1\sigma$ (light blue shading).

## 4.2  Candidate selection in a multi-variate context

The results in terms of integrated multi-variate skill scores $e_{tot}(\theta_i)$ are presented in the Figure 7. Among the 12 selected candidates, 2 lead to an incomplete simulation and will not be presented here. None of the 10 remaining candidates (light blue dots) show a multi-variate skill-score lower than the CNRM-CM6-1 reference model (orange dashed line). However, all of them have a lower error than the PPE mean (red disk) and 3 of them are in the low tail of the PPE distribution (bellow the red dash). Moreover, most of the CMIP6 models have undergone a tuning process and are considered to represent the control climate satisfactorily. We can therefore use the CMIP6 ensemble as an indicator of the tolerance that can be given to this multi-variate error. Here we considered the outputs of 40 CMIP6 models that have been interpolated onto the CNRM-CM grid before computing the error. It appears that 9 CNRM-CM candidates selected here have a lower error than the mean of the 40 CMIP6 models (green disk). These 9 CNRM-CM candidates are part of the interval of plus or minus one standard deviation of the CMIP6 error centered around the error of the CNRM-CM6-1 reference model (orange area), indicating that they can be considered "as good as" the CNRM-CM6-1 reference model given the tolerance considered here. The 10th candidate is above this interval, but is still very close to the CMIP6 ensemble mean and better performing than several CMIP6 models.

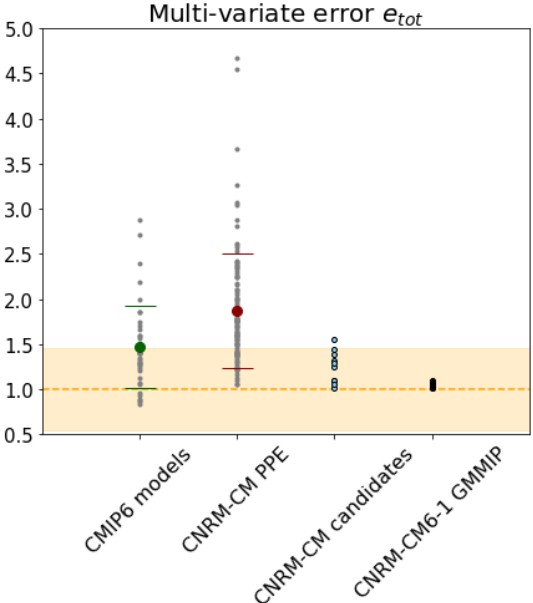

**Figure 7.** Multi-variate error $e_{tot}$ for the CMIP6 models, the CNRM-CM PPE members, the selection of 10 CNRM-CM candidates and the GMMIP dataset. Each small dots correspond to a model, the bigger dots correspond to the ensemble means and the dashes are the standard deviations. The orange dashed line at $1.0$ represents the CNRM-CM6-1 reference model error. The orange area indicates the interval of plus or minus one standard deviation of the CMIP6 errors, centered around the CNRM-CM6-1 reference model error.

The Figure 7 is also presenting the multi-variate error among the 10 simulations from the GMMIP dataset. The 10 perturbed parameter candidates are much more diverse in terms of integrated model error that the 10 perturbed initial conditions mem-

bers. When considering a multi-variate score, it is clear that the effect of internal variability is very small compared to the effect of varying the model parameters.

### 4.3   Diversity of error patterns among candidates

As described in Figure 7, the 10 CNRM-CM candidates present a satisfactory multi-variate error compared to the CMIP6
ensemble, with 9 of them performing comparably to the CNRM-CM6-1 reference model, while showing a significant diversity compared to the CNRM-CM GMMIP ensemble. We are now interested to see how this diversity translates in terms of spatial patterns of the uni-variate errors and trade-offs among the variables.

Here again, for practical reason, a sub-set of 4 candidates is presented in Figure 8, the rest of the candidates can be seen
in Figure E1. Within the 4 candidates presented in Figure 8, we have selected the best performing model (Candidate 5) and the worst performing model (Candidate 1). All the candidates have features common to the CNRM reference model (Roehrig et al., 2020) : an over-estimate of the tropical precipitation and large SW fluxes biases over the mid-latitude eastern border of Atlantic and Pacific oceans. However, they all have a better representation of surface temperature than the reference model. None of the candidate show a better representation of precipitation, sea level pressure or LW outgoing fluxes than the reference
and Candidates 1, 2, 5 and 9 provide a better representation of the SW outgoing fluxes.

Moreover, some differences exist between the spatial patterns of the candidate errors. Candidate 10 is the model configuration with the lowest MSE of precipitation ($e_{pr}(\hat{\theta}_{10}) = 2.18\ mm/day$, still higher than the reference), but is also showing important tropical biases in the radiative fluxes (positive in SW and negative in LW), in the same regions were the model over-
estimate the tropical precipitation, suggesting a biased representation of tropical clouds. Candidate 5 on the other hand, have a better representation of the radiative fluxes in these regions, with a better representation of SW fluxes than the reference and the best representation of LW among the candidates, suggesting a better representation of tropical clouds. However, candidate 5 is presenting the same biases in precipitation as Candidate 10, with an even higher MSE.

Candidate 1 in the worst performing model of the whole selection, with a total MSE of $e_{tot}(\hat{\theta}_1) = 1.56$. This is mostly due to important biases in precipitation, sea level pressure ans LW fluxes representations. Candidate 1 presents strong positive tropical biases in LW fluxes, over the northern part of South America, central Africa and Indonesia. These areas corresponds to dry biases in the map of precipitation. Over the tropical oceans, it is one of the candidate that is not showing the negative LW and positive SW tropical biases, other examples can be found in candidates 3, 8 and 9 (Figure E1). These candidates all
have positive LW biases over the tropical continents, and less biases over the tropical oceans. Interestingly, this is one of the candidate with the best representation of SW fluxes, with candidate 9 (Figure E1), which has a lower MSE. The SW fluxes biases over the mid-latitude eastern border of Atlantic and Pacific oceans seem to be reduced in these 2 candidates compared to the other models and the reference.

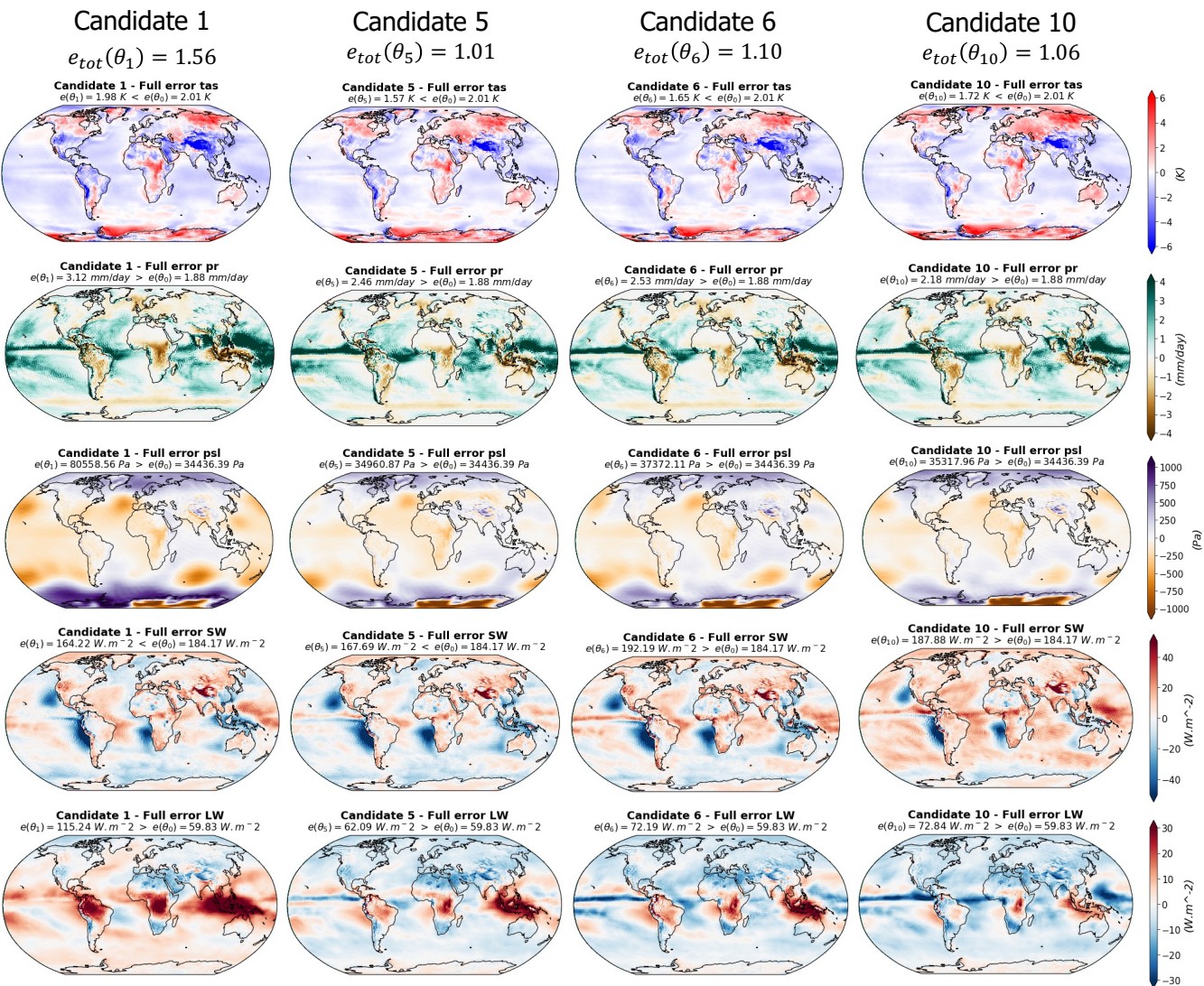

**Figure 8.** Differences between the simulations and the observations (Table 1), for the 4 model candidates and the 5 variables considered : surface temperature, precipitation, sea level pressure, short and long waves top-of-atmosphere fluxes. Each column represents a model candidate and each raw corresponds to a variable. The green dots highlights the cases for which the RMSE is lower than the CNRM reference model.

Candidate 5 is the best performing candidate in terms of multi-variate score and shows errors lower than the CNRM reference
model for surface temperature and SW fluxes (Figure 8). Candidate 5 shows a LW error map similar to candidate 6 an d 10 but with a reduction of the biases amplitudes. We can assume that the model is better representing tropical clouds, but this does not translate into the best representation of tropical precipitation within the selection.

### 4.4 Examples of discrepancy term decomposition

Following the method described in Section 2.4, the full error patterns presented in Figure 8 can be decomposed into a parametric components (Figure 9) and non-parametric components (Figure 10). The EOF truncation lengths used for this decomposition are based on the examples given in Figure 5 : 18 modes for *tas* and *pr*, 8 modes for *psl*, 28 for *SW* and 22 for *LW*.

As expected, the candidates parametric component error patterns resemble the full error patterns, with as much diversity in between the candidates (Figure 9). The non-parametric components on the other hands, are more patchy, are smaller in terms of amplitude and are common to all the candidates (Figure 10). This validates the method : we were able to select a set of candidates with diverse error patterns and to isolate the error component that is unaffected by parameter variation from the component that varies during model tuning.

A notable feature of these candidate error decompositions is the SW error patterns. The non-parametric component of the SW error appears very patchy, but contains a small part of the negative biases over the oceanic mid-latitude eastern border that we described in the full error patterns, directly at the continental border (Figure 10). The main part of these biases are presented in the parametric component of the error (Figure 9). This result suggests that such biases could be enhanced or reduced by varying the model parameters, but part of them is non-parametric and directly linked to the physics of the model.

In conclusion, when considering error patterns and multi-variate illustration, the effective degrees of freedom in model performance optimization might be smaller than expected. Our method allowed for an empirical exploration of the key trade-offs that could be made during the tuning, providing interesting information about model non-parametric biases and examples of alternative model configurations.

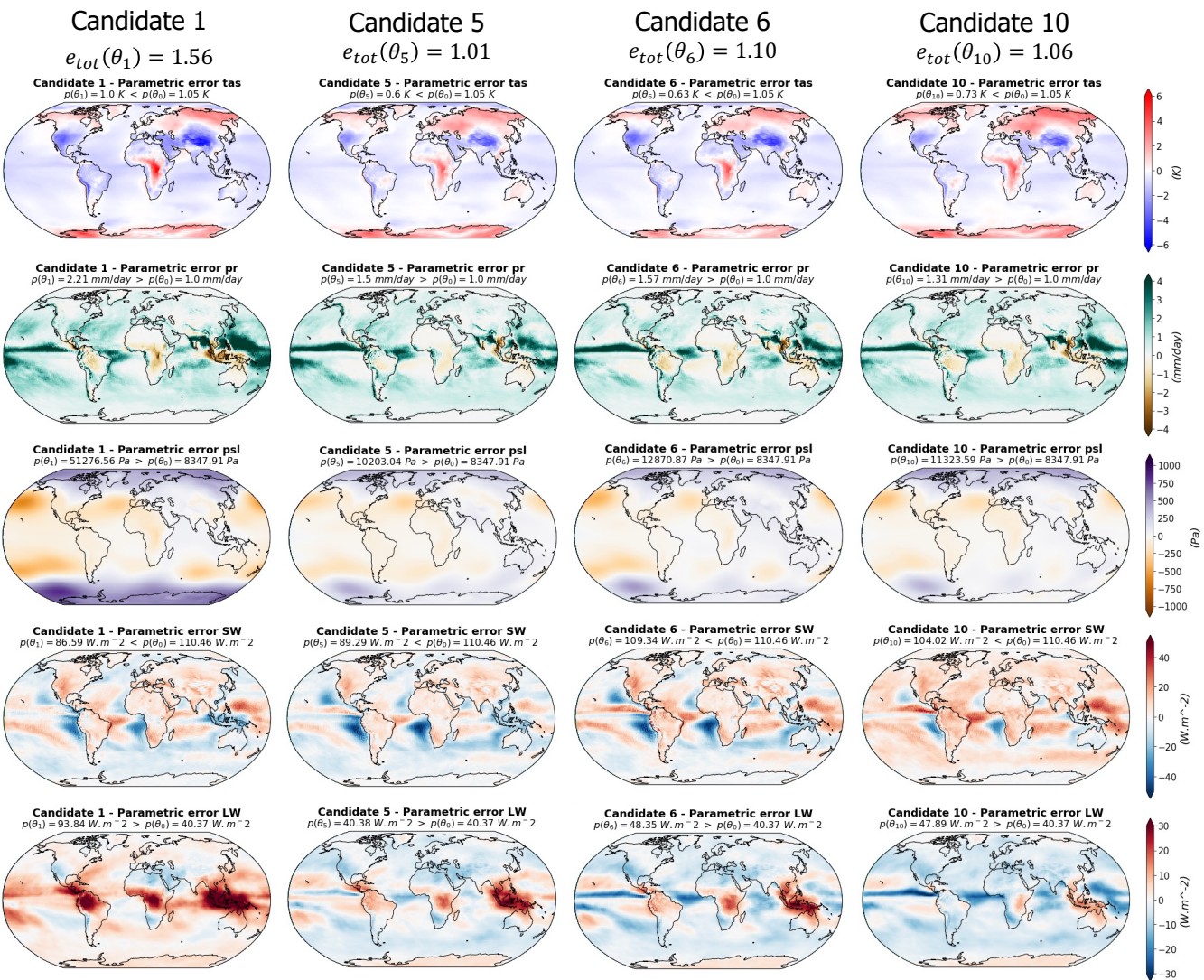

**Figure 9.** Parametric component of the differences between the simulations and the observations (Table 1), for the 4 model candidates and the 5 variables considered. Each column represents a model candidate and each raw corresponds to a variable. The decomposition of model errors in parametric and non-parametric components is based on the methodology described in the Section 2.4, with the EOF bases truncated following examples given in Figure 5 : 18 modes for *tas* and *pr*, 8 modes for *psl*, 28 modes for *SW* and 22 modes for *LW*.

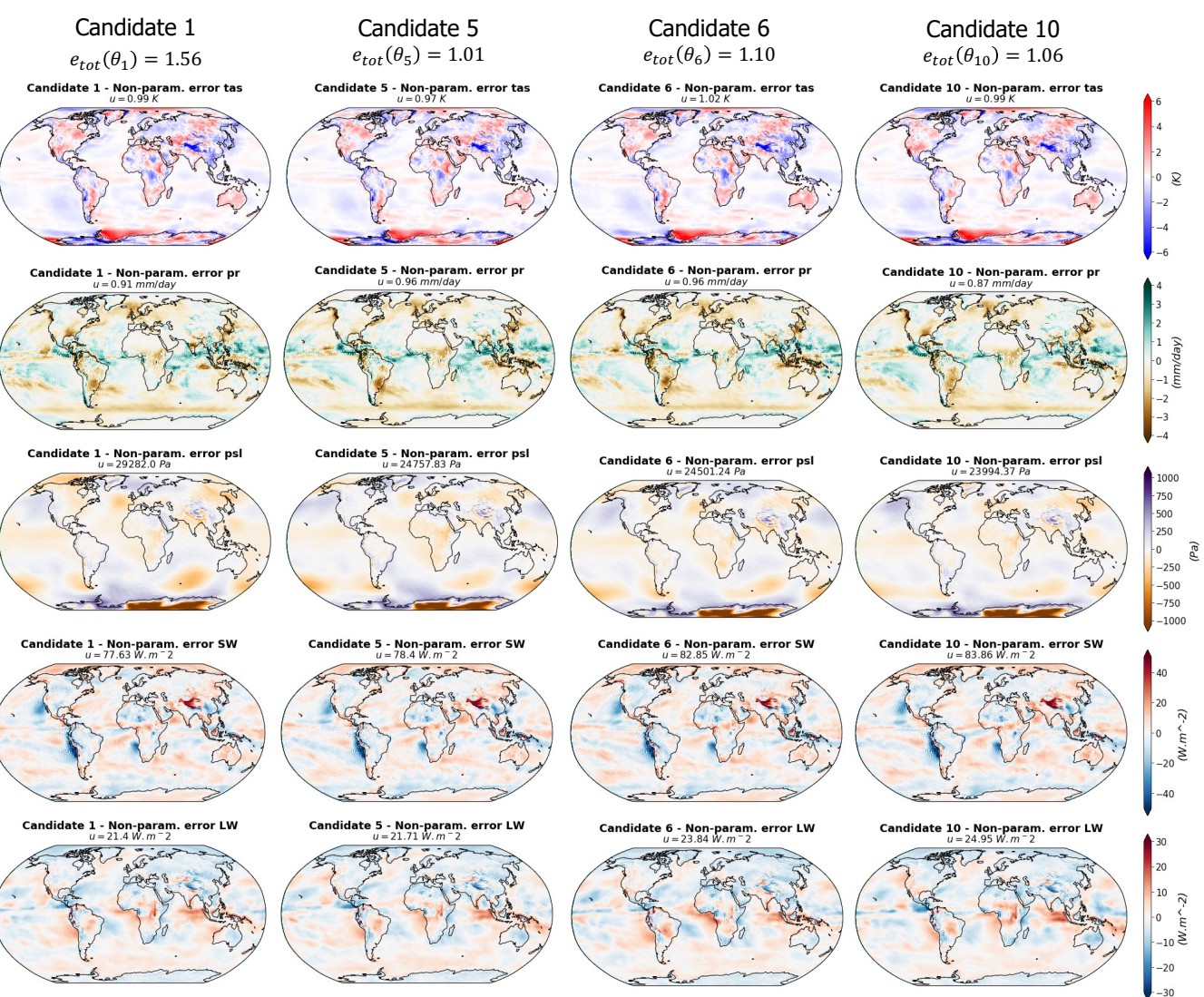

**Figure 10.** Non-parametric component of the differences between the simulations and the observations (Table 1), for the 4 model candidates and the 5 variables considered. These components are the differences between the Figure 8 and 9.

## 5 Conclusions

This study presented a new framework, based on a PPE of a CMIP6 General Circulation Model, allowing for the empirical selection of diverse near-optimal candidate calibrations. We have demonstrated that this approach is practically useful for different reasons :

1. The effective degrees of freedom in model performance response to parameter input are in fact relatively small, allowing a convenient exploration of key tradeoffs

2. Higher modes of variability should not be included because they cannot be reliably emulated and they do not contribute significantly to the component of model error controlled by model parameters

3. As such, reference model version shows the lowest integrated performance metric and historical common practices for parameter tuning could be more robust than often assumed

4. However there remains potential for comparably performing parameter configurations making different model trade-offs

Using the 'best input' assumption, Rougier (2007) we assume that these candidates sample the distribution of atmospheric model discrepancy term. These discrepancy term can be decomposed in parametric and non-parametric components using PPE derived EOF basis. The candidates are selected from a PPE of the CNRM-CM atmospheric model. The optimization is based on multi linear predictions of the parametric components of the model errors, from a 100,000 LH sampling of the perturbed parameters. The candidates are considered near-optimal when their emulated parametric components are lower than the reference parametric component and are selected to exhibit pattern errors as diverse as possible within this near-optimal sub-space using a k-medians clustering algorithm. As such, the sub-set of candidates offer a diversity of model errors sampling the CNRM-CM model discrepancy term distribution, while exploring different trade-offs.

The decomposition of the discrepancy terms depends on the truncation choice : the non-parametric component increases when retaining more EOF modes, at the expense of the parametric component. However, we argue that there are no particular benefits in retaining high-order EOF modes, for two reasons. Firstly, the performance of the predictions quickly decreases for the high rank EOF modes, which suggests that these modes are not very predictable from the parameter values. Then, the fact that the first few modes are sufficient to reconstruct the PPE variance of the model errors for the 5 climatic fields considered here and that high modes explain a very small fraction of the PPE variance. Therefore, retaining more EOF modes will increase the part of the model error represented by the EOF basis, called the parametric component, but will not improve the optimization.

In the first step, the method was validated for surface temperature error, revealing a diversity of trade-offs among different EOF modes when considering diverse but near-optimal candidates. These trade-offs indicate the presence of a parametric component in the discrepancy terms, which no candidates could eliminate completely. The non-parametric component, on the other hand, is independent of parameter choice and very similar from one candidate to another. These model candidate errors are considered to represent empirical examples of the model discrepancy term for temperature and can offer insights for model developers.

In the second step, the framework was applied in a multi-variate context. Trade-offs were observed in error patterns across climatic fields, with different candidates excelling in various aspects. All of the candidates were selected with an emulated

parametric error lower than the reference but showed, in practice, higher parametric errors. This result can be attributed to the limitations of the emulators. However, as discussed in Appendix C, our capacity to train emulators is fundamentally limited by the sample size available, which rather small in this study (102 simulations). The use of a non-linear emulator, such as a Gaussian Process, often used in automatic tuning applications Williamson et al. (2013); Hourdin et al. (2023), could help improve predictions, provided we can increase the size of the PPE. In summary, 9 candidates achieved integrated multi-variate scores within CMIP6 ensemble standards, but none of them performed better than the reference model.

Though we do not attempt it here, the discrepancy estimate could be used in parallel with a history matching approach such as (Salter et al., 2019), or a Bayesian calibration (Annan et al., 2005) to yield a formal probabilistic result. Enhancing the PPE size would allow for better statistical predictions, maybe through the use of Gaussian Processes as statistical models. We could also consider seasonal metrics instead of annual average, as suggested in (Howland et al., 2022). Another important caveat of this study is that we did not the observational uncertainty. Indeed, additional analyses suggest that our results are sensitive to the observational dataset used. Therefore, defining a formal way to include observational uncertainty in our method for candidate selection would be a valuable improvement of the method. Finally, performing sensitivity analysis could help better understand the effect of each parameters on the biases we observed, potentially leading to a selection of a meaningful sub-set of parameter for a new wave of simulation, in an iterative process.

In summary, we argue that the model discrepancy term can be represented as a sum of two parts - a component which is insensitive to model parameter changes, and a component which represents parameter trade-offs, which manifest as an inability to simultaneously reduce different components of the model bias (e.g. in joint optimization of different regions or fields). We further argue that parameter calibration by hand could be more tractable than often assumed and the reference versions may often be the best model configuration achievable in terms of integrated multi-variate metrics. A feature we see evidenced here by the high performance of the reference simulation, but also reported in similar past PPE efforts (Sanderson et al., 2008; Li et al., 2019). Finally, we demonstrate a practical method for utilising these concepts for the identification of a set of comparably performing candidate models can inform developers on the diversity of possible trade-offs. The selection of diverse candidates can help better understanding the limits of model tuning to reduce model error, identify non-parametric biases that are not visible when looking at the full model error and help choose the model configuration best suited to the research interest. Moreover, the diversity of model errors can reflect a diversity of future climate responses (Peatier et al., 2022; Hourdin et al., 2023)) and selecting diverse candidates will help the quantification of uncertainty in climate change impact studies.

## Appendix A: Perturbed parameter

**Table A1.** Description of the 30 perturbed parameters

| Name | Minimum | Maximum | Reference | Description | Units |
|------|---------|---------|-----------|-------------|-------|
| AKN | 0.06 | 0.28 | 0.126 | Strength of the turbulent mixing | - |
| ALPHAT | 0.5 | 3.0 | 1.13 | Strength of the turbulent mixing for temperature (Prandtl number) | - |
| ALD | 0.5 | 3.0 | 1.18 | Strength of the turbulent kinetic energy dissipation | - |
| ALMAVE | 0 | 30 | 10 | Lower bound of the mixing length | m |
| AGREF | $-0.5$ | $-0.01$ | $-0.36$ | Parameter in the boundary-layer-top entrainment parameterization | - |
| AGRE1 | 0 | 10 | 5.5 | Parameter in the boundary-layer-top entrainment parameterization | - |
| AGRE2 | 0 | 10 | 0 | Parameter in the boundary-layer-top entrainment parameterization | - |
| RAUTEFR | $0.5 \times 10^{-3}$ | $1 \times 10^{-2}$ | $1 \times 10^{-3}$ | Inverse timescale for liquid autoconversion | $s^{-1}$ |
| RQLCR | $0.5 \times 10^{-4}$ | $1 \times 10^{-3}$ | $2 \times 10^{-4}$ | Critical liquid water content for liquid autoconversion | $kg\,kg^{-1}$ |
| RAUTEFS | $0.5 \times 10^{-3}$ | $1 \times 10^{-2}$ | $5.2 \times 10^{-3}$ | Inverse timescale for ice autoconversion | $s^{-1}$ |
| RQICRMIN | $0.1 \times 10^{-5}$ | $0.1 \times 10^{-7}$ | $0.1 \times 10^{-6}$ | Critical ice content for ice autoconversion at low negative temperatures | $kg\,kg^{-1}$ |
| RQICRMAX | $0.05 \times 10^{-4}$ | $1 \times 10^{-4}$ | $0.21 \times 10^{-4}$ | Critical ice content for ice autoconversion at high negative temperatures | $kg\,kg^{-1}$ |
| TFVL | 0.001 | 0.2 | 0.02 | Falling speed of cloud water droplets | $m\,s^{-1}$ |
| TFVI | 0.001 | 0.2 | 0.04 | Falling speed of cloud ice crystals | $m\,s^{-1}$ |
| TFVR | 0.1 | 6.0 | 3.0 | Falling speed of rain | $m\,s^{-1}$ |
| TFVS | 0.1 | 6.0 | 0.6 | Falling speed of snow | $m\,s^{-1}$ |
| RKDN | $3 \times 10^{-5}$ | $7 \times 10^{-5}$ | 5e-05 | Minimum drag for the convective updraft vertical velocity | $Pa^{-1}$ |
| RKDX | $8 \times 10^{-5}$ | $6 \times 10^{-4}$ | $1 \times 10^{-4}$ | Maximum drag for the convective updraft vertical velocity | $Pa^{-1}$ |
| TENTR | $2 \times 10^{-6}$ | $1 \times 10^{-5}$ | $4 \times 10^{-6}$ | Minimum turbulent entrainment in the convective updraft | $Pa^{-1}$ |
| TENTRX | $3 \times 10^{-5}$ | $1 \times 10^{-4}$ | $6 \times 10^{-5}$ | Maximum turbulent entrainment in the convective updraft | $Pa^{-1}$ |
| VVN | $-1$ | $-5$ | $-2$ | Critical convective updraft Vertical velocity for maximum entrainment and drag | $Pa\,s^{-1}$ |
| VVX | $-25$ | $-50$ | $-35$ | Critical convective updraft Vertical velocity for minimum entrainment and drag | $Pa\,s^{-1}$ |
| ALFX | 0.01 | 0.1 | 0.04 | Maximum convective updraft area fraction | - |
| FNEBC | 0 | 20 | 10 | Parameter for computing the convective cloud fraction | - |
| RLWINHF_ICE | 0.5 | 1.0 | 0.9 | Ice cloud heterogeneity coefficient in the longwave spectrum | - |
| RLWINHF_LIQ | 0.5 | 1.0 | 0.9 | Liquid cloud heterogeneity coefficient in the longwave spectrum | - |
| RSWINHF_ICE | 0.5 | 1.0 | 0.71 | Ice cloud heterogeneity coefficient in the shortwave spectrum | - |
| RSWINHF_LIQ | 0.5 | 1.0 | 0.71 | Liquid cloud heterogeneity coefficient in the shortwave spectrum | - |
| RELFCAPE | 0.2 | 10.0 | 2.0 | Parameter used in the convection scheme Convective Available Potential Energy closure | - |

## Appendix B: Clustering analysis and sensitivity to the number of clusters

Clustering is a data mining technique that divides a dataset into different categories, based on the similarity between data. The k-medians analysis is a centroid-based algorithm which divide the data into $k$ categories in order to maximize the similarity of data within a same cluster (Hastie et al., 2009; Pedregosa et al., 2011). Here, the index to measure similarity between data is the Euclidean distance. The k-medians analysis is sensitive to the choice of cluster numbers $k$, which depends on the dataset being classified.

The inertia can help to estimate how well a dataset was clustered by k-medians. It is defined as the sum of the squared distances between each data point and the centroid within a same cluster. The Elbow method consists in finding the inflexion point in the k-means performance curve, where the decrease in inertia begins, to find the good trade-off : a good model is one with low inertia and low number of clusters $k$ (Cui et al., 2020). Another criteria we can look at is the Dunn index : the ratio between the minimal inter-cluster distances and the maximal intra-cluster distances. A higher Dunn index represents a higher distance in between the centroids (clusters are far away from each other) and a lower distance in between the data points and the centroid of a same cluster (clusters are compact).

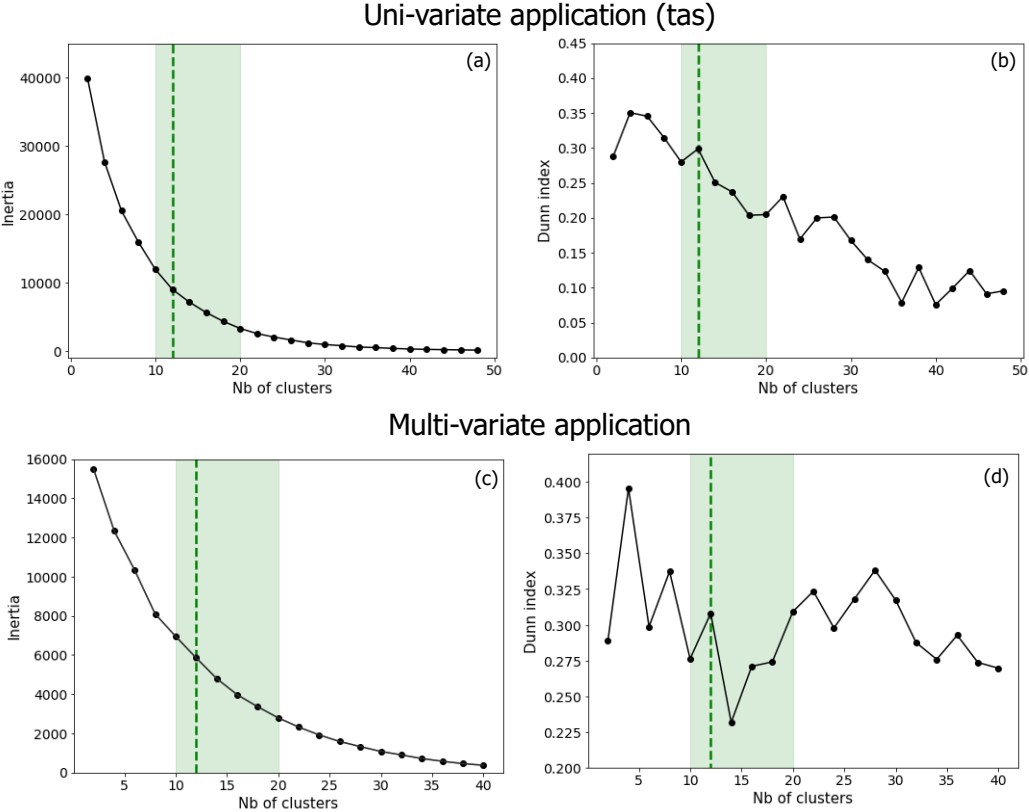

**Figure B1.** Sensitivity test of the clustering analyses for the uni-variate (first row) and multi-variate (second row) applications. The inertia criteria ((a), (b)) and the Dunn indexes ((b), (d)) are shown depending on the number of clusters (x axes). The green shaded areas present the acceptable number of clusters following the Elbow method applied to the inertia. The green dashed line shows the number of clusters retained for our analyses : $k = 12$ in both applications.

For both applications, the k-medians analysis is repeated 10 times for different values of $k$ and the average of inertia and Dunn indexes are presented in Figure B1. The inertia sensitivity test suggests that we could chose a value of $k$ in between 10 and 20 to be in the Elbow of the curve, for both applications. Then, even though it is less obvious for the multi-variate

application, the results suggest that we should not take a value of $k$ too high, as the Dunn index tends to decrease. Based on these two criteria, we have decides to keep 12 number of clusters for the analysis : $k = 12$.

## Appendix C:  Evaluation of the statistical predictions

The emulators used in this study are Multi Linear Regressions (MLR) taking the model parameters as input and predicting the Principal Components (PC) used to reconstruct the 3D variables and the parametric model errors when comparing with observations. The ensemble size of the PPE is very limited (102 simulations) and our capacity to train emulators is fundamentally limited by the sample size available. However, in 10 random selections of out-of-sample test sets, we obtain an average correlation of 0.7 between the predictions and the true values of total error (Figure C1 (c)), with a RMSE between predictions and true values representing $8\%$ of the total parametric error (Figure C1 (f)), which is sufficient to validate the use of this model for our study.

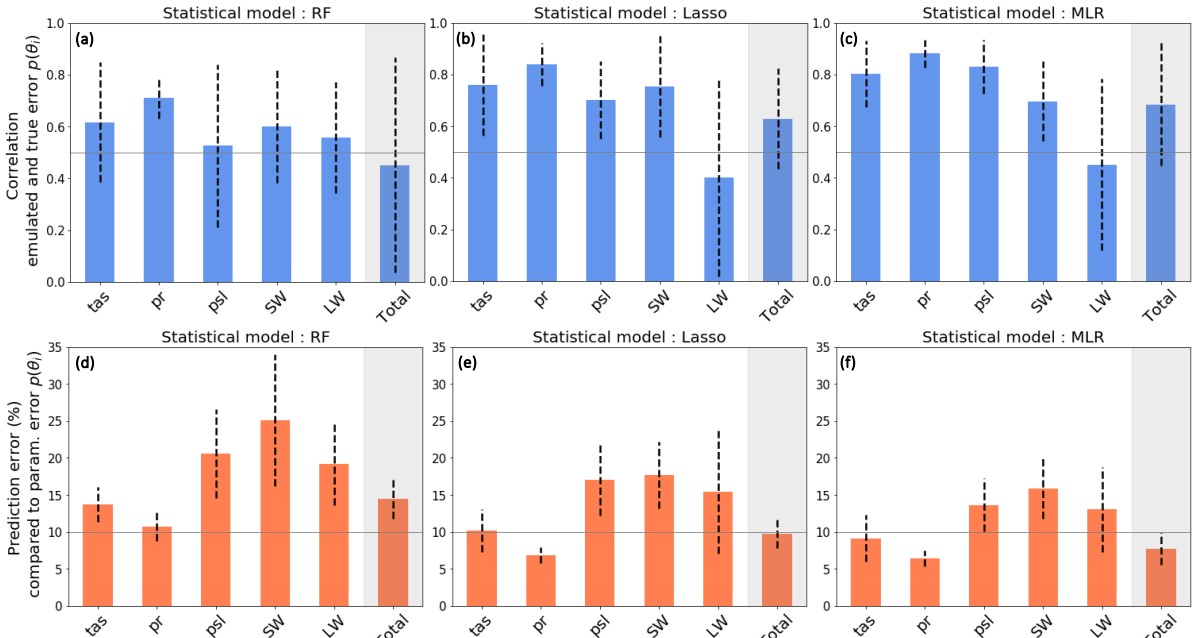

**Figure C1.** Correlations and RMSE (in $\%$ compared to the true values) between emulated and true parametric components of the errors within a test set representing $10\%$ of the dataset. The evaluation is repeated 10 times with random sampling of training and test sets and the mean and standard deviation among these 10 evaluations are represented by the bars and the dashed lines, respectively. Performances are shown for (a), (d) a Random Forest, (b), (e) a LASSO regression and (c), (f) the Multi Linear Regression used in this analysis and. The EOF truncation lengths used to compute the parametric error are presented in Figure 2 and 5.

However, results suggest that there is room for improvement, especially in the prediction of the LW errors, and that another model could improve the predictions, as it is the case of the Random Forest model. The error bars associated with the prediction

of the total error suggests that the MLR performance is sensitive to the test set selected and that the model will perform unevenly across the parameter space. Thanks to variable selection and regularization, the Lasso model seems a bit less sensitive to the test set selection for the prediction of total error, but the prediction of LW error is still a limitation. It seems that using a non-linear emulator could improve certain aspects of the predictions, though enhancing the size of the ensemble would be a necessary prerequisite to try to improve our statistical predictions.

# Appendix D: First application : additional candidates

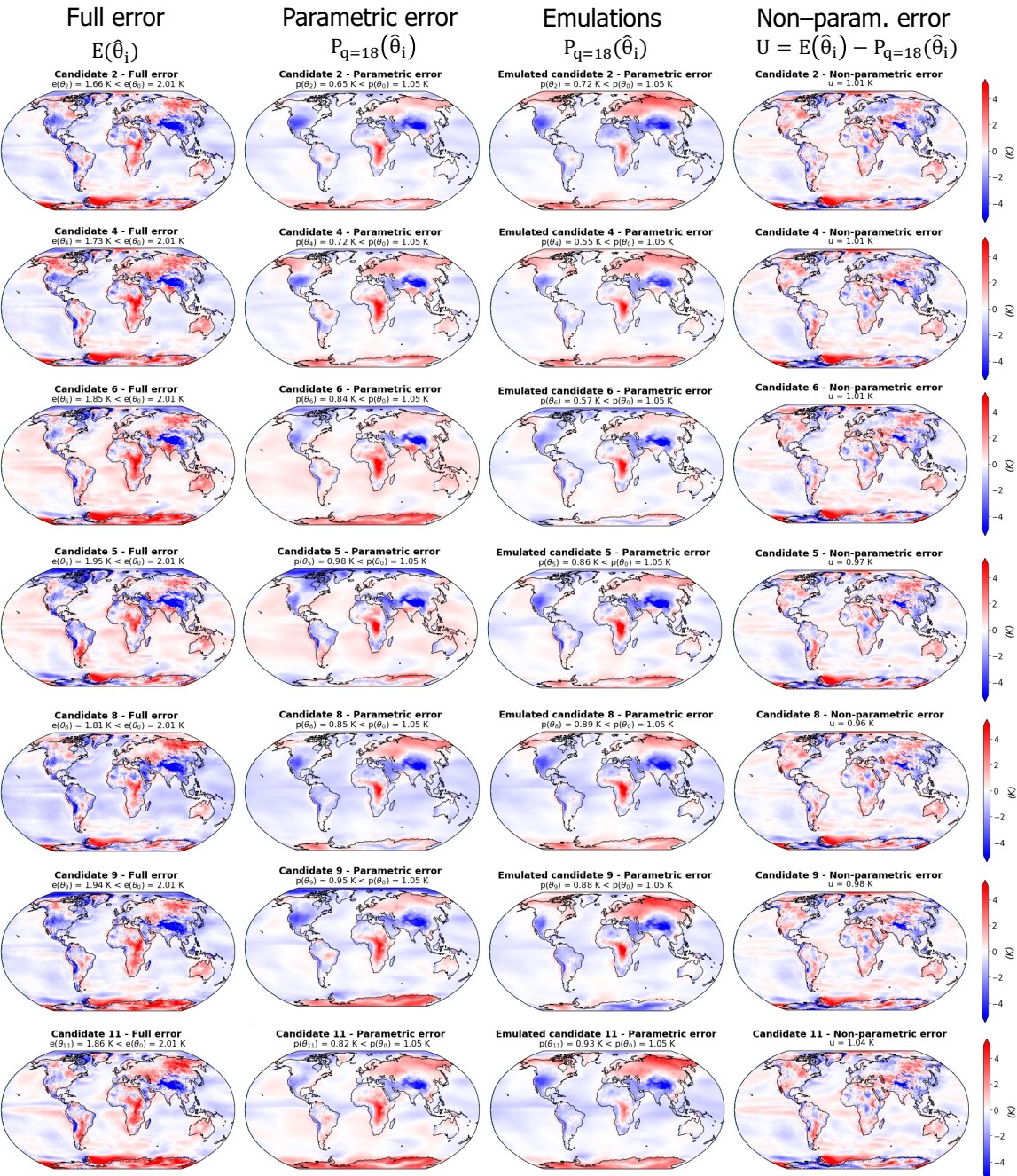

**Figure D1.** Decomposition of surface temperature error in the first sub-set of candidates. Same caption as Figure 4 for additional candidates.

# Appendix E: Second application : additional candidates

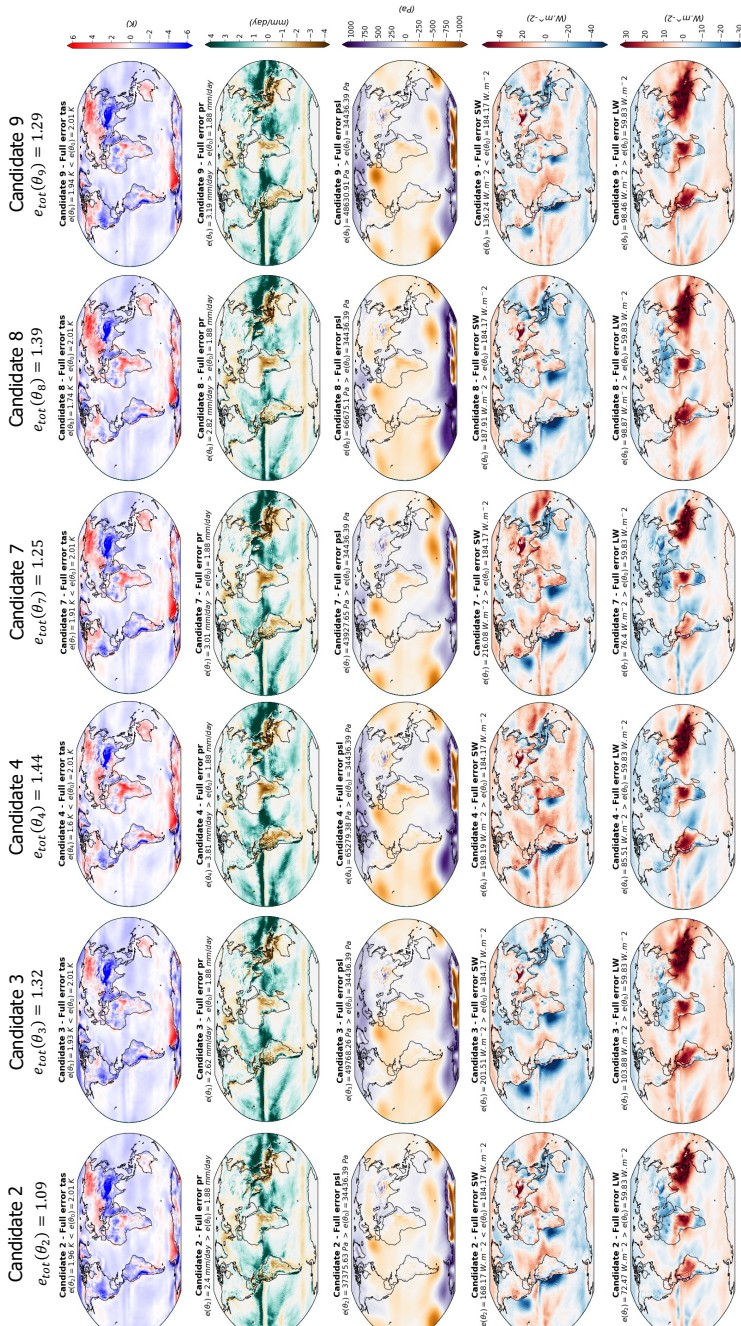

**Figure E1.** Full model errors in the second sub-set of candidates. Same caption as Figure 8 for additional canidates.

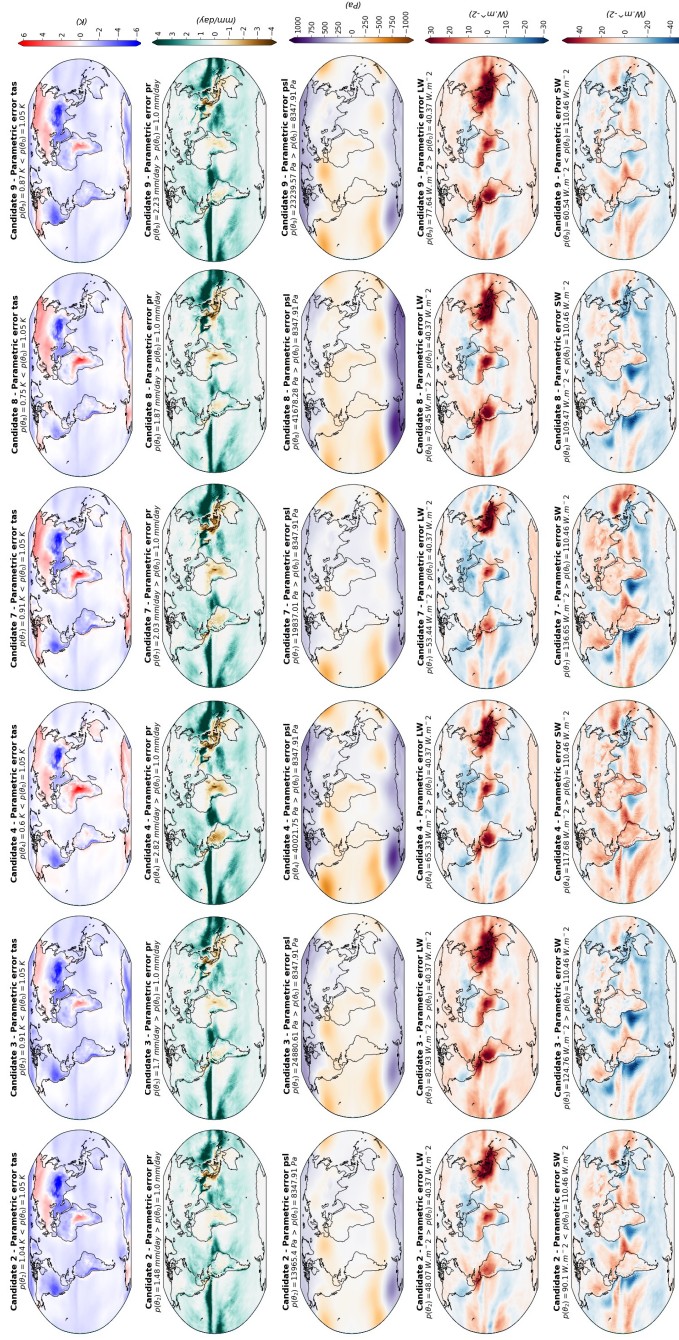

**Figure E2.** Parametric model errors in the second sub-set of candidates. Same caption as Figure 9 for additional canidates.

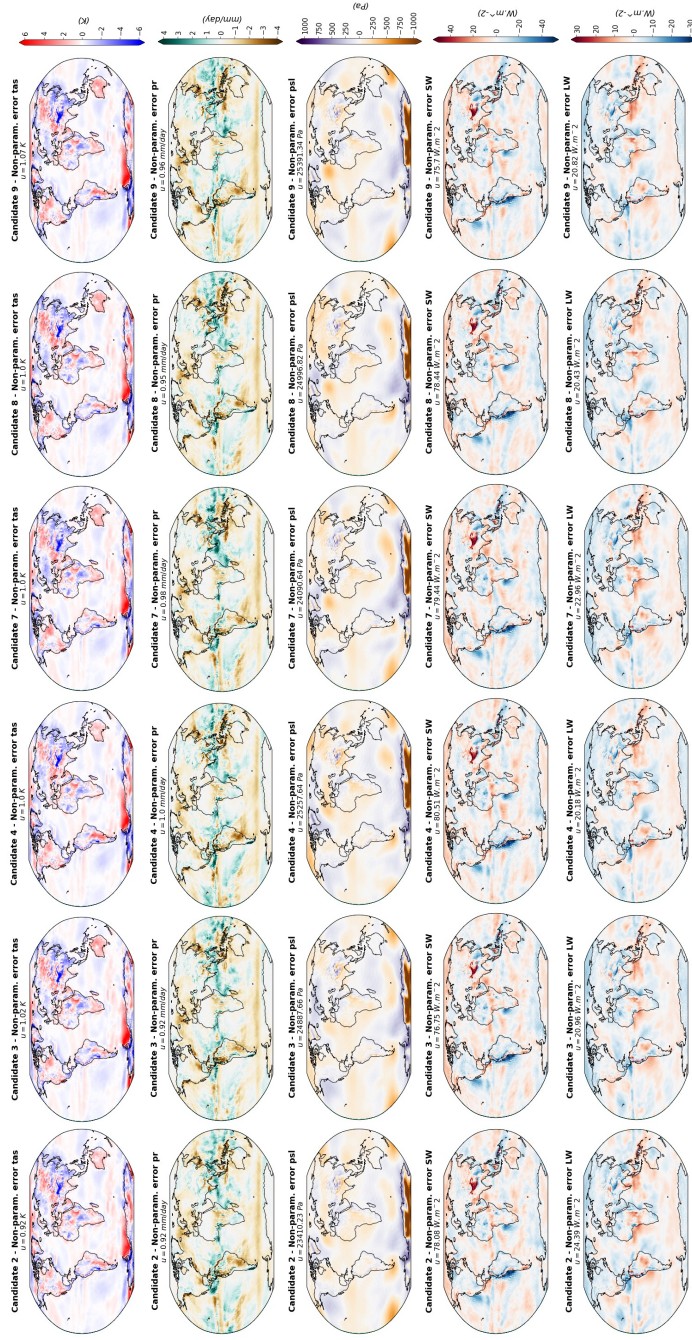

**Figure E3.** Non-parametric model errors in the second sub-set of candidates. Same caption as Figure 10 for additional canidates.

*Author contributions.* SP carried out the simulations and the analysis. SP prepared the manuscript with contributions from all co-authors. BS developed the initial theoretical formalism. SP, BS and LT conceived the analysis. LT supervised the findings of this work.

*Competing interests.* The authors declare that they have no conflict of interest.

*Acknowledgements.* This work is partly funded by the French National Research Agency, project no. ANR-17-MPGA-0016. BS, LT and SP are supported by the H2020 project ESM2025, (Grant ID: 101003536). The authors thank the CNRM-CERFACS modeling group for developing and supporting the CNRM-CM6-1 model.

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
