# Peer review of "Addressing discrepancy in the calibration of a climate model"

_EGUsphere, 2023_

## Referee Comment (RC1)

**Review of: "On the spatial calibration of imperfect climate models" - S. Peatier, B. Sanderson, L. Terray**

**Summary**

My comments and suggestions are based on the following interpretation of the content:

Calibration of Earth System Models (ESMs) is subject to different sources of uncertainty. Principally among these are parametric (from constant values set within the ESM) and non-parametric (the residual resulting from the ESM set to values achieving minimal parametric error). To explore these errors, the authors analyze a previously-constructed perturbed physics ensemble (PPE) of the atmosphere model in CNRM-CM6 climate model. The PPE is a set of 102 simulations from a parametric sweep across 30 model parameters. The goal of the analysis is to use this PPE to produce a small number of (new) representative parameter values whose simulations achieve a low parametric error, and display diverse behavior. The methodology is as follows (i) separate parametric and non-parametric error components displayed by the PPE (ii) dimension reduction (iii) fit an emulator for parametric-error and sample it across the parameter space (iv) threshold out values with large emulated parametric-error compared to a reference (v) find values from this subset with a large "feature" distance between them. This is tested on two example data, RMSE to surface air temperature, and MSE to five descriptive fields. The approach validation is largely through presentation of the spatial fields of the final selection of five diverse cases with similar parametric and nonparametric error decomposition .

**Conclusion**

I believe this is a strongly presented and interesting paper. The literature is largely comprehensive, the methods are walked through in precise detail, and the presentation of results are given in a collection of well-chosen figures. Most of the conclusions are supported by the results. I believe the scientific perspective is interesting, and a strong proof-of-concept framework is presented.

It is not a perfect manuscript however, it suffers occasionally by informal terminology, including even in the title. The conclusions also did not convince me that the results are really consequential beyond mild scientific interest. Most critically, I believe the methods within the framework are not evaluated properly; and this leads to caveats in results where the methods used for emulation are not exactly fit for purpose, and the methods to "maximize" diversity, may not necessarily do so.

I will recommend the manuscript for publication under addressing the comments below.

**Major comments**

- **Method deficiencies (i) the emulator:** The choice and use of emulator are not discussed, but in several places in the results it is indicated that there is room for improvement here.
E.g. in Figure 3 - often the emulated values (that determine also the concept of "diversity" above) did not lead to close or hugely diverse simulations. I came to this conclusion as (1) the dots and triangles are often far from each other, and (2) the dots are typically more spread than the triangles, (3) one dot lead to a simulation crash for its corresponding triangle.
Likewise in Figure 7: 4 diverse candidates were selected with emulated $p < 1$, yet all in practice most were $> 1$. Though this was noted in the document it was not attributed to the emulation stage being unable to well-represent the underlying model.
Particularly in the results, the shortcomings due to the choice of emulator should be attributed clearly, and additional discussion added to the conclusions in terms of how to address this in future investigation.

- **Method deficiencies (ii) exploring diversity:** Perhaps the weakest part of the investigation involved the exploration of "diversity of solutions". Though I agree that the approach removes solutions that have similar parametric components, it is mentioned L284 that the authors "maximize" the diversity in solutions, or L177 "a selection … as diverse as possible", yet their approach is not a seemingly well defined optimization procedure. In fact the approach the authors use appears to be non-standard for such exploration, likewise with the termination of "5" final candidates is completely unmotivated. Finally the sensitivity to the choice of the threshold was also not explored. Typically for sourcing representative simulations a practitioner might use a clustering algorithms (k-means/medians/mediods etc.) over the data, and draw representative samples from the cluster centers to illustrate diversity. Such approaches are scalable and robust, and one can quantitatively assess sensitivity to e.g. the number of clusters the data appears, thresholding etc. The random removal methodology presented here appears untested. For example, if the authors ran the method over 100 trials, would they expect to find the same nodes each time, or if not, would they expect the selections of 5 display similar diversity, how are would they quantitatively measure this? How would they compare the performance of terminating at different numbers of solutions.
Some references and/or computational validation experiments for the approach taken are needed, otherwise, again some clear admission in the methods and final discussion that this is a proof of concept, and suggest other approaches that readers can try in future to explore diversity.
From a naive inspection of the samples in Figure 3, it does not seem out of the question that there are more representative samples one could find. I also think the use of emphatic words such as "large" diversity (L452 and elsewhere), "the key" trade-offs (L507 and elsewhere) should really be removed when one is not able to really quantify how representative these differences are, or how many trade-offs are observed due to

the selection procedure.

- **Optimality and near-optimality:** As a reader, this is the walkthrough of optimality in this manuscript.
  - An optimal solution is defined to be an exact quantity f(theta*) eqn(9).
  - This is then redefined in L158 where optimal is an input that minimized e_j.
  - Then in L168 the optimal solution is considered to be the input that minimizes an emulated objective (maybe this should be called e_{em,j}).
  - In L172 The introduction of a black-box "reference calibration" CNRM-CM (6?) is introduced as a threshold for optimal candidates (this is the first time this is mentioned in the document and it is unreferenced), it is used to rule out all candidates with greater error. It is not clear if this is calculated with real or emulated coefficients. Is the intention to have a "typical" level of accuracy, arising from current "tuning procedures"?
  - In L520 there is mention of near-optimal configurations
  - In L528 optimality and an optimal space are then unified with near optimality?

  I request that the authors solidify these notions and be consistent throughout. I also request that the authors use concrete notation for where they are using emulated vs actual scores. Finally, the threshold should be presented in a more clear fashion and explanation as to why this choice should define optimality.

- **Why explore the diversity of solutions?:** Given that the investigation is centered around exploring the diversity of these solutions of low parametric error, It was mentioned only in the final sentence as to why one wishes to do this beyond scientific curiosity. I would argue the authors should more strongly present their case throughout the text, particularly how this specific approach of selecting a few diverse candidate simulations can fit into the practitioners workflow, or how it relates to uncertainty quantification.

**Minor comments**

- Please change the title. Having read the document I still do not know what "Spatial Calibration" is. It is in fact this term is not once mentioned in this entire manuscript. One suggestion: "Exploration of diverse solutions from imperfect climate model calibration"?
- My own background implies that "model error" = "structural error" = "non-parametric error" = "model discrepancy" here some of these are treated differently, please be clear in defining all of these terms and ensure consistency through the text.
- L19 The literature review missed the growing works of the CliMA group: Here, for example, Idealized GCM's have been calibrated with Bayesian Formalism, using tools from data assimilation and accelerated samplers (https://doi.org/10.1029/2020MS002454, https://doi.org/10.1029/2021MS002735), the calibration approach has shown scalable to higher dimensional parameter spaces in different settings.

- L25 Stating that hand-tuning "has proven remarkably robust", is easily misunderstood by the reader, the authors add a long caveat. Instead why not state something clear, such as "Such approaches remain popular in operational settings"
- L40 For history matching / NROY approaches also cite the studies of (https://doi.org/10.1029/2020MS002217, https://doi.org/10.1029/2020MS002225)?
- L72 This investigation explicitly explores the role of parametric error (whereas structural error often refers to an error incurred due to mis-specification of model structure, i.e. non-parametric error),
- L84 Are there findings from (Peatier et al 2022) about the validity of this PPE also be provided - it seems that results are critically linked to the exploration of this ensemble. My initial concern is that 100 members across 30 dimensions leads to poor exploration without very tight bounds and well chosen points. Perhaps the size of the non-parametric error obtained in this investigation can also shed some light here?
- L86 what is an element, why are they weighted, what is their corresponding area..? Was this sentence misplaced?
- L151 - in other applications it is also seen as a Gaussian Process (following https://doi.org/10.1111/1467-9868.00294)
- L360 "quantitatively", should be "qualitatively". No rigorous scoring was assigned to the amount of trade-offs, and although I am sure the "multiple optima" behavior is true, the authors did actually evaluate any optimal solutions in this investigation, rather solutions achieving below a user-chosen parametric error threshold.
- L519, L553 state that by-hand calibrations are "tractable". This is not true, it may be the case that hand-tuned models can still be performant, but this is not the same as tractability - which additionally implies the procedure of hand-tuning to be easy, and straightforward practice, while the reality is that it is more a "dark art" of the climate modeling community.

**Typos etc.**

- L211 dead link
- L241-2 repeated phrase "high-order modes"
- L245 - "poitn"
- single quotes backwards e.g. L7, L522.,
- Typically latex for "theta star" has a superscript \theta^*.
- Final sentence of L16, was this meant to be here? Seems out of place

---

## Referee Comment (RC2)

**Review of "On the spatial calibration of imperfect climate models" by Peatier et al. (2023)**

**1    Summary**

In this study, the authors propose an approach of calibration that take into account the spatial parameters uncertainty. Indeed, Earth System Models deals with a lot of parameters those values are uncertain and required to be tuned. Generally, a manual approach is done to calibrate a model by minimising a metric. This lead to unique parameters values. However, different sets of parameter values can lead to comparably model skills. Here, the originality of this work is that the objective is to find not one but $n_k$ sets of parameter values that lead to similar model skills. But, instead of selecting randomly $n_k$ sets of parameter values that optimised a metric, they propose these $n_k$ sets must be as various as possible in order to be representative to the whole parametric uncertainty. To that purpose, the authors propose the following method:

1. generate a training data set to built an emulator: for each members (102 members in this study) of a Perturbed Parameters Ensemble, compute the Empirical Orthogonal Function (EOF) coefficients in order to reduce the dimensionality of the problem

2. built a multi-linear emulator, that predicts EOF coefficients directly from parameter values.

3. with the emulator, generate 100.000 emulated EOF coefficients from a 100.000 sampling of parameters values using a Latin Hypercube Sampling

4. from EOF coefficients, compute the simulation error (the metric to optimise) and decompose it as parametric and non-parametric ones.

5. select as optimised candidates, all emulated simulation that have a smaller parametric error than a reference

6. reduce the number of the optimised candidates by selecting only $n_k$ ones: maximise the distance, in the EOF basis, between the $n_k$ candidates in order to get $n_k$ candidates as various as possible in terms of spatial error patterns

This method is applied on the CNRM-CMIP6 model, with a parametric error due to 30 parameters and on 2 cases: a uni-variate case on surface temperature and a multi-variate one. The method is evaluated by comparing the emulated EOF results and real EOF results coming from real simulations using the parameter values obtained from the optimisation process and by comparing spatial pattern diversity obtained from the optimisation process. A deep analysis of the spatial patterns of the different errors (parametric, non-parametric, full) between the $n_k$ optimised candidates is realised.

**2    Major comments**

This study is really interesting and brings a new view of the calibration problem that is in the scope of ESD. The authors described the state-of-the art and contextualise their own method. The new method is described, tested and analysed. This seems to be completed. I really appreciate the fact that author went in details in the method section. Figures show sufficient results to support the interpretation and conclusions. However, I think some points must be addressed before to consider a publication.

- Structure: the paper is quite well-structured, but sometimes, some methodology details are presented in the results section or even in the figure captions. I think, the author must put all the method elements in the method section.

- Method:

    1. To built the emulator, only 102 simulations are considered for a uncertainty due to 30 parameters. I think 102 simulations is clearly not enough to built a robust emulator. Generally, we consider 10 simulations by number of parameters. So here, the authors needs something like 300 simulations, that is very far away from the 102

simulations considered. Even if the emulator shows very good results, the fact that an emulated optimized candidate crash when the author use the real model suggests that there is a lack of data in the training process that leads the emulator to be not enough constraint. I think the author should better evaluate the metamodel in order to demonstrate that in their case 102 simulations is enough.

2. The cross validation method is not clearly presented and the method seems to change between the uni-variate and multi-variate case: the training data is 90% of the data in the uni-variate case, while 80% in the multivariate case. Furthermore, using 90% to train the emulator, means using 92 simulations. As already said, I am not sure it is enough. Authors should be consistent and use the same percentage to define the training and testing datasets.

3. The evaluation of the emulator is also problematic. The emulator is evaluated by analysing spatial patterns or correlation between predicted and true data. A emulator must be evaluated by calculating an error (not a correlation).

4. To define the truncation of the EOF modes, the author used a threshold on the correlation. I think it is not robust. I suggest to define a threshold on the variance.

5. The authors do not explain how they get the number of optimized candidates $n_k$. It seems to be completely subjective.

6. Some inconsistencies have been found. For example, at some points, the authors used $10^6$ emulated simulations and other points, they used 100.000.

7. The reference simulation is not presented.

- Title: This study does not really deal with spatial calibration. It is more a calibration of CNRM-CMIP6 parameters taking into account a spatial uncertainty. I suggest to reformulate the title. For example: "Considering an ensemble of calibration in CNRM-CMIP6 in order to represent spatial model uncertainty"

- Abstract: The abstract do not fully synthesise the paper. I mean, some results (concerning the comparison with perturbed initial condition) are mentioned in the abstract while their appear in few sentences in the paper and are even not mentioned in the conclusions. While, some main results are not present in the abstract.

- Clarity of the paper: Some sentences of the paper must be clarified and some inconsistency have been found (see Minor Comments). Furthermore, as it is not clearly specify if they analyse the results of the emulator or the results of the real model, it is quite confusing.

**3 Minor comments**

**3.1 Abstract**

1. L. 3: "it impossible to find **one** model version"

2. L. 5: Delete the PPE abbreviation (not necessary in the abstract)

3. L. 9: Results on initial conditions perturbation are clearly not a main result of this study. I suggest to do not put it in the abstract, but instead, to present the two applications (uni-variate, multi-variate cases)

**3.2 Introduction**

4. L. 21 - 24: Sentence too long. Please, rewrite it.

5. L. 25: "Robust". I am not sure this a right way to say it.

6. L. 24 - 27: Sentence too long. Please, rewrite it.

7. L. 28: As the ensemble model approach is inserted here, maybe it is better to insert here also the notion of PPE model.

8. L. 35: Put the list of emulators in another sentence.

9. L. 40: Can you clarify the "low dimentionnal output space" ?

10. L. 44: Maybe add a comment on the fact that it depends on subjective advises from physical experts.

11. L. 39: Does the "NROY" abbreviation really need to be defined as it is only used once ?

12. L. 48 - 52: Sentence too long. Please, rewrite it.

13. L. 48: I do not understand the "grid scale". I suppose you want to say "represent spatial performance"

14. L. 48: Clarify that it is for the development of the model.

15. L. 50: Use "grid points" instead of "pixels"

16. L. 54: "addition of additional", please rephrase.

17. L. 53: "structural". What do you mean by this word ?

18. L. 46 - 56: Please, clarify here that you speak about spatial calibration.

19. L. 59: "(PCA; e.g. Hidgon et al., 2008; Sexton et al., 2012; Wilkinson et al. 2010; ...)

20. L. 63: Define the PPE abbreviation here (if nor defined before)

21. L. 72: Delete the definition of PPE abbreviation. Can you clarify the "structural error" ? Do you mean "spatial error" ?

22. L. 71: "it can inform model development". Please reformulate

23. L. 77: Add also a mention to the conclusions in section 5.

**3.3 Methods**

24. L. 79 - 86: Put that paragraph in a subsection named "Used model", for example.

25. L. 79: Here, you refer the model as CNRM-CM6. Later in the paper, you use the name "CNRM-CMIP6", or "CNRM-CM". Please, use the same name everywhere and clarify with ARPEGE-Climat. If you consider that the model used is the full Earth System model, use "CNRM-CMIP6", while if you consider that you run only the atmospheric component, please use "ARPEGE-Climat".

26. L. 79: Specify that CNRM-CM6 is one of the CMIP models.

27. L. 80: To avoid confusion with emulator, I would say "climate model" instead of "climate simulator".

28. L. 80: Definition of PPE abbreviation not necessary

29. L. 81: Can you define AMIP abbreviation ?

30. L. 81: I would say "102 simulations differing by their parameters value".

31. L. 82: That could be interesting to give some example of parameters. Do they all appears in physical parameterization ? A table with the whole list of parameter could be added in appendix.

32. L. 81: Why do you choose a period of 3 years ? And why this period ? Is there a reason ?

33. L. 82: I am not sure it is enough to use 102 simulations for 30 perturbed parameters. Some later point of your paper make me questioning this design of experiment.

34. L. 82: Did you use a classic Latin Hypercube Sampling ? Or did you use a maximin method with a centered design ? In Peatier et al (2022), I understand you used, a maximin method. However as 68 simulations crashed, the design of experiment may not sample some part of the parameters. That could explains why you got a crash with Candidate #1 (in uni-variate case). Please, comment this.

35. L. 83: Please define $\theta$ and define $n$ earlier.

36. L. 84: Can you clarify the "climatological annual means" ? I understand that it is the mean over the period of 3 years and not the mean for each year. Because you need to reduce your calibration to 2 dimensions (parameters and spatial dimension) and thus, delete the temporal dimension.

37. L. 86: "Elements.. area": which elements ? Which area ? What are these "weights" ?

**3.3.1 EOF Analysis**

38. L. 95: What is $\gamma$ ?

39. L. 100: Can you clarify which mean $\mu$ is ? Ensemble and temporal mean over the 3 years period ? Is there a spatial mean ? I suppose no.

40. L. 101: Introduce the "PC" abbreviation here.

41. L. 105: If $r_f$ includes $\mu$, why does $\mu$ appears in the equation ? Is there a mistake ?

42. L. 107: Generally $y$ is used for the result of a function (for example a model). For observation, I would suggest $o$.

**3.3.2 Model error partitioning**

43. L. 116: Replace "partitioning" to "decomposition"

44. L. 119: $\theta_i$ is not a calibration, it is a value of parameters. Delete the word "calibration".

45. L. 122: Can you argue the use of MSE instead of RMSE or bias, or other error definition ? Furthermore, in the rest of the paper, it is mentionned sometimes MSE, some other times RMSE. Is it possible to be consistent and say RMSE if RMSE have been used in the two applications ?

46. L. 123: What is $c_r$ ?

**3.3.3 The discrepancy term**

47. L. 132: Correct the $*$ position: $\theta^*$

48. L. 132: I think the formulation is wrong. $\theta^*$ is not an optimal calibration. It is the optimal set of parameter $\theta$. Am I wrong ?

49. L. 134: Delete "how informative the climate model is about the true climate, and it measures"

50. L. 135: Replace "real" by "measured". Even observations are not perfect. The reality is between models and observations.

51. L. 138: There is also the fact that in numerical models, the Earth system is discretized and not continuous. The fact that is is discretized is also a source of error.

52. L. 138: Generally, to address this issue (parameters not included in calibration process) and in order to include all parameters that must be calibrated, sensitivity analyses are performed (Saltelli et al, 2004). They can estimate the influence of parameters on model outputs and thus give a list a parameter that must be calibrated.

53. L. 139: See comments for L. 132 ("calibration")

54. L. 139: Why do you present the uni-variate example and not the multi-variate one ? The uni-variate one is just a specific case of the multi-variate one.

55. L. 147: This sentence repeats what have been already said in L.134

56. L. 154: Can you clarify the word "operational" ?

57. L. 155 - 156: reformulate the sentence. For example, "in this work, we propose several $\hat{\theta}$ that approximate $\theta^*$".

58. L. 156 - 157: What is $m$ ? I suggest : "to select $m$ optimal model candidates"

**3.3.4 Emulator design and optimization**

59. L. 162: PC abbreviation not defined

60. L. 163: Does the linearity assumption is right ? Can you justify such choice of emulator. Did you compare to other emulators ?

61. L. 168: Can you clarify "comparably performing" ?

62. L. 169: Can you clarify "objective function" ?

63. L. 171: Please specify the characteristic of this new LHS sampling. Is it a centered one ? with maximin space-filling method ?

64. L. 171: instead of "distribution", use "ensemble of "

65. L. 172: What is the reference calibration ? I suppose it is the model using the default value of parameters that have been calibrated manually.

66. L. 172: This lays on the fact that you suppose the reference as calibrated and also to the fact that the error difference between the optimal simulation and the reference is smaller than the emulator error. Did you verify this second point ?

67. L. 177: "pattern error". Do you mean "spatial error" ?

**3.3.5 Selection of diverse candidate calibrations**

68. L. 180: "plausible **optimal** model configurations"

69. L. 185: How do you define $n_k$ ? What is the difference with $k$ defined in L.180 ?

70. L. 185: Why don't you fix $n_k = 5$ for the two applications ? Why don't you select randomly 5 candidates, 100 times, and select the group of 5 candidates amongst the 100 groups which have the highest variance ?

71. L. 185: Maybe clarify that "diverse" means the selected candidates have a high variance

72. L. 186: What is $n_j$ ?

**3.4 First application : surface temperature error**

73. L. 192: At equation (5), you consider a minimization of MSE, while here, you consider RMSE. Please, if you really use the RMSE, use the RMSE in equation (5)

**3.4.1 Assessing meaningful number of degrees of freedom**

74. L. 210: At line 192, you consider RMSE, while here MSE. Please clarify.

75. L. 213 - 215: This is not coherent with lines 113-114. Even if $q = n = 102$, because of the observations, the non-parametric component (residual) is not null, so the reconstruction of the full error $e$ is not perfect. Therefore, it is the best that you can get not the perfect one. (The same for L. 224 and L.252)

76. Figure 1: I suggest this caption: "Full model error $e_{tas}$ and its parametric component $p_{tas}(\theta_i)$ for different truncation length : $q = 5$ (red dots), $q = 20$ (blue dots),$q = 50$ (pink dots), $q = 102$ (orange dots). a: Full error partitioning in parametric and non-parametric components in the PPE members $f(\theta_i)$ ranked from lowest to highest error. b: correlation between the full error $e_{tas}$ and its parametric component $p_{tas}(\theta_i)$ within the PPE."

77. Figure 1: I would recommend to add a) and b) in the upper part of the figure in order to differentiate the left and right panels and to refer to the "Figure 1a" and "Figure 1b", in the text. And I would do the same suggestion for all figures.

78. Figure 1: The non-parametric error is symbolised with dashed line. Logically, the non-parametric error should be the difference between the full error and the parametric error. So, the dashed line should be between the black dots and the colors dots, not between the full error and 0. Am I wrong ?

79. Figure 1: In the rest of the paper, you choose $q = 18$. Why don't you present the result for $q = 18$ in this figure ?

80. L. 217: "A number of feature are notable in Figure 1." Not sure that this sentence is really pertinent.

81. L. 217 - 222: Why do you begin to analyse the right panel ? If you want to analyse it first, put the right panel on the left.

82. L. 219: Instead of using a minus (-) symbol use a colon (:)

83. L. 221: Instead of "variation", I would use "spread" or "variance". (The same for L. 228)

84. L. 221: Maybe add a comment to say, that you will focus in the rest of the paper, on the 5 first modes.

85. L. 225: The percentage is 26% in average over all PPE. However, it can be drastically different between the best and worst PPE. Can you give a range of this ratio ? Is it ok if the major part of the error is non-parametric for the best PPE, particularly if $q = 5$ ? Can you comment it ?

86. L. 227-229: Can you better explain your reasoning ?

**3.4.2    Truncation and parametric emulation**

87. L. 233: Why don't you use the Leave-One-Out Cross validation method ? Maybe add a subsection in the Method section to present the validation method you used for validating the emulator.

88. L. 237: No, you cannot assess the predictive performance of the emulator properly by looking at the correlation between prediction and true data.

89. L. 240: What is $x$ ? The abscissa ?

90. L. 246: The chosen threshold is quite subjective. Maybe it is better to fix a variance at 95% and deduce a threshold in correlation, instead of fixing, arbitrary, a correlation at 0.5.

91. L. 249: Is this the ratios for the emulated PPE ? Is there a link between the left and right panels ? If not, maybe consider 2 different figures for these two plots.

92. L. 249: Help the reader and add reference to the colour of the line : "PPE parametric (blue line) ..."

93. L. 251: Delete the word "example"

94. L. 253: Give the equation $\frac{p(\theta_i)}{e(\theta_i)}$ since line 251.

95. L. 257: Introduce GMMIP

96. L. 267: "This emulator is then optimized...". Not sure it is an optimization of an emulator, but more an optimization of the chosen set of calibrated parameter.

97. L. 271 - 273: Please reformulate

98. Figure 2: I suggest to do not present the method on the caption : "Truncation choice based on parametric emulation and error decomposition. a: Correlation between the emulated and true PCs coefficient of the surface temperature EOF, for the different modes of variability and for : the training set (blue curve), test set (orange curve). Mean is represented by dots and standard deviation by error bars. Averaged correlation over the modes cumulatively is shown by the red curve and the standard deviation by the red shading. b: Ratio as function of the number of modes of variability retained, of the error components compared to : the full error $e_{tas}(\theta_i)$ (in green), the PPE parametric error (dark blue), the PPE non-parametric error (light blue), the reference calibration parametric error (red dotted curve) and the GMMIP parametric error (orange). The lines are the ensemble means and the shadings represent the standard deviations. The black vertical line represents the truncation at q = 18."

**3.4.3 Trade-offs in models candidates**

99. L. 283: at line 171, you precise that you used $10^6$ emulated simulations, while here, it is indicated 100.000. Is there an error ?

100. L. 284: "The five selected parameters calibrations were then used..." -> "The five calibrated set of parameters were then used..."

101. L. 286: The fact that there is a crash by using the real model, suggests a problem in the emulator building, maybe a lack of data in the training phase of the metamodel. Can the author comment this crash ?

102. L. 286: Concerning the other simulation using the 4 calibrated set of parameters, it can be interesting to quantify the emulation error (relative error between emulation and real results). That could be also, another way to verify the ability of the metamodel to emulate ARPEGE-Climat.

103. L. 291: "provides some confidence in [..] the emulation skill". I am not convinced by this conclusion, particularly because the emulated and real simulation do not give similar results. For EOF3 according to EOF1 (1st columns, 3rd line in Figure 3), I agree. But for EOF2 according to EOF1 (1rst column, 2nd line), the emulated set #3 (orange dot) is closer to the real set #4 (blue triangle) than the real simulation #3 (orange triangle)... Maybe a calculation of distance between all pairs of emulated and real simulations can justify your conclusion better. Verify that, for example, the mean (average over all EOF couple) distance between the emulated simulation #1 and the real simulation #1 is smaller than with the other real simulations. Do that calculation for each emulated candidate simulation, and each EOF couple.

104. L. 301: "Figure 3 also allows to see", please rewrite this formulation.

105. L. 301: if the "PC" abbreviation is used and defined, you can replace "Principal Components" by its abbreviation

106. L. 304: Mode 2 is not constraint according to Figure 3. Why the author says "stronger than on the modes 3 to 5" and not "stronger than the other modes" ? Please, clarify.

107. L. 309: "perform equally well on all modes". I understood that is it not true for EOF5 (in line 296) so not "on all modes". Am I wrong ?

108. L. 309-315: Is it possible to calculate the distance between calibrated simulation (real or emulated ones) with observations, in order to justify your comment ?

109. L. 309-315: Clarify that if you consider here the emulated or the real candidates.

110. L. 310: Help the reader and add a "in green" after "candidate 2".

111. L. 312-313: Not sure it is pertinent to comment on mode 5 as observations is outside the emulated ensemble.

112. L. 317: Do the 5 candidates really have a comparable $e(\hat{\theta}_i)$, while candidate #4 performs better in terms of $p$ ? You give the $e$ value in line 336. Maybe, you can discuss this RMSE in section 3.3 instead of 3.4 (in that case, delete the RMSE appearing in Figure 4).

113. Section 3.3: Add a comment to the fact that emulated candidate #1 is generally far away from the other candidates.

114. Figure 3: I suggest this caption: "Correlation between the different standardised PC (obtained from the 102 member PPE EOF) for the 100.000 emulated simulations (light gray), the optimal emulated simulations (dark grey, parametric error lower than the reference), the 5 selected emulated candidates (colored dots), the 4 real ARPEGE-Climat simulations (colored triangles), the reference simulation (star) and the observation (cross).

115. Figure 3: Please, clarify between CNRM-CM, CNRM-CM6-1 and ARPEGE-Climat. If you use one model, please use the same name all along your paper (see comment L.172.).

**3.4.4 Example of temperature discrepancy term partitioning**

116. L. 333: At line 279, you have already mention that you will considered, in the rest of the paper, $q = 18$. It is not necessary to mention it again.

117. L. 334: Specify that you consider the emulated or real calibrated simulation. I understand you consider here the emulated candidates, but why not the real one ?

118. L. 337: Please, add a comment on Candidates #5 which has a smaller $p$ than the reference but a higher $e$.

119. L. 339: Are you comparing the 3th line plots at 1st and 3rd columns ? I see an overestimation of the negative bias or a underestimation of positive bias (not an overestimation of positive bias as written) in the emulated simulation (3rd column compared to 1st). There is also a too large negative bias in Antarctica in the emulated candidate #4. Can you comment it ?

120. L. 346: According to the colorbar, positive bias is in red, and in the Figure, mountain regions are in blue : there is not a positive bias in mountain region. The same for Africa, the bias is positive, not negative as written. Is there a mistake or did I misunderstood the analysis ? Please clarify.

121. L. 348: "vary from a model to another" : the model does not changed (it is still ARPEGE-Climat or CNRM-CMIP6, whatever the name you want to give). Maybe you mean "vary from a parameter set to another" ?

122. L. 348: I see the opposite: strong negative bias on North America and strong positive bias on central Africa.

123. Figure 4: I suggest to change the order of the column : emulated parametric error in column 1, parametric error in column2, non parametric error in column3 and full error in column4. But it is just a suggestion.

**3.5  Second application : multi-variate error**

**3.5.1  Variables, EOF analysis and truncations**

124. Table 1: It is quiet confusing to use radiative data from the 2000-2002 period for a study on the period 1979-1981. Can you explain this choice ?

125. Table 1: Instead of "field", use "observable variables" as in the caption and instead of "citation", use "data product reference" for example.

126. L. 366: "spanning model components": can you clarify ?

127. L. 370: here, the author used the MSE and not the RMSE contrarily to the uni-variate case. Is it possible to justify ?

128. L. 372: Is it really the annual mean or is it the mean over the 3 years period ?

129. L. 377 - 410: As already said for section 3.2, do not consider only the correlation to validate your emulator but also the relative error.

130. L. 380 -385: same comments as in L.246. To stay robust, you should fix the percentage of variance you want, and deduce to that percentage, the number of modes. A correlation of 0.5 does not mean the same for each variable.

131. L. 386: You do not show results for q=4, but for q=5 in Figure 6. So the variance model error is already very-well represented by the first 5 modes, instead of 4.

132. L. 387: I suggest to present Figure 6 before to analyse it.

133. L. 388: "66%" Is it enough ?

134. L. 390: Do you consider here, the emulated simulations or the 102 real ones ? I understand the real one, but precise it.

135. L. 390: As already said, do not consider correlation only for your validation. Consider also the relative errors, here to validate your EOF truncation.

136. L. 390: Why does the non-parametric error is averaged but not the parametric component ? Is it the averaged according to the 5 climatological field ?

137. L. 393: "As expected, the PPE mean non-parametric components decrease as higher EOF modes are retained for the reconstruction but is never equal to 0 (even for a full reconstruction of q = 102). **This is** due to the fact that observations can never be fully captured by their projections into the model EOF basis (Figure 6)"

138. L. 402-404 : Be consistent with model name all along the paper.

139. L. 402: as already said, define your reference calibration.

140. L. 405. Maybe, there is a known bias in surface temperature in the ARPEGE-Climat model, (logically a bias that appears due to the use of the SURFEX surface parameterization). Can you discuss about it ?

141. L. 405: ". This is a simple illustration of a complex tuning problem, and based on the results we obtained in the uni-variate application. **I**t seems likely that comparably performing parameter configurations potentially exist for a multi-variate tuning problem, making different model trade-offs among both climatic fields and EOF modes representations of uni-variate errors (Figure 3)"

142. Figure 5: You do not comment the 2nd column of the figure. If you did not need it to support your discussion, these graphs can be delete of your paper.

143. Figure 6: What is the grey shading ?

144. Figure 6: Instead of "(y axis)" and "(x axis), use "coordinate" and "abscissa".

145. Figure 6: Logically, all your 102 CNRM PPE data must be used for training and testing your emulator. So, all dots must be in green or orange, no one in black. I don't understand your figure.

146. Figure 6: As you should sampling the training and testing datasets 10 times (according to L.236), the trained and testing datasets are not fixed: some point should appear in green for a sampling, but maybe in orange for another sampling. I still not understand the orange and green dots.

147. Figure 6: Please, do not present the training dataset (80%) and testing dataset (20%) in the figure caption.

148. Figure 6: Why do you use a training dataset of 80% for the multi-variate case while 90% for the uni-variate case ? Did you also repeat the training 10 times ?

**3.5.2 Candidates selection in a multi-variate context**

149. L. 412 - 426: This paragraph must be in the Method section.

150. L. 413 : For the subset candidate selection, you maximise the variance of the multi-variate metric. The minimisation of the multi-variate metric is not the process to select a set of optimal candidate.

151. L. 425 : Why $n_k = 4$ and not $n_k = 5$ as in the uni-variate application ?

152. L. 429 - 434: Help the reader by adding in parenthesis to which feature in the Figure it corresponds : "Among the 4 selected candidates **(blue dots)**", "than the reference model **(yellow dashed line**", "PPE mean **(red dot)**", "mean of the 40 CMIP6 models **(green dot)**"

153. L. 433: "CNRM **model** grid before"

154. L. 433: Precise here (not in the figure caption) that observation have been interpolated also.

155. L. 435: Is there a justification to apply the uncertainty (standard deviation) of CMIP6 model to CNRM reference model and not the uncertainty of CNRM model itself (CNRM-PPE) ?

156. L. 436: ",indicating ..." -> "This indicates"

157. L. 440: Maybe present all the data used for this work in the Method section instead of presenting them in the results section.

158. Figure 7: I suggest this caption: "Multi-variate error $e_{tot}$ for the CMIP6 models, the CNRM-CMIP6 PPE members, the 4 optimal CNRM-CMIP6 candidates and the 10 members of CNRM reference model with different initial conditions. Each small dots correspond to a model, the bigger dots correspond to the ensemble means and the dashes are the standard deviations. The orange dashed line at 1.0 represents the CNRM reference model error. The green area indicates the interval of plus or minus one standard deviation of the CMIP6 errors, centered around the CNRM reference model error. "

159. Figure 7: "available CMIP6 model" : Does this mean that all models are not used ?

**3.5.3   Diversity of error patterns among candidates**

160. L. 451: delete "for the selection"

161. L. 451 - 475: I suggest to present the results in the order of candidate number : discuss firstly candidate 1, then candidate 2, ...

162. L. 433: to support your analysis, cite the value of RMSE.

163. L. 451 - 475: Is it possible to attribute theses differences to particular parameters value ?

164. L. 471: "everywhere" -> "on the whole domain"

165. L. 472: Delete "(Figure8)"

166. L. 472: "not the worst of the selection" not coherent with "is the worst performing" at L.471

167. L. 475: Sentence not finished...

168. Figure 8: Candidate 1 has no one green dot : all RMSE are higher than the reference. Please, add a comment on it.

169. Figure 8: Why is it always $\hat{\theta}_1$ in the grey rectangle ? Why the 1 ?

170. Figure 5 - 6 - 8 - 9 - 10: Keep variables appearing in the same order for all figures

**3.5.4   Examples of discrepancy term partitioning**

171. Figures 9 and 10: add $p$ and $u$ in the grey rectangle, as you added $e$ in the Figure 8

172. L. 483: delete some "are"

173. L. 484: I am not sure that it validates the method properly. But, at least, it shows that the objectives are achieved.

174. L. 488 - 505: The analysis is only conducted for SW. Do you have any comments on the other variables ?

175. L. 499 - 502: split this sentences in two different ones.

176. L. 506: please, explain better the link with the effective degrees of freedom

**3.6   Conclusions**

177. L. 511: "perturbed physics" -> "perturbed parameters"

178. L. 512: "diverse" : reformulate

179. L. 513: "a number of" -> "different"

180. L. 513: delete "which we illustrate ... (General Circulation Model)"

181. L. 523: delete "examples"

182. L. 525: "CNRM-CM", please use the same name for the same model

183. L. 525: Really $10^6$ simulations ? Or is it 100.000 as written in the paper ?

184. L. 525: "of the perturbed parameter of the parametric components of the model errors", please reformulate

185. L. 527: Use a more appropriated vocabulary than "diverse"

186. L. 529: "CNRM discrepancy" -> "CNRM **model** discrepancy"

187. L. 547: Add a precision about the fact that candidate performs better than the reference in terms of $e$ , while it has been optimised according to $p$

188. Add some perspectives. For example, it could be interesting to analyse the parameters value between the different candidates, in order to explain which parameters lead to such biases.

**4  Technical corrections**

1. L. 21, 27, 33, 59-60, 91, 152, 188-189, 560: put the citation in chronological order

2. L. 23: correct the parenthesis "(CMIP; Eyring et al, 2016)"

3. L. 40: correct the parenthesis "(such as global mean quantities, Bellprat et al., 2012 ; Williamson et al., 2015)

4. L. 43: correct the parenthesis "(sometimes referred as an "iteratire refocusing" approach; Williamson et al, 2017) ...

5. L. 59: PCA not defined

6. L. 52, 64, 69, 131, 148, 149, 163, 196, 206, 243, in Table 1: correct the parenthesis in the citation (parenthesis around the date and not the name)

7. L. 159: Error in the section referring: "Section 2.4" and "2.5" instead of "Section 2.3" and "2.4"

8. L. 207: Error in the section referring: "Section 3.1" and "3.2" instead of "Section 3.2" and "3.3"

9. L. 211: error in the section referring number

10. L. 232: I think you mean "equation 14", not "Section 14".

11. L. 242: "high-order modes" repeated twice.

12. L. 245: "point"

13. L. 268: "for" repeated twice

14. L. 383 : "e.g.", belonging to which sentence ?

15. L. 420 : "an the" -> "an" or "the" but not the two ones.

16. L. 451: "The Figure" -> "Figure"

17. L. 483: "on the other ends" -> "on the other hands"

18. L. 484: remove the space before ":"

19. L. 490: (Figure 10) -> (Figure 8)

20. L. 525: LH abbreviation not defined

21. L. 533-536: Did you want to do a list as in lines 514-521 ?

22. L. 552: opimization -> optimization

23. Figure 7: "dasehd" -> "dashed"

24. Figure 7: "arounr" -> "around"

---

## Author Comment (AC1)

**Response to Reviewer #1**

First of all, we would like to thank the reviewers for the comments on our manuscript. We appreciate all valuable suggestions, which helped us improve the quality of the paper. Following the reviewers' suggestions, we have improved the method to select optimal and diverse candidates. It is now based on a clustering analysis, and sensitivity tests regarding the number of clusters to consider is presented in an appendix of the paper. Given this new method, we found the need to consider around 10 clusters to explore the diversity of error patterns in the ensemble. Therefore, 10 candidate calibrations have been tested in the climate model and are presented in the paper. Most of the figures related to the candidate selection and their associated comments will be updated based on these modifications. Another appendix will be added regarding the choice and evaluation of the statistical models, and the paper will be slightly restructured to present all the methodological details in the Method section. Note also that, as requested by the reviewers, the title will be modified to : "Exploration of diverse solutions for the calibration of imperfect climate models".

Our responses to the reviewer #1 comments are described below. In the following, the comments from reviewers are in black, our answers are in blue.

**Major comments**

Method deficiencies (i) the emulator: The choice and use of emulator are not discussed, but in several places in the results it is indicated that there is room for improvement here. E.g. in Figure 3 - often the emulated values (that determine also the concept of "diversity" above) did not lead to close or hugely diverse simulations. I came to this conclusion as (1) the dots and triangles are often far from each other, and (2) the dots are typically more spread than the triangles, (3) one dot lead to a simulation crash for its corresponding triangle. Likewise in Figure 7: 4 diverse candidates were selected with emulated $p < 1$, yet all in practice most were $> 1$. Though this was noted in the document it was not attributed to the emulation stage being unable to well-represent the underlying model. Particularly in the results, the shortcomings due to the choice of emulator should be attributed clearly, and additional discussion added to the conclusions in terms of how to address this in future investigation.

As a better discussion about the choice and performance of the emulators was also requested by the Reviewer #2, we have decided to add an appendix showing the performance of our statistical model to emulate the parametric components of the individual and total model errors. We also compare their performances with other statistical models : a Random Forest and a LASSO regression.

First of all, the ensemble size of the PPE is very limited (102 simulations) and our capacity to train emulators is fundamentally limited by the sample size available. The emulators used in this study are Multi Linear Regressions (MLR) taking the model parameters as input and predicting the Principal Components (PC) used to reconstruct the 3D variables and the parametric model errors when comparing with observations.

In 10 random selections of out-of-sample test sets, we obtain an average correlation of 0.7 between the predictions and the true values of total error (Figure 1 (c)), with a RMSE between predictions and true values representing 8% of the total parametric error (Figure 1 (f)), which is sufficient to validate the use of this model for our study. However, results suggest that there is room for improvement, especially in the prediction of the LW errors, and that another model could improve the predictions, as it is the case of the Random Forest model. The error bars associated with the prediction of the total error suggests that the MLR performance is sensitive to the test set selected and that the model will perform unevenly across the parameter space. Thanks to variable selection and regularization, the Lasso model seems a bit less sensitive to the test set selection for the prediction of total error, but the prediction of LW error is still a limitation.

[Figure]

**Fig 1.** Correlations and RMSE (in % compared to the true values) between emulated and true parametric components of the errors within a test set representing 10% of the dataset. The evaluation is repeated 10 times with random sampling of training and test sets and the mean and standard deviation among these 10 evaluations are represented by the bars and the dashed lines, respectively. Performances are shown for (a), (d) a Random Forest, (b), (e) a LASSO regression and (c), (f) the Multi Linear Regression used in this analysis and. The EOF truncation lengths used to compute the parametric error are presented in Figure 2 and 5 of the paper.

In conclusion, it seems that using a non-linear emulator could improve certain aspects of the predictions, though enhancing the size of the ensemble would be a necessary prerequisite to try to improve our statistical predictions. Figure 1 will be added in an appendix and we will add sentences in the main paper clearly stating the limitations of the emulators. Gaussian Processes are statistical models often used in PPE analysis and even though we did not test them in this study, we will add sentences in the conclusion suggesting it as a potential perspective to this work.

Indeed, Figure 3 of the paper was highlighting a loss of diversity in the actual runs compared to the statistical predictions, but this might be linked to our previous method of candidate selection. The clustering analysis seems to show a bit less diversity in the emulated candidates, but a gain in diversity in the climate runs (Figure 2). There is still a prediction error (the dots and triangles can be far from each other in Figure 2 and we expect around 8% of prediction error, as presented in Figure 1), but the use of clustering analysis and the sampling of 10 instead of 5 candidates allow a better exploration of the optimal sub-set of emulations.

[Figure]

**Fig 2**. Standardised principal components associated with the 5 first modes of the surface temperature EOF basis computed on the 102 members of the PPE. The Figure presents the projections of the 100 000 emulated members (light gray), the 'optimal' emulated members (with a parametric error lower than the reference CNRM-CM6-1, in dark grey), the 10 emulated candidates (colored dots) and the 10 candidates CNRM-CM (colored triangles). This Figure will replace Figure 3 in the main paper.

Finally, we agree that the fact that the candidates were selected with emulated parametric error p<1 but shows, in practice, higher parametric error should be attributed to the limitations of the emulators and we will add a sentence clearly stating that in the main paper. Table 1 is showing the values of the parametric errors for the

new candidates selected with the clustering analysis : only 2 of them have lower aggregated error than the reference.

| Candidate 1 | 1.68 | Candidate 6 | 1.03 |
|---|---|---|---|
| Candidate 2 | 1.0 | Candidate 7 | 1.09 |
| Candidate 3 | 0.88 | Candidate 8 | 1.7 |
| Candidate 4 | 1.05 | Candidate 9 | 1.22 |
| Candidate 5 | 0.91 | | |

**Table 2**. Parametric component p of the error in RMSE for the 9 candidates of the multivariate application. A candidate has a lower error than the reference when the parametric error is lower than 1. Note that 10 candidates were sampled originally, but one of them crashed.

Method deficiencies (ii) exploring diversity: Perhaps the weakest part of the investigation involved the exploration of "diversity of solutions". Though I agree that the approach removes solutions that have similar parametric components, it is mentioned L284 that the authors "maximize" the diversity in solutions, or L177 "a selection ... as diverse as possible", yet their approach is not a seemingly well defined optimization procedure. In fact the approach the authors use appears to be non-standard for such exploration, likewise with the termination of "5" final candidates is completely unmotivated. Finally the sensitivity to the choice of the threshold was also not explored. Typically for sourcing representative simulations a practitioner might use a clustering algorithms (k-means/medians/mediods etc.) over the data, and draw representative samples from the cluster centers to illustrate diversity. Such approaches are scalable and robust, and one can quantitatively assess sensitivity to e.g. the number of clusters the data appears, thresholding etc. The random removal methodology presented here appears untested. For example, if the authors ran the method over 100 trials, would they expect to find the same nodes each time, or if not, would they expect the selections of 5 display similar diversity, how would they quantitatively measure this? How would they compare the performance of terminating at different numbers of solutions. Some references and/or computational validation experiments for the approach taken are needed, otherwise, again some clear admission in the methods and final discussion that this is a proof of concept, and suggest other approaches that readers can try in future to explore diversity.

From a naive inspection of the samples in Figure 3, it does not seem out of the question that there are more representative samples one could find. I also think the use of emphatic words such as "large" diversity (L452 and elsewhere), "the key" trade-offs (L507 and elsewhere) should really be removed when one is not able to really quantify how representative these differences are, or how many trade-offs are observed due to the selection procedure.

We agree that the selection of diverse candidates was done through a non-standard approach and that the sensitivity of the results to the number of candidates was not tested in the submitted version of the paper. Following reviewers comments, we have changed our methodology and replaced the previous algorithm by a clustering method,

applied to the Principal Components of the different variables, normalized by the principal components of the reference model. We have chosen the Euclidean distance as a measure of similarity in the clustering analysis and we have selected the centroids as our set of "diverse candidates". As a result, we rewrote the paper Section "**2.5 Selection of diverse candidate calibrations**" to present the clustering analysis.

Moreover, as stated by the reviewer, the clustering analyses are sensitive to the choice of cluster number $k$, which depends on the dataset to be classified. Figure 3 presents a sensitivity test of the k-means analysis to the number of clusters for the univariate and multivariate application. The inertia is defined as the sum of the squared distances between each data point and the centroid within a same cluster. The Elbow method consists in finding the inflexion point in the k-means performance curve (Figure 3 (a), (c)), where the decrease in inertia begins, to find the good trade-off for the number of clusters. Another criteria we looked at is the Dunn index (Figure 3 (b), (d)) : the ratio between the minimal inter-cluster distances and the maximal intra-cluster distances. A higher Dunn index represents a higher distance in between the centroids (clusters are far away from each other) and a lower distance in between the data points and the centroid of a same cluster (clusters are compact).

Even though it is not as pronounced in the multivariate application, the inertia sensitivity test suggests that we should choose a value of $k$ in between 8 and 18 to be in the Elbow of the curve. Then, it appears that we should not take a value of $k$ too high, as the Dunn index tends to decrease for both applications for a number of clusters higher than 10. Based on these two criteria, we have decided to keep 10 clusters for the analysis and we carried out the analyses for 10 optimal and diverse candidates. Figure 3 will be part of an appendix detailing the choice of $k$.

[Figure]

**Fig 3**. Sensitivity test of the clustering analyses for the uni-variate (first row) and multi-variate (second row) applications. The inertia criteria ((a), (b)) and the Dunn index ((b), (d)) are shown depending on the number of clusters (x axes). The green shaded areas present the acceptable number of clusters following the Elbow method applied to the inertia. The green dashed line shows the number of clusters retained for our analyses : k=10 in both applications. The grey thin lines show the ± 1σ range for a repetition of 10 evaluations with random selection of initial centroids (the range is negligible for the inertia).

Optimality and near-optimality
As a reader, this is the walkthrough of optimality in this manuscript :
- An optimal solution is defined to be an exact quantity f(theta*) eqn(9).
- This is then redefined in L158 where optimal is an input that minimized e_j.
- Then in L168 the optimal solution is considered to be the input that minimizes an emulated objective (maybe this should be called e_{em,j}).
- In L172 The introduction of a black-box "reference calibration" CNRM-CM (6?) is introduced as a threshold for optimal candidates (this is the first time this is mentioned in the document and it is unreferenced), it is used to rule out all candidates with greater error. It is not clear if this is calculated with real or emulated coefficients. Is the intention to have a "typical" level of accuracy, arising from current "tuning procedures"?
- In L520 there is mention of near-optimal configurations
- In L528 optimality and an optimal space are then unified with near optimality?

I request that the authors solidify these notions and be consistent throughout. I also request that the authors use concrete notation for where they are using emulated vs actual scores. Finally, the threshold should be presented in a more clear fashion and explanation as to why this choice should define optimality.

Agreed - we will better define the notion of "optimal" in the paper by using the term "near-optimal" to refer the vectors of parameter values $\hat{\theta}_i$ associated with emulated parametric errors lower than the reference model configuration CNRM-CM6-1.

The reference model configuration CNRM-CM6-1 results from a tuning by the developers for the CMIP6 exercise. This tuning was done following the historical common practices for tuning a climate model (Hourdin et al. (2017), Schmidt et al. (2017)) and has been validated by model developpers. This reference model will be better defined and cited in the paper : "The reference model will be the model CNRM-CM6-1, tuned by the model developers for the CMIP6 exercise (Roehrig et al. (2020)). This reference model has been validated by the experts and can serve as a threshold to define whether a model calibration is near-optimal."

We will also note the emulated scores $p_{em,j}(\theta_i)$ and the actual scores $p_j(\theta_i)$ in order to differentiate whether we discuss statistical predictions or climate model outputs.

**References :**

Hourdin, F., Mauritsen, T., Gettelman, A., Golaz, J. C., Balaji, V., Duan, Q., ... & Williamson, D. (2017). The art and science of climate model tuning. *Bulletin of the American Meteorological Society*, *98*(3), 589-602.

Schmidt, G. A., Bader, D., Donner, L. J., Elsaesser, G. S., Golaz, J. C., Hannay, C., ... & Saha, S. (2017). Practice and philosophy of climate model tuning across six US modeling centers. *Geoscientific Model Development*, *10*(9), 3207-3223.

Why explore the diversity of solutions?: Given that the investigation is centered around exploring the diversity of these solutions of low parametric error, It was mentioned only in the final sentence as to why one wishes to do this beyond scientific curiosity. I would argue the authors should more strongly present their case throughout the text, particularly how this specific approach of selecting a few diverse candidate simulations can fit into the practitioners workflow, or how it relates to uncertainty quantification.

Agreed - we will add a paragraph in the introduction better motivating this exploration of diverse solutions.

**Minor comments**

Please change the title. Having read the document I still do not know what "Spatial Calibration" is. It is in fact this term is not once mentioned in this entire manuscript. One suggestion: "Exploration of diverse solutions from imperfect climate model calibration"?

We agree that the title of the paper was not the most appropriate and we changed it to : "Exploration of diverse solutions for the calibration of imperfect climate models".

My own background implies that "model error" = "structural error" = "non-parametric error" = "model discrepancy" here some of these are treated differently, please be clear in defining all of these terms and ensure consistency through the text.

Agreed - we will make sure all the terms are defined and consistent.

L19 The literature review missed the growing works of the CliMA group: Here, for example, Idealized GCM's have been calibrated with Bayesian Formalism, using tools from data assimilation and accelerated samplers (https://doi.org/10.1029/2020MS002454, https://doi.org/10.1029/2021MS002735), the calibration approach has shown scalable to higher dimensional parameter spaces in different settings.

Agreed - the above references will be added in the literature review

L25 Stating that hand-tuning "has proven remarkably robust", is easily misunderstood by the reader, the authors add a long caveat. Instead why not state something clear, such as "Such approaches remain popular in operational settings"

Corrected

L40 For history matching / NROY approaches also cite the studies of (https://doi.org/10.1029/2020MS002217, https://doi.org/10.1029/2020MS002225)?

Agreed - the above references will be added in the literature review

L72 This investigation explicitly explores the role of parametric error (whereas structural error often refers to an error incurred due to mis-specification of model structure, i.e. non-parametric error),

We replaced 'structural error' by 'model error'

L84 Are there findings from (Peatier et al 2022) about the validity of this PPE also be provided - it seems that results are critically linked to the exploration of this ensemble.

My initial concern is that 100 members across 30 dimensions leads to poor exploration without very tight bounds and well chosen points. Perhaps the size of the non-parametric error obtained in this investigation can also shed some light here?

We agree that the small size of the ensemble is a limiting factor in this analysis and that a larger PPE will improve parameter space observation and might affect the amplitude of the non-parametric component ... We will add a few words in the conclusion stating this caveat.

L86 what is an element, why are they weighted, what is their corresponding area..? Was this sentence misplaced?

Indeed, this was misplaced

L151 - in other applications it is also seen as a Gaussian Process (following https://doi.org/10.1111/1467-9868.00294)

Agreed
L360 "quantitatively", should be "qualitatively". No rigorous scoring was assigned to the amount of trade-offs, and although I am sure the "multiple optima" behavior is true, the authors did actually evaluate any optimal solutions in this investigation, rather solutions achieving below a user-chosen parametric error threshold.

Agreed

L519, L553 state that by-hand calibrations are "tractable". This is not true, it may be the case that hand-tuned models can still be performant, but this is not the same as tractability - which additionally implies the procedure of hand-tuning to be easy, and straightforward practice, while the reality is that it is more a "dark art" of the climate modeling community.

We will remove the word 'tractable' and change the sentence to : "Reference model version shows one of the lowest integrated performance metric and historical common practices for parameter tuning is more robust than often assumed"

**Typos etc.**

L211 dead link - Ok
L241-2 repeated phrase "high-order modes" - Ok
L245 - "point" - Ok
single quotes backwards e.g. L7, L522., - Ok
Typically latex for "theta star" has a superscript \theta^*. - Ok

Final sentence of L16, was this meant to be here? Seems out of place - We have changed this whole paragraph

---

## Author Comment (AC2)

**Response to Reviewer #2**

First of all, we would like to thank the reviewers for the comments on our manuscript. We appreciate all valuable suggestions, which helped us improve the quality of the paper. Following the reviewers' suggestions, we have improved the method to select optimal and diverse candidates. It is now based on a clustering analysis, and sensitivity tests regarding the number of clusters to consider is presented in an appendix of the paper. Given this new method, we found the need to consider around 10 clusters to explore the diversity of error patterns in the ensemble. Therefore, 10 candidate calibrations have been tested in the climate model and are presented in the paper. Most of the figures related to the candidate selection and their associated comments will be updated based on these modifications. Another appendix will be added regarding the choice and evaluation of the statistical models, and the paper will be slightly restructured to present all the methodological details in the Method section. Note also that, as requested by the reviewers, the title will be modified to : "Exploration of diverse solutions for the calibration of imperfect climate models".

Our responses to the reviewer #2 comments are described below. In the following, the comments from reviewers are in black, our answers are in blue.

**Major comments**

This study is really interesting and brings a new view of the calibration problem that is in the scope of ESD. The authors described the state-of-the art and contextualise their own method. The new method is described, tested and analysed. This seems to be completed. I really appreciate the fact that author went in details in the method section. Figures show sufficient results to support the interpretation and conclusions. However, I think some points must be addressed before to consider a publication.

Structure : the paper is quite well-structured, but sometimes, some methodology details are presented in the results section or even in the figure captions. I think, the author must put all the method elements in the method section.

Agreed - we will restructure the paper to put all the methodology elements in the method section. The definition of the near-optimal subset of emulations and the selection of diverse candidates will be presented for both the univariate and multivariate applications in the method section, instead of later in the paper for the multivariate case.

Method

1. To built the emulator, only 102 simulations are considered for a uncertainty due to 30 parameters. I think 102 simulations is clearly not enough to built a robust emulator. Generally, we consider 10 simulations by number of parameters. So here, the authors needs something like 300 simulations, that is very far away from the 102 simulations considered. Even if the emulator shows very good

results, the fact that an emulated optimized candidate crash when the author use the real model suggests that there is a lack of data in the training process that leads the emulator to be not enough constraint. I think the author should better evaluate the metamodel in order to demonstrate that in their case 102 simulations is enough.

2. The cross validation method is not clearly presented and the method seems to change between the univariate and multivariate case: the training data is 90% of the data in the univariate case, while 80% in the multivariate case. Furthermore, using 90% to train the emulator, means using 92 simulations. As already said, I am not sure it is enough. Authors should be consistent and use the same percentage to define the training and testing datasets.

3. The evaluation of the emulator is also problematic. The emulator is evaluated by analysing spatial patterns or correlation between predicted and true data. A emulator must be evaluated by calculating an error (not a correlation).

As a better discussion about the choice and performance of the emulators was also also requested by the Reviewer #1, we have decided to add an appendix showing the performance of our statistical model to emulate the parametric component of the individual and total model errors. We also compare their performances to other statistical models : a Random Forest and a LASSO regression. Finally, as suggested by reviewer #2 , we have added an evaluation of the emulators based on the root-mean square error (RMSE) between the predictions and the true values of the parametric components of the model errors, in addition to the evaluation based on the correlation.

First of all, the ensemble size of the PPE is very limited (102 simulations) and our capacity to train emulators is fundamentally limited by the sample size available. The emulators used in this study are Multi Linear Regressions (MLR) taking the model parameters as input and predicting the Principal Components (PC) used to reconstruct the 3D variables and the parametric model errors when comparing with observations. In 10 random selections of out-of-sample test sets, we obtain an average correlation of 0.7 between the predictions and the true values of total error (Figure 1 (c)), with a RMSE between predictions and true values representing 8% of the total parametric error (Figure 1 (f)), which is sufficient to validate the use of this model for our study.

[Figure]

**Fig 1.** Correlations and RMSE (in % compared to the true values) between emulated and true parametric components of the errors within a test set representing 10% of the dataset. The evaluation is repeated 10 times with random sampling of training and test sets and the mean and standard deviation among these 10 evaluations are represented by the bars and the dashed lines, respectively. Performances are shown for (a), (d) a Random Forest, (b), (e) a LASSO regression and (c), (f) the Multi Linear Regression used in this analysis and. The EOF truncation lengths used to compute the parametric error are presented in Figure 2 and 5 of the paper.

These results suggest that there is room for improvement, especially in the prediction of the LW error, and that another model could potentially improve the predictions, as it is the case of the Random Forest model. The error bars associated with the prediction of the total error suggests that the MLR performance is sensitive to the test set selected and that the model will perform unevenly across the parameter space. Thanks to variable selection and regularization, the Lasso model seems a bit less sensitive to the test set selection for the prediction of total error, but the prediction of LW error is still a limitation.

In conclusion, it seems that using a non-linear emulator could improve certain aspects of the predictions, though enhancing the size of the ensemble would be a necessary prerequisite to try to improve our statistical predictions. Figure 1 will be added in an appendix and we will add sentences in the main paper clearly stating the limitations of the emulators. Gaussian Processes are statistical models often used in PPE analysis and even though we did not test them in this study, we will add sentences in the conclusion suggesting it as a potential perspective to this work.

4. To define the truncation of the EOF modes, the author used a threshold on the correlation. I think it is notrobust. I suggest to define a threshold on the variance.

The idea of a threshold based on the correlation between true and predicted values is to rule out the EOF modes poorly emulated by our statistical models. We argue that these higher modes will introduce noise in the analysis, and are not strongly affected by the change in parameter values anyway, as they explain a very small part of the PPE variance. However, we also want to make sure that we consider enough EOF modes to represent a large fraction of the PPE variance. There was already an implicit threshold, since all of the truncations chosen were explaining more than 85% of the PPE variance. In order to make this point clearer in the paper, we added another y axis on Figures 2 and 5 to show the % of explained variance depending on the truncation, and we draw the threshold of 85% for each variable to illustrate that our chosen truncation is explaining more than this threshold. An example of the modifications is shown here for Figure 2, which is the Figure 2 of the paper (Figure 5 will also be updated accordingly).

[Figure]

**Fig 2.** Truncation choice based on parametric emulation and error decomposition. The left plot (a) shows the correlation between the emulated and true PCs of the surface temperature EOF, for the different modes of variability. The correlation is shown within the training set (blue curve) and the test set (orange curve). The evaluation is repeated 10 times with random sampling of training and test sets, the mean and the standard deviation among these 10 evaluations are represented by the dots and the error bars, respectively. The red curve and shading is the mean correlation averaged over the modes cumulatively. The solid green curve represents the percentage of variance explained when retaining up to x modes of the EOF. The dashed green horizontal line shows a threshold of 85% of explained variance and the solid vertical line is the truncation length needed to satisfy this threshold. The right plot (b) shows the ratio of the error components compared to the full error $e_{tas}(\theta_i)$ (in green) as a function of the number of modes of variability retained. The lines are the ensemble means and the shadings represent the standard deviations. The plot shows the ratios of the PPE parametric error (dark blue), the PPE non-parametric error (light blue), the reference calibration parametric error $p_{tas}(\theta_0)$ (red dotted curve) and the GMMIP parametric error (orange). An example of truncation at q = 18 is represented on both plots by the black vertical line.

5. The authors do not explain how they get the number of optimized candidates $n_k$. It seems to be completely subjective.

We agree that the number of optimized candidates was completely subjective, which was a design choice. Following reviewers comments, we have changed our methodology and replaced the previous algorithm by a clustering method, applied to the Principal Components of the different fields, normalized by the Principal Components of the reference model. WWe have chosen the Euclidean distance as a measure of similarity in the clustering analysis and we have selected the centroids as our set of "diverse candidates". As a result, we rewrote the paper Section "**2.5 Selection of diverse candidate calibrations**" to present the clustering analysis.

Moreover, the clustering analyses are sensitive to the choice of cluster numbers $k$, which depends on the dataset to be classified. Figure 3 presents a sensitivity test of the k-means analysis to the number of clusters for the univariate and multivariate application. The inertia is defined as the sum of the squared distances between each data point and the centroid within a same cluster. The Elbow method consists in finding the inflexion point in the k-means performance curve (Figure 3 (a), (c)), where the decrease in inertia begins, to find the good trade-off for the number of clusters. Another criteria we looked at is the Dunn index (Figure 3 (b), (d)) : the ratio between the minimal inter-cluster distances and the maximal intra-cluster distances. A higher Dunn index represents a higher distance in between the centroids (clusters are far away from each other) and a lower distance in between the data points and the centroid of a same cluster (clusters are compact).

Even though it is not as pronounced in the multivariate application, the inertia sensitivity test suggests that we should choose a value of $k$ in between 8 and 18 to be in the Elbow of the curve. Then, it appears that we should not take a value of $k$ too high, as the Dunn index tends to decrease for both applications for a number of clusters higher than 10. Based on these two criteria, we have decided to keep 10 clusters for the analysis and we carried out the analyses for 10 optimal and diverse candidates. Figure 3 will be part of an appendix detailing the choice of $k$.

[Figure]

**Fig 3**. Sensitivity test of the clustering analyses for the uni-variate (first row) and multi-variate (second row) applications. The inertia criteria ((a), (b)) and the Dunn indexes ((b), (d)) are shown depending on the number of clusters (x axes). The green shaded areas present the acceptable number of clusters following the Elbow method applied to the inertia. The green dashed line shows the number of clusters retained for our analyses : k=10 in both applications. The grey thin lines show the $\pm$ 1σ range for a repetition of 10 evaluations with random selection of initial centroids (the range is negligible for the inertia).

6. Some inconsistencies have been found. For example, at some points, the authors used $10^6$ emulated simulations and other points, they used 100.000.

Yes, this was a mistake in the text, we are using 100 000 everywhere, the mistake will be corrected.

7. The reference simulation is not presented.

The reference model configuration CNRM-CM6-1 results from a tuning by the developers for the CMIP6 exercise. This tuning was done following the historical common practices for tuning a climate model (Hourdin et al. (2017), Schmidt et al. (2017)) and has been validated by model developpers. This reference model will be better defined and cited in the paper : "The reference model will be the model CNRM-CM6-1, tuned by the model developers for the CMIP6 exercise (Roehrig et al. (2020)). This reference model has been validated by the experts and can serve as a threshold to define whether a model calibration is near-optimal."

**References :**

Hourdin, F., Mauritsen, T., Gettelman, A., Golaz, J. C., Balaji, V., Duan, Q., ... & Williamson, D. (2017). The art and science of climate model tuning. *Bulletin of the American Meteorological Society*, *98*(3), 589-602.

Schmidt, G. A., Bader, D., Donner, L. J., Elsaesser, G. S., Golaz, J. C., Hannay, C., ... & Saha, S. (2017). Practice and philosophy of climate model tuning across six US modeling centers. *Geoscientific Model Development*, *10*(9), 3207-3223.

Title: This study does not really deal with spatial calibration. It is more a calibration of CNRM-CMIP6 parameters taking into account a spatial uncertainty. I suggest to reformulate the title. For example: "Considering an ensemble of calibration in CNRM-CMIP6 in order to represent spatial model uncertainty"

We agreed and changed the title for : "Exploration of diverse solutions for the calibration of imperfect climate models".

Abstract: The abstract do not fully synthesise the paper. I mean, some results (concerning the comparison with perturbed initial condition) are mentioned in the abstract while their appear in few sentences in the paper and are even not mentioned in the conclusions. While, some main results are not present in the abstract.

Agreed - the abstract will be updated considering the new results and the sentence about the comparison with the initial condition ensemble will be added in the conclusion as well.

**Abstract :** "The calibration of Earth System Model parameters is subject to both data, time and computational constraints. The high dimensionality of this calibration problem, combined with errors which arise from model structural assumptions, makes it impossible to find model versions fully consistent with historical observations. Therefore, the potential for multiple plausible configurations presenting different tradeoffs between skills in various variables and spatial regions remains usually untested. In this study, we lay out a formalism for making different assumptions about how ensemble variability in a Perturbed Physics Ensemble relates to model error, proposing an empirical but practical solution for finding diverse near-optimal solutions. We argue that the effective degrees of freedom in model performance response to parameter input (the 'parametric component') is, in fact, relatively small, illustrating why manual calibration is often able to find near-optimal solutions. The results explore the potential for comparably performing parameter configurations making different trade-offs in model errors. These model candidates can inform model development and could potentially lead to significantly different future climate evolution."

Clarity of the paper: Some sentences of the paper must be clarified and some inconsistency have been found (see Minor Comments). Furthermore, as it is not clearly specify if they analyse the results of the emulator or the results of the real model, it is quite confusing.

We will answer each Minor Comment in the next section. Moreover, we will note the emulated scores $p_{em,j}(\theta_i)$ and the actual scores $p_j(\theta_i)$ in the paper, in order to differentiate whether we discuss statistical predictions of climate model outputs.

**Minor comments**

Abstract

1. L. 3: "it impossible to find one model version"

Ok

2. L. 5: Delete the PPE abbreviation (not necessary in the abstract)

Agreed

3. L. 9: Results on initial conditions perturbation are clearly not a main result of this study. I suggest to do not putit in the abstract, but instead, to present the two applications (uni-variate, multi-variate cases)

Agreed

Introduction
1. L. 21 - 24: Sentence too long. Please, rewrite it.

Agreed - new sentence : "However, more complex models such as GCM present a number of difficulties for objective calibration which have resulted in a *status quo* in which manual calibration remains the default approach (Mauritsen et al., 2012; Hourdin et al., 2017)."

2. L. 25: "Robust". I am not sure this a right way to say it.
3. L. 24 - 27: Sentence too long. Please, rewrite it.

Agreed - new sentences : "Such approaches have not yet been operationally replaced by objective calibration approaches, but leave large intractable uncertainties. In particular, the potential existence of comparably performing alternative configurations with significantly different future climate evolution (Hourdin et al., 2023; Ho et al., 2012)."

4. L. 28: As the ensemble model approach is inserted here, maybe it is better to insert here also the notion of PPEmodel.
5. L. 35: Put the list of emulators in another sentence.

We inserted a sentence line 34-35 to introduce the notion of PPE and rewrite the list of emulators :

"Approaches to date with GCMs have mainly relied on Perturbed Parameter Ensembles (PPE) of simulations, allowing an initial stochastic sample of the parametric response of the model. The construction of meta-models is then needed to emulate this parametric response and enhance the number of samples. The meta-models can be quadratic (Neelin et al., 2010), logistic regression (Bellprat et al., 2012), Gaussian process emulators (Salter and Williamson, 2016) or neural networks (Sanderson et al., 2008)."

6. L. 40: Can you clarify the "low dimentionnal output space" ?

It is already written L40 "low dimentionnal output space (such as global mean)".

Indeed, the emulators can not predict the high dimensionality of a model output, with the numerous grid points and time steps. The example of global mean given L.40 is rarely used in PPE analysis and the EOF (or PCA analysis) presented L.57 is the most common way to reduce the output dimensionality for such applications.

L.57 : "In order to reduce the complexity of the emulation problem, and to preserve the covariance structure of the model output, it is common to reduce the dimensionality of

the output through Principal Component Analysis (PCA) (Higdon et al., 2008; 60 Sexton et al., 2012; Wilkinson, 2010)."

7. L. 44: Maybe add a comment on the fact that it depends on subjective advises from physical experts.

Agreed - L.44 : "Emulators can be improved in promising sub-regions of the parameter space by running a new PPE in a reduced parameter space to increase the ensemble density (sometimes referred as an "iterative refocussing" approach) (Williamson et al., 2017). However, the choice of which region to initially focus on depends on advice from model developers and is itself subject to error in emulation."

8. L. 39: Does the "NROY" abbreviation really need to be defined as it is only used once ?

No, agreed

9. L. 48 - 52: Sentence too long. Please, rewrite it.
10. L. 48: I do not understand the "grid scale". I suppose you want to say "represent spatial performance"
11. L. 48: Clarify that it is for the development of the model.
12. L. 50: Use "grid points" instead of "pixels"
13. L. 46 - 56: Please, clarify here that you speak about spatial calibration.

New sentences L.46 - 52 : "Climate models produce high dimensional output across space, time and variable dimensions. Performance is often addressed by integrated output spanning these dimensions (Gleckler et al., 2008; Sanderson et al., 2017) and so calibration techniques must be able to represent spatial performance in order to be useful to development. In a low dimensional space defined by global mean quantities, it is possible to find one model version which is consistent with observations (Williamson et al., 2015), but this is not true when considering the high dimensionality of climate model outputs. When considering an assessment of model error integrated over a large number of grid points and variables, structural trade-offs may arise between model outputs which cannot be simultaneously optimized by adjusting model parameters."

14. L. 54: "addition of additional", please rephrase.

Agreed - L.54 : "In another case, structural errors in an atmospheric model were found to increase significantly with the addition of variables to a spatial metric (Sanderson et al., 2008)."

15. L. 53: "structural". What do you mean by this word ?

We refer here to the impossibility to "tune down" all the variables simultaneously, and to the fact that the discrepancy term associated with an optimal parameterization increases when adding variables to the multi-variate score considered.

Citation from (Sanderson et al. (2008)) : "Requiring models to match all observations simultaneously proved a more difficult task for all of the ensembles. The ANN simulated ensemble suggested that model parameters could at best be tuned to a compromise

configuration with a finite error from the observations. This "best model discrepancy" was found to increase with the inclusion of increasing numbers of separate observations, and was not itself a strong function of S."

16. L. 59: "(PCA; e.g. Hidgon et al., 2008; Sexton et al., 2012; Wilkinson et al. 2010; ...)

Agreed

17. L. 63: Define the PPE abbreviation here (if nor defined before)

It is now defined L.34

18. L. 72: Delete the definition of PPE abbreviation. Can you clarify the "structural error" ? Do you mean "spatial error" ?

No, we mean the part of the error that does not relate to the model tuning. This is the part of error arising because of processes not or poorly represented in the model. But we do not need to use this world here and we are interested in model error in general.

New sentence : "In this study, we lay out an alternative formalism which makes different assumptions about how ensemble variability in a Perturbed Parameter Ensemble relates to model error and how it can inform model development."

19. L. 71: "it can inform model development". Please reformulate

We think this sentence is clear enough : the model error decomposition presented here can help identify the part of error that is not affected by model tuning, which can bring information about the limitations of a climate model. Additionally, identifying model candidates presenting diverse regional biases can be of interest for better understanding the representation of certain mechanisms in the model, or for considering parameter uncertainty in studies of climate change impact.

20. L. 77: Add also a mention to the conclusions in section 5.

Agreed

**Methods**

1.  L. 79 - 86: Put that paragraph in a subsection named "Used model", for example. Ok
2.  L. 79: Here, you refer the model as CNRM-CM6. Later in the paper, you use the name "CNRM-CMIP6", or "CNRM-CM". Please, use the same name everywhere and clarify with ARPEGE-Climat. If you consider that the model used is the full Earth System model, use "CNRM-CMIP6", while if you consider that you run only the atmospheric component, please use "ARPEGE-Climat".

Ok

3. L. 79: Specify that CNRM-CM6 is one of the CMIP models.

Ok

4. L. 80: To avoid confusion with emulator, I would say "climate model" instead of "climate simulator".

Ok

5. L. 80: Definition of PPE abbreviation not necessary Ok
6. L. 81: Can you define AMIP abbreviation ?

Ok (we added a reference)

7. L. 81: I would say "102 simulations differing by their parameters value". Ok
8. L. 82: That could be interesting to give some example of parameters. Do they all appears in physical parameterization ? A table with the whole list of parameter could be added in appendix.

Ok - see Table 1, added in Appendix

| Name | Minimum | Maximum | Reference | Description | Units |
|---|---|---|---|---|---|
| AKN | 0.06 | 0.28 | 0.126 | Strength of the turbulent mixing | - |
| ALPHAT | 0.5 | 3.0 | 1.13 | Strength of the turbulent mixing for temperature (Prandtl number) | - |
| ALD | 0.5 | 3.0 | 1.18 | Strength of the turbulent kinetic energy dissipation | - |
| ALMAVE | 0 | 30 | 10 | Lower bound of the mixing length | m |
| AGREF | $-0.5$ | $-0.01$ | $-0.36$ | Parameter in the boundary-layer-top entrainment parameterization | - |
| AGRE1 | 0 | 10 | 5.5 | Parameter in the boundary-layer-top entrainment parameterization | - |
| AGRE2 | 0 | 10 | 0 | Parameter in the boundary-layer-top entrainment parameterization | - |
| RAUTEFR | $0.5 \times 10^{-3}$ | $1 \times 10^{-2}$ | $1 \times 10^{-3}$ | Inverse timescale for liquid autoconversion | $s^{-1}$ |
| RQLCR | $0.5 \times 10^{-4}$ | $1 \times 10^{-3}$ | $2 \times 10^{-4}$ | Critical liquid water content for liquid autoconversion | $kg\ kg^{-1}$ |
| RAUTEFS | $0.5 \times 10^{-3}$ | $1 \times 10^{-2}$ | $5.2 \times 10^{-3}$ | Inverse timescale for ice autoconversion | $s^{-1}$ |
| RQICRMIN | $0.1 \times 10^{-5}$ | $0.1 \times 10^{-7}$ | $0.1 \times 10^{-6}$ | Critical ice content for ice autoconversion at low negative temperatures | $kg\ kg^{-1}$ |
| RQICRMAX | $0.05 \times 10^{-4}$ | $1 \times 10^{-4}$ | $0.21 \times 10^{-4}$ | Critical ice content for ice autoconversion at high negative temperatures | $kg\ kg^{-1}$ |
| TFVL | 0.001 | 0.2 | 0.02 | Falling speed of cloud water droplets | $m\ s^{-1}$ |
| TFVI | 0.001 | 0.2 | 0.04 | Falling speed of cloud ice crystals | $m\ s^{-1}$ |
| TFVR | 0.1 | 6.0 | 3.0 | Falling speed of rain | $m\ s^{-1}$ |
| TFVS | 0.1 | 6.0 | 0.6 | Falling speed of snow | $m\ s^{-1}$ |
| RKDN | $3 \times 10^{-5}$ | $7 \times 10^{-5}$ | 5e-05 | Minimum drag for the convective updraft vertical velocity | $Pa^{-1}$ |
| RKDX | $8 \times 10^{-5}$ | $6 \times 10^{-4}$ | $1 \times 10^{-4}$ | Maximum drag for the convective updraft vertical velocity | $Pa^{-1}$ |
| TENTR | $2 \times 10^{-6}$ | $1 \times 10^{-5}$ | $4 \times 10^{-6}$ | Minimum turbulent entrainment in the convective updraft | $Pa^{-1}$ |
| TENTRX | $3 \times 10^{-5}$ | $1 \times 10^{-4}$ | $6 \times 10^{-5}$ | Maximum turbulent entrainment in the convective updraft | $Pa^{-1}$ |
| VVN | $-1$ | $-5$ | $-2$ | Critical convective updraft Vertical velocity for maximum entrainment and drag | $Pa\ s^{-1}$ |
| VVX | $-25$ | $-50$ | $-35$ | Critical convective updraft Vertical velocity for minimum entrainment and drag | $Pa\ s^{-1}$ |
| ALFX | 0.01 | 0.1 | 0.04 | Maximum convective updraft area fraction | - |
| FNEBC | 0 | 20 | 10 | Parameter for computing the convective cloud fraction | - |
| RLWINHF_ICE | 0.5 | 1.0 | 0.9 | Ice cloud heterogeneity coefficient in the longwave spectrum | - |
| RLWINHF_LIQ | 0.5 | 1.0 | 0.9 | Liquid cloud heterogeneity coefficient in the longwave spectrum | - |
| RSWINHF_ICE | 0.5 | 1.0 | 0.71 | Ice cloud heterogeneity coefficient in the shortwave spectrum | - |
| RSWINHF_LIQ | 0.5 | 1.0 | 0.71 | Liquid cloud heterogeneity coefficient in the shortwave spectrum | - |
| RELFCAPE | 0.2 | 10.0 | 2.0 | Parameter used in the convection scheme Convective Available Potential Energy closure | - |

**Table 1**. Description of the 30 parameters.

9. L. 81: Why do you choose a period of 3 years ? And why this period ? Is there a reason ?

> The choice of the period was detailed in Peatier et al. (2022), it was the time needed for the feedback to stabilize

10. L. 82: I am not sure it is enough to use 102 simulations for 30 perturbed parameters. Some later point of your paper make me questioning this design of experiment.

> We agree that 102 members is really restrictive. Enhancing the PPE size would be required in order to improve our approach, and we will make sure to highlight this point in the discussion.

11. L. 82: Did you use a classic Latin Hypercube Sampling ? Or did you use a maximin method with a centered design ? In Peatier et al (2022), I understand you used, a maximin method. However as 68 simulations crashed, the design of experiment may not sample some part of the parameters. That could explains why you got a crash with Candidate #1 (in uni-variate case). Please, comment this.

> We use exactly the same PPE as in Peatier et al. (2022). This is the same LHS with the maximin method, and we indeed had some crashes in the simulations. We agree that we are not sampling all of the parameter space, because of these crashes, and we expect to have some crashes also in our candidates (as some areas of the parameter space seem to lead to crashed simulations). This design could be improved by using sensitivity analysis to better understand which sub-space of the parameter led to the crashes, re-sampling the space avoiding these areas, and running another PPE better exploring the newly designed parameter space.

12. L. 83: Please define $\theta$ and define $n$ earlier.

> Ok

13. L. 84: Can you clarify the "climatological annual means" ? I understand that it is the mean over the period of 3 years and not the mean for each year. Because you need to reduce your calibration to 2 dimensions (parameters and spatial dimension) and thus, delete the temporal dimension.

> Ok - yes, this is the annual means average over the 3 years

14. L. 86: "Elements.. area": which elements ? Which area ? What are these "weights" ?

> This sentence was out of place

New paragraph L.79 - 84 :

**" 2.1 Model and Perturbed Parameter Ensemble (PPE)**

The model used in this study is ARPEGE-Climat, the atmospheric component of the CNRM-CM6 climate model, referred to as f, the climate model. The reference

configuration of this model will be referred to as CNRM-CM6-1 and has been tuned by the model developers for the CMIP6 exercise (Roehrig et al., 2020). A PPE of this model f is created, containing 102 amip simulations (Eyring et al. (2016)) differing by their parameter values, representing the period 1979-1981 (3 years), with pre-specified Sea Surface Temperatures (SSTs) (Peatier et al. (2022)). Thirty model parameters (see Appendix A1) are perturbed with a Latin Hypercube Sampling (LHS) strategy, producing a variety of simulated climate states in the experiment : F = (f(θ1),...,f(θn)) based on a space-filling maximin design θ = (θ1,...,θn) (Peatier et al., 2022), with n = 102 and θi a vector of 30 parameter values. For the present study, we consider the annual means averaged over the whole 1979-1981 period. We write the model output f(θi) as a vector of length I, such that F has dimension I×n, where n is the number of ensemble members (n=102) and I the number of grid points (I=32 768)."

**New reference :** Eyring, V., Bony, S., Meehl, G. A., Senior, C. A., Stevens, B., Stouffer, R. J., and Taylor, K. E.: Overview of the Coupled Model Intercomparison Project Phase 6 (CMIP6) experimental design and organization, Geosci. Model Dev., 9, 1937–1958, https://doi.org/10.5194/gmd-9-1937-2016, 2016.

1. EOF Analysis

1. L. 95: What is $\gamma$ ?

   These are the different modes of variability (eigen vectors) of the EOF basis

2. L. 100: Can you clarify which mean $\mu$ is ? Ensemble and temporal mean over the 3 years period ? Is there a spatial mean ? I suppose no.

   It is the ensemble mean of the annual means averaged over the 3 years period (temporal means). \mu is a 2D matrix (lat x lon), without spatial mean.

3. L. 101: Introduce the "PC" abbreviation here.

   Ok

4. L. 105: If $r_f$ includes $\mu$, why does $\mu$ appears in the equation ? Is there a mistake ?

   Yes, there is a mistake. Written as it is, \mu is not part of the residual rf. We will remove \mu from the equation and write that rf and contains \mu

5. L. 107: Generally $y$ is used for the result of a function (for example a model). For observation, I would suggest $o$.

   (Salter et al. (2019)) and (Rougier (2007) use z to refer to the observations. We will replace "y" with "z" in the paper to match their notations.

**2. Model error partitioning**

1. L. 116: Replace "partitioning" to "decomposition"

   Ok

2. L. 119: $\theta_i$ is not a calibration, it is a value of parameters. Delete the word "calibration".

   Ok

3. L. 122: Can you argue the use of MSE instead of RMSE or bias, or other error definition ? Furthermore, in the rest of the paper, it is mentionned sometimes MSE, some other times RMSE. Is it possible to be consistent and say RMSE if RMSE have been used in the two applications ?

   Ok

4. L. 123: What is $c_r$ ?

   This is a typo, this should be c_y

**4. The discrepancy term**

1. L. 132: Correct the $*$ position: $\theta^*$ - OK
2. L. 132: I think the formulation is wrong. $\theta^*$ is not an optimal calibration. It is the optimal set of parameter $\theta$. Am I wrong ? OK
3. L. 134: Delete "how informative the climate model is about the true climate, and it measures" OK
4. L. 135: Replace "real" by "measured". Even observations are not perfect. The reality is between models and observations. OK
5. L. 138: There is also the fact that in numerical models, the Earth system is discretized and not continuous. The fact that is is discretized is also a source of error. OK
6. L. 138: Generally, to address this issue (parameters not included in calibration process) and in order to include all parameters that must be calibrated, sensitivity analyses are performed (Saltelli et al, 2004). They can estimate the influence of parameters on model outputs and thus give a list a parameter that must be calibrated.

   Not entirely, because we would still be limited by the choice of the initial parameter space to sample. A sensitivity analysis such as the one described in (Saltelli et al., 2004) requires an emulator mapping from the parameter space to the climate model outputs, and such emulators have to be trained on a PPE. Given the huge number of parameters used in complex models such as GCM, it seems unlikely that we would be able to run a PPE perturbing all of the model parameters. Subjective choices, informed by expert advice, have to be made

when selecting the parameters to perturb in the first place, and a sensitivity analysis will never be able to go beyond this choice. We would gain information about the effect of these selected parameters on the model output variance, but we would not be able to know whether perturbing a parameter not included in this initial sampling would have a greater effect.

7. L. 139: See comments for L. 132 ("calibration") OK
8. L. 139: Why do you present the uni-variate example and not the multi-variate one ? The uni-variate one is just a specific case of the multi-variate one.

   We do not present the uni-variate example, j could be any variable. In a multi-variate context, there would be as many discrepancy terms as variables considered - or we would have a general discrepancy term of 3 dimensions (lat, lon, nj), with nj the number of variables considered for the performance assessment. For more clarity, we will change the sentence L.139 "Considering a variable j, the discrepancy term nj is defined as : "

9. L. 147: This sentence repeats what have been already said in L.134

   OK

10. L. 154: Can you clarify the word "operational" ?

    OK

11. L. 155 - 156: reformulate the sentence. For example, "in this work, we propose several $\hat{\theta}$ that approximate $\theta^*$".

    OK

12. L. 156 - 157: What is $m$ ? I suggest : "to select $m$ optimal model candidates"

    OK

**5. Emulator design and optimization**

1. L. 162: PC abbreviation not defined OK
2. L. 163: Does the linearity assumption is right ? Can you justify such choice of emulator. Did you compare to other emulators ?

   Yes - see answer to major comment and Appendix

3. L. 168: Can you clarify "comparably performing" ?

   Their emulated parametric errors are bellow the parametric error of the reference model.

4. L. 169: Can you clarify "objective function" ?

   We mean "trade-offs in the multi-variate spatial errors"

5. L. 171: Please specify the characteristic of this new LHS sampling. Is it a centered one ? with maximin space-filling method ?

It is a maximin space-filling, this will specified

6. L. 171: instead of "distribution", use "ensemble of " OK

7. L. 172: What is the reference calibration ? I suppose it is the model using the default value of parameters that have been calibrated manually.

Yes it is, we will detail this in the new Section **2.1 Model and Perturbed Parameter Esnemble (PPE)**

8. L. 172: This lays on the fact that you suppose the reference as calibrated and also to the fact that the error difference between the optimal simulation and the reference is smaller than the emulator error. Did you verify this second point ?

Yes, the reference model version has been validated by experts and we assume its calibration to be near-optimal. We did not include the emulator error in this threshold, explaining also why we end up with actual candidate simulations showing higher error than the reference. This is a design choice aiming at preserving diversity in the sub-set of candidates, allowing them to show slightly higher integrated error than the reference.

9. L. 177: "pattern error". Do you mean "spatial error" ? Yes

**6. Selection of diverse candidate calibrations**

This section has been completely rewritten to present the clustering analysis. We have better justified the choice of nk (or k) and presented the method for both the univariate and the multivariate applications.

1. L. 180: "plausible optimal model configurations" We will call them "near-optimal"
2. L. 185: How do you define $n_k$? What is the difference with $k$ defined in L.180 ? No difference, we will just call them k
3. L. 185: Why don't you fix $n_k$ = 5 for the two applications ? Why don't you select randomly 5 candidates, 100 times, and select the group of 5 candidates amongst the 100 groups which have the highest variance ? See answer to Major comments
4. L. 185: Maybe clarify that "diverse" means the selected candidates have a high variance See answer to Major comments
5. L. 186: What is $n_j$? The number of variables considered

**First application : surface temperature error**

1. L. 192: At equation (5), you consider a minimization of MSE, while here, you consider RMSE. Please, if you really use the RMSE, use the RMSE in equation (5)

We use the MSE, we will change L.192

**1. Assessing meaningful number of degrees of freedom**

1. L. 210: At line 192, you consider RMSE, while here MSE. Please clarify.

   This is MSE

2. L. 213 - 215: This is not coherent with lines 113-114. Even if $q = n = 102$, because of the observations, the non-parametric component (residual) is not null, so the reconstruction of the full error $e$ is not perfect. Therefore, it is the best that you can get not the perfect one. (The same for L. 224 and L.252)

   We call "perfect reconstruction" the reconstruction that uses all of the 102 EOF modes, and Figure 1 illustrates the fact that, even retaining all of the EOF modes, the non-parametric error is indeed, not null. The "perfect reconstruction" does not reach the amplitude of the full error, because of the projection of observation.

3. Figure 1: I suggest this caption: "Full model error $e_{tas}$ and its parametric component $p_{tas}(\theta_i)$ for different truncation length : $q = 5$ (red dots), $q = 20$ (blue dots),$q = 50$ (pink dots), $q = 102$ (orange dots). a: Full error partitioning in parametric and non-parametric components in the PPE members $f(\theta_i)$ ranked from lowest to highest error. b: correlation between the full error $e_{tas}$ and its parametric component $p_{tas}(\theta_i)$ within the PPE."

   OK

4. Figure 1: I would recommend to add a) and b) in the upper part of the figure in order to differentiate the left andright panels and to refer to the "Figure 1a" and "Figure 1b", in the text. And I would do the same suggestion for all figures.

   OK

5. Figure 1: The non-parametric error is symbolised with dashed line. Logically, the non-parametric error should bethe difference between the full error and the parametric error. So, the dashed line should be between the black dots and the colors dots, not between the full error and 0. Am I wrong ?

   Yes, this figure was confessing. We simplified it :

[Figure]

**Fig 4**. Full model error $e_{tas}$ and its parametric component $p_{tas}(\theta_i)$ for different truncation length : $q = 5$ (red dots), $q = 20$ (blue dots), $q = 50$ (pink dots), $q = 102$ (orange dots). Full error partitioning in parametric and non-parametric components in the PPE members $f(\theta_i)$ ranked from lowest to highest error.

6. Figure 1: In the rest of the paper, you choose $q = 18$. Why don't you present the result for $q = 18$ in this figure ?

   Because the choice of q=18 is based on results presented in Figure 2, there is no reason to highlight this particular choice of q=18 in Figure 1.

7. L. 217: "A number of feature are notable in Figure 1." Not sure that this sentence is really pertinent.

   Agreed

8. L. 217 - 222: Why do you begin to analyse the right panel ? If you want to analyse it first, put the right panel on the left.

   OK

9. L. 219: Instead of using a minus (-) symbol use a colon (:)

   OK

10. L. 221: Instead of "variation", I would use "spread" or "variance". (The same for L. 228)

OK

11. L. 221: Maybe add a comment to say, that you will focus in the rest of the paper, on the 5 first modes.

We do not focus on the 5 fist modes, we chose a truncation at q=18 for the analysis. This is only for graphical purposes that Figure 3 shows the 5 first modes (to avoid overloading the figure with 18 by 18 pairwise plots).

12. L. 225: The percentage is 26% in average over all PPE. However, it can be drastically different between the best and worst PPE. Can you give a range of this ratio ? Is it ok if the major part of the error is non-parametric for the best PPE, particularly if $q = 5$ ? Can you comment it ?

The percentage of non-parametric error is not drastically different from one model to another. When considering the perfect reconstruction (q=102), as it is the case in the sentence L.225, the range is actually negligible. It can be seen on Figure 5 (Figure 2 in the paper), where the range of non-parametric error depending on truncation length is represented in teal. For q=102, there is almost no range, and the non-parametric error represents 26% of the full error (as stated L.225). However, we agree that for a truncation at q=5, the percentage of non-parametric error does vary a little bit from one model to another (see range FIgure 5), and Figure 4 (Figure 1 in the paper) suggests that the best models have the biggest percentage of non-parametric error. This might indicate that a truncation at q=5 will not be adequate in order to fully capture the parametric error, and that the best models tend to reduce errors in the first 5 modes of the EOF.

[Figure]

**Fig 5**. Truncation choice based on parametric emulation and error decomposition. The plot shows the ratio of the error components compared to the full error etas(θi) (in green) as a function of the

number of modes of variability retained. The lines are the ensemble means and the shadings represent the standard deviations. The plot shows the ratios of the PPE parametric error (dark blue), the PPE non-parametric error (light blue), the reference calibration parametric error ptas(θ0) (red dotted curve) and the GMMIP parametric error (orange). An example of truncation at q = 18 is represented on both plots by the black vertical line.

13. L. 227-229: Can you better explain your reasoning ?

Agreed, new sentence L. 227-229 : "However, even for the perfect reconstruction of the model error (when q = 102), a non-null non-parametric component exists, and its ratio corresponds to 26% of the full model errors averaged over the PPE members. This ratio increases when retaining less EOF modes, and a large fraction of the model error pattern is not represented within the parametric component. For a truncation of q = 5, the non-parametric component of the error u_tas is 53% of the total e_tas(θi), in average over the PPE. Together, this implies that the variance in model error seen in the PPE can be explained by a small number of modes, but a significant fraction of this error is not represented within the parametric component of the error decomposition."

**2. Truncation and parametric emulation**

1. L. 233: Why don't you use the Leave-One-Out Cross validation method ? Maybe add a subsection in the Method section to present the validation method you used for validating the emulator.

   Agreed - see answer to major comments

2. L. 237: No, you cannot assess the predictive performance of the emulator properly by looking at the correlation between prediction and true data.

   See answer to major comments

3. L. 240: What is $x$ ? The abscissa ?

   Yes, but this should be "q" in order to stay consistent with previous notation

4. L. 246: The chosen threshold is quite subjective. Maybe it is better to fix a variance at 95% and deduce a threshold in correlation, instead of fixing, arbitrary, a correlation at 0.5.

   See answer to major comments

5. L. 249: Is this the ratios for the emulated PPE ? Is there a link between the left and right panels ? If not, maybe consider 2 different figures for these two plots.

   No, it is the ratio within the actual PPE. The link between the two figures is that they are both used to detail and justify the choice of truncation length.

6. L. 249: Help the reader and add reference to the colour of the line : "PPE parametric (blue line) ..."

Agreed

7. L. 251: Delete the word "example"

Agreed

8. L. 253: Give the equation since line 251.

Agreed

9. L. 257: Introduce GMMIP

Agreed

10. L. 267: "This emulator is then optimized...". Not sure it is an optimization of an emulator, but more an optimization of the chosen set of calibrated parameter.

No, here it is an actual example of optimization, where we used an optimizer to look for a minimum. This is not a method that is presented in the rest of the paper and this is not what is used for the selection of candidates (where we use a clustering method). This is just an illustration of the effects the addition of higher modes can have on an optimization.

11. L. 271 - 273: Please reformulate

New sentence : "The difference between the PPE mean and this example of optimal calibration becomes constant when q = 7 or more, suggesting that there are no improvements of the optimization when adding modes higher than 7."

12. Figure 2: I suggest to do not present the method on the caption : "Truncation choice based on parametric emulationand error decomposition. a: Correlation between the emulated and true PCs coefficient of the surface temperature EOF, for the different modes of variability and for : the training set (blue curve), test set (orange curve). Mean is represented by dots and standard deviation by error bars. Averaged correlation over the modes cumulatively is shown by the red curve and the standard deviation by the red shading. b: Ratio as function of the number of modes of variability retained, of the error components compared to : the full error $e_{tas}(\theta_i)$ (in green), the PPE parametric error (dark blue), the PPE non-parametric error (light blue), the reference calibration parametric error (red dotted curve) and the GMMIP parametric error (orange). The lines are the ensemble means and the shadings represent the standard deviations. The black vertical line represents the truncation at q = 18."

Agreed

3. Trade-offs in models candidates

1. L. 283: at line 171, you precise that you used $10^6$ emulated simulations, while here, it is indicated 100.000. Is there an error ?

   Yes it is, we use a 100.000 LHS throughout

2. L. 284: "The five selected parameters calibrations were then used..." -> "The five calibrated set of parameters were then used..."

   Agreed

3. L. 286: The fact that there is a crash by using the real model, suggests a problem in the emulator building, maybe a lack of data in the training phase of the metamodel. Can the author comment this crash ?

   We had some crashes in the simulations of the PPE. Because of these crashed, we might not be sampling all of the parameter space, and we expect to have some crashes also in our candidates (as some areas of the parameter space seem to lead to crashed simulations). This design could be improved by using sensitivity analysis to better understand which sub-space of the parameter led to the crashes, re-sampling the space avoiding these areas, and running another PPE better exploring the newly designed parameter space.

4. L. 286: Concerning the other simulation using the 4 calibrated set of parameters, it can be interesting to quantify the emulation error (relative error between emulation and real results). That could be also, another way to verify the ability of the metamodel to emulate ARPEGE-Climat.

   See answer to major comments

5. L. 291: "provides some confidence in [..] the emulation skill". I am not convinced by this conclusion, particularly because the emulated and real simulation do not give similar results. For EOF3 according to EOF1 (1st columns, 3rd line in Figure 3), I agree. But for EOF2 according to EOF1 (1rst column, 2nd line), the emulated set #3 (orange dot) is closer to the real set #4 (blue triangle) than the real simulation #3 (orange triangle)... Maybe a calculation of distance between all pairs of emulated and real simulations can justify your conclusion better. Verify that, for example, the mean (average over all EOF couple) distance between the emulated simulation #1 and the real simulation #1 is smaller than with the other real simulations. Do that calculation for each emulated candidate simulation, and each EOF couple.

   See answer to major comments

6. L. 301: "Figure 3 also allows to see", please rewrite this formulation.

   Agreed "Figure 3 illustrates ... "

7. L. 301: if the "PC" abbreviation is used and defined, you can replace "Principal Components" by its abbreviation

*Agreed*

8. L. 304: Mode 2 is not constraint according to Figure 3. Why the author says "stronger than on the modes 3 to 5" and not "stronger than the other modes" ? Please, clarify.

   *Agreed - "stronger than the other modes"*

9. L. 309: "perform equally well on all modes". I understood that is it not true for EOF5 (in line 296) so not "on all modes". Am I wrong ?

   *We wrote "it is impossible for the model candidates to perform equally well on all modes and fit observations perfecty", so there is no contradiction with what was saif L.296 statting that a lack of diversity in the candidates appears in mode 5*

10. L. 309-315: Is it possible to calculate the distance between calibrated simulation (real or emulated ones) with observations, in order to justify your comment ?

    *See answer to major comments*

11. L. 309-315: Clarify that if you consider here the emulated or the real candidates.

    *We refer to real candidates - agreed*

12. L. 310: Help the reader and add a "in green" after "candidate 2".

    *Agreed*

13. L. 312-313: Not sure it is pertinent to comment on mode 5 as observations is outside the emulated ensemble.

    *We think that it is still interesting to note*

14. L. 317: Do the 5 candidates really have a comparable $e(\hat{\theta}_i)$, while candidate #4 performs better in terms of $p$ ? You give the $e$ value in line 336. Maybe, you can discuss this RMSE in section 3.3 instead of 3.4 (in that case, delete the RMSE appearing in Figure 4).

    *We discuss the values of p and e later on in the paper*

15. Section 3.3: Add a comment to the fact that emulated candidate #1 is generally far away from the other candidates.

    *This is the candidate that crashes, so we might think that he is in a poorly sampled part of the parameter space. Figure 3 also suggests that he might present outputs very different from the other candidates. This is an interesting point ...*

    *However the candidates will whange with the new methodology.*

16. Figure 3: I suggest this caption: "Correlation between the different standardised PC (obtained from the 102 memberPPE EOF) for the 100.000 emulated simulations (light gray), the optimal emulated simulations (dark grey, parametric

error lower than the reference), the 5 selected emulated candidates (colored dots), the 4 real ARPEGE-Climat simulations (colored triangles), the reference simulation (star) and the observation (cross).

Ok

17. Figure 3: Please, clarify between CNRM-CM, CNRM-CM6-1 and ARPEGE-Climat. If you use one model, pleaseuse the same name all along your paper (see comment L.172.).

Agreed

**4. Example of temperature discrepancy term partitioning**

1. L. 333: At line 279, you have already mention that you will considered, in the rest of the paper, $q$ = 18. It is not necessary to mention it again.

Agreed

2. L. 334: Specify that you consider the emulated or real calibrated simulation. I understand you consider here the emulated candidates, but why not the real one ?

Figure 4 presents both the emulated and the real ones next to each others

3. L. 337: Please, add a comment on Candidates #5 which has a smaller $p$ than the reference but a higher $e$.

Candidates will whange with the new methodology

4. L. 339: Are you comparing the 3th line plots at 1st and 3rd columns ? I see an overestimation of the negative bias or a underestimation of positive bias (not an overestimation of positive bias as written) in the emulated simulation (3rd column compared to 1st). There is also a too large negative bias in Antarctica in the emulated candidate #4. Can you comment it ?

Candidates will whange with the new methodology

5. L. 346: According to the colorbar, positive bias is in red, and in the Figure, mountain regions are in blue : there is not a positive bias in mountain region. The same for Africa, the bias is positive, not negative as written. Is there a mistake or did I misunderstood the analysis ? Please clarify.

There is indeed a mistake in the text about the sign of the bias, but candidates will whange with the new methodology

6. L. 348: "vary from a model to another" : the model does not changed (it is still ARPEGE-Climat or CNRM-CMIP6, whatever the name you want to give). Maybe you mean "vary from a parameter set to another" ?

"A model" refer here to $f(\theta_i)$, a particular configuration of ARPEGE-Climat

7. L. 348: I see the opposite: strong negative bias on North America and strong positive bias on central Africa.

*There is indeed a mistake in the text about the sign of the bias, but candidates will whange with the new methodology*

8. Figure 4: I suggest to change the order of the column : emulated parametric error in column 1, parametric error in column 2, non parametric error in column3 and full error in column4. But it is just a suggestion.

*Agreed*

**Second application : multi-variate error**

**1. Variables, EOF analysis and truncations**

1. Table 1: It is quiet confusing to use radiative data from the 2000-2002 period for a study on the period 1979-1981. Can you explain this choice ?

*The CERES observational data starts in 2000*

2. Table 1: Instead of "field", use "observable variables" as in the caption and instead of "citation", use "data productreference" for example.

*Agreed*

3. L. 366: "spanning model components": can you clarify ?

*We mean that the atmospheric component is not the only one that needs to be tuned when considering a climate model. The ocean, the surface and the sea-ice models all need to be tuned, which makes the tuning process and the uncertainty quantification associated, even more complex.*

4. L. 370: here, the author used the MSE and not the RMSE contrarily to the uni-variate case. Is it possible to justify ?

*Indeed, we always use the MSE*

5. L. 372: Is it really the annual mean or is it the mean over the 3 years period ?

*It is the mean over the 3 years period*

6. L. 377 - 410: As already said for section 3.2, do not consider only the correlation to validate your emulator but also the relative error.

*See answer to major comments*

7. L. 380 -385: same comments as in L.246. To stay robust, you should fix the percentage of variance you want, and deduce to that percentage, the number of modes. A correlation of 0.5 does not mean the same for each variable.

*See answer to major comments*

8. L. 386: You do not show results for q=4, but for q=5 in Figure 6. So the variance model error is already very-well represented by the first 5 modes, instead of 4.

   Agreed

9. L. 387: I suggest to present Figure 6 before to analyse it.

   Ok

10. L. 388: "66%" Is it enough ?

    Here we are not talking about the variance explained by the truncated EOF, which is 92% for sea level pressure, we are referring to the part of error included in the parametric component.

11. L. 390: Do you consider here, the emulated simulations or the 102 real ones ? I understand the real one, but precise it.

    Yes, the real ones. We do not have access to the non-parametric components of the emulations, as we only emulate within the EOF space. So we are always talking about actual simulations when refering to non-parametric components.

12. L. 390: As already said, do not consider correlation only for your validation. Consider also the relative errors, here to validate your EOF truncation.

    See answer to reviewers

13. L. 390: Why does the non-parametric error is averaged but not the parametric component ? Is it the averaged according to the 5 climatological field ?

    To validate the fact that, with enough EOF modes retained, we can reconstruct the full error from the parametric component, when using the mean non-parametric component as an approximation. It is averaged separately on the different variables.

14. L. 393: "As expected, the PPE mean non-parametric components decrease as higher EOF modes are retained for the reconstruction but is never equal to 0 (even for a full reconstruction of q = 102). This is due to the fact that observations can never be fully captured by their projections into the model EOF basis (Figure 6)"

    Agreed

15. L. 402-404 : Be consistent with model name all along the paper.

    Ok

16. L. 402: as already said, define your reference calibration.

    Ok

17. L. 405. Maybe, there is a known bias in surface temperature in the ARPEGE-Climat model, (logically a bias that appears due to the use of the SURFEX surface parameterization). Can you discuss about it ?

Candidates will change with the new selection method, so this point about the candidate models outperforming the reference in terms of surface temperature might not be true anymore. But even if there was a known bias in the reference model, what is pointed out here goes beyond that, noting that we could reduce the surface temperature bias with another set of model parameters.

18. L. 405: ". This is a simple illustration of a complex tuning problem, and based on the results we obtained in the uni-variate application. It seems likely that comparably performing parameter configurations potentially exist for a multi-variate tuning problem, making different model trade-offs among both climatic fields and EOF modes representations of uni-variate errors (Figure 3)"

    Agreed

19. Figure 5: You do not comment the 2nd column of the figure. If you did not need it to support your discussion, thesegraphs can be delete of your paper.

    We think they are interesting. We will add a comment about them

20. Figure 6: What is the grey shading ?

    The PPE means non-parametric component, the part that has been added to all the points in order to reconstruct the full error.

21. Figure 6: Instead of "(y axis)" and "(x axis), use "coordinate" and "abscissa".

    Agreed

22. Figure 6: Logically, all your 102 CNRM PPE data must be used for training and testing your emulator. So, all dotsmust be in green or orange, no one in black. I don't understand your figure.

    The black dots are the actual runs, the 102 PPE dots. The green dots are the emulations of the train set and the orange dots are the emulations of the test set.

23. Figure 6: As you should sampling the training and testing datasets 10 times (according to L.236), the trained andtesting datasets are not fixed: some point should appear in green for a sampling, but maybe in orange for another sampling. I still not understand the orange and green dots.

    Here we do not say that we are sampling the training and testing datasets 10 times. That was the case for Figures 2 and 5, but in Figure 6 we are just showing a one time splitting of the dataset between train and test set, and a reconstruction of the full error.

24. Figure 6: Please, do not present the training dataset (80%) and testing dataset (20%) in the figure caption.

25. Figure 6: Why do you use a training dataset of 80% for the multi-variate case while 90% for the uni-variate case ?Did you also repeat the training 10 times ?

    This is a different analysis than in Figures 2 and 5, that brings a different message than a validation of the emulators. We are mostly interested in the

reconstruction of the full error from the emulation of the parametric components depending on the truncation choice and the variables. We used another ratio of training/test and we did not repeat the analysis 10 times as it was the case for Figures 2 and 5.

**2. Candidates selection in a multi-variate context**

1. L. 412 - 426: This paragraph must be in the Method section.

   Agreed

2. L. 413 : For the subset candidate selection, you maximise the variance of the multi-variate metric. The minimisation of the multi-variate metric is not the process to select a set of optimal candidate.

   The candidate selection changed completely - see answer to major comments

3. L. 425 : Why $n_k = 4$ and not $n_k = 5$ as in the uni-variate application ?

   The candidate selection changed completely - see answer to major comments

4. L. 429 - 434: Help the reader by adding in parenthesis to which feature in the Figure it corresponds : "Among the 4 selected candidates (blue dots)", "than the reference model (yellow dashed line", "PPE mean (red dot)", "mean of the 40 CMIP6 models (green dot)"

   Agreed

5. L. 433: "CNRM model grid before"

   Agreed

6. L. 433: Precise here (not in the figure caption) that observation have been interpolated also.

   Agreed

7. L. 435: Is there a justification to apply the uncertainty (standard deviation) of CMIP6 model to CNRM reference model and not the uncertainty of CNRM model itself (CNRM-PPE) ?

   The PPE variance can not be used as an estimate of the tolerance we should have when considering the model performance, as the PPE members have not been tuned and many of them will be showing really bad performances. On the other hand, the CMIP6 models have all been tuned and validated by experts and they are all considered to show a plausible representation of the historical climate. We can use the variance of the error within the CMIP6 ensemble as a tolerance for the evaluation of our candidates.

8. L. 436: ",indicating ..." -> "This indicates"

OK

9. L. 440: Maybe present all the data used for this work in the Method section instead of presenting them in the results section.

Thanks for the suggestion, but we think that the method section is already quite dense, and this dataset is only used once in the paper. So it makes sense to introduce it in the results section.

10. Figure 7: I suggest this caption: "Multi-variate error $e_{tot}$ for the CMIP6 models, the CNRM-CMIP6 PPE members, the 4 optimal CNRM-CMIP6 candidates and the 10 members of CNRM reference model with different initial conditions. Each small dots correspond to a model, the bigger dots correspond to the ensemble means and the dashes are the standard deviations. The orange dashed line at 1.0 represents the CNRM reference model error. The green area indicates the interval of plus or minus one standard deviation of the CMIP6 errors, centered around the CNRM reference model error. "

Agreed

11. Figure 7: "available CMIP6 model" : Does this mean that all models are not used ?

We used all models that ran the amip-hist experiment and that made their outputs available on the ESGF platform.

3. Diversity of error patterns among candidates

1. L. 451: delete "for the selection"

Agreed

2. L. 451 - 475: I suggest to present the results in the order of candidate number : discuss firstly candidate 1, then candidate 2, ...
3. L. 433: to support your analysis, cite the value of RMSE.

Agreed

4. L. 451 - 475: Is it possible to attribute theses differences to particular parameters value ?

We could run a sensitivity analysis to identify the effect of the different parameter perturbation on model performances and error patterns. However, this is beyond the scope of the present study.

5. L. 471: "everywhere" -> "on the whole domain"

Agreed

6. L. 472: Delete "(Figure8)"

Agreed

7. L. 472: "not the worst of the selection" not coherent with "is the worst performing" at L.471

The candidates will change with the new selection method, as will the associated comments. But it is the worst performing when looking at multi-variate score, but it is not the worst of the selection when looking at the radiative fluxes only.

8. L. 475: Sentence not finished...

Agreed, thanks

9. Figure 8: Candidate 1 has no one green dot : all RMSE are higher than the reference. Please, add a comment on it.

The candidates will change with the new selection method, as will the associated comments.

10. Figure 8: Why is it always $\hat{\theta}_1$ in the grey rectangle ? Why the 1 ?

This was a mistake. It should be \theta_1, \theta_2, \theta_3, \theta_4

11. Figure 5 - 6 - 8 - 9 - 10: Keep variables appearing in the same order for all figures

This is the case

4. Examples of discrepancy term partitioning

1. Figures 9 and 10: add $p$ and $u$ in the grey rectangle, as you added $e$ in the Figure 8

Agreed

2. L. 483: delete some "are"

Agreed

3. L. 484: I am not sure that it validates the method properly. But, at least, it shows that the objectives are achieved.

Agreed

4. L. 488 - 505: The analysis is only conducted for SW. Do you have any comments on the other variables ?

The candidates will change with the new selection method, as will the associated comments.

5. L. 499 - 502: split this sentences in two different ones.

   The candidates will change with the new selection method, as will the associated comments.

6. L. 506: please, explain better the link with the effective degrees of freedom

   Agreed

**Conclusions**

1. L. 511: "perturbed physics" -> "perturbed parameters"

   Agreed

2. L. 512: "diverse" : reformulate

   This is the term that has been used throughout the paper, we think it is more consistent to use it also in the conclusions

3. L. 513: "a number of" -> "different"

   Agreed

4. L. 513: delete "which we illustrate ... (General Circulation Model)"

   Agreed

5. L. 523: delete "examples"

   Agreed

6. L. 525: "CNRM-CM", please use the same name for the same model

   Agreed

7. L. 525: Really $10^6$ simulations ? Or is it 100.000 as written in the paper ?

   Yes, 100.000

8. L. 525: "of the perturbed parameter of the parametric components of the model errors", please reformulate

   Agreed : "The optimization is based on multi linear predictions of the parametric components of the model errors, from a 10^5 LH sampling of the perturbed parameters"

9. L. 527: Use a more appropriated vocabulary than "diverse"

   This is the term that has been used throughout the paper, we think it is more consistent to use it also in the conclusions

10. L. 529: "CNRM discrepancy" -> "CNRM model discrepancy"

    Agreed

11. L. 547: Add a precision about the fact that candidate performs better than the reference in terms of $e$, while it has been optimised according to $p$

    Agreed

12. Add some perspectives. For example, it could be interesting to analyse the parameters value between the differentcandidates, in order to explain which parameters lead to such biases.

    Agreed

**Technical corrections**

1. L. 21, 27, 33, 59-60, 91, 152, 188-189, 560: put the citation in chronological order

OK

2. L. 23: correct the parenthesis "(CMIP; Eyring et al, 2016)"

OK

3. L. 40: correct the parenthesis "(such as global mean quantities, Bellprat et al., 2012 ; Williamson et al., 2015)

OK

4. L. 43: correct the parenthesis "(sometimes referred as an "iteratire refocusing"

   approach; Williamson et al, 2017) ... 5. L. 59: PCA not defined

OK

6. L. 52, 64, 69, 131, 148, 149, 163, 196, 206, 243, in Table 1: correct the parenthesis in the citation (parenthesis aroundthe date and not the name)

OK

7. L. 159: Error in the section referring: "Section 2.4" and "2.5" instead of "Section 2.3" and "2.4"

OK

8. L. 207: Error in the section referring: "Section 3.1" and "3.2" instead of "Section 3.2" and "3.3"

OK

9. L. 211: error in the section referring number

OK

10. L. 232: I think you mean "equation 14", not "Section 14".

OK

11. L. 242: "high-order modes" repeated twice.

OK

12. L. 245: "point"

OK

13. L. 268: "for" repeated twice

OK

14. L. 383 : "e.g.", belonging to which sentence ?

OK

15. L. 420 : "an the" -> "an" or "the" but not the two ones.

OK

16. L. 451: "The Figure" -> "Figure"

OK

17. L. 483: "on the other ends" -> "on the other hands"

OK

18. L. 484: remove the space before ":"

OK

19. L. 490: (Figure 10) -> (Figure 8)

OK

20. L. 525: LH abbreviation not defined

OK

21. L. 533-536: Did you want to do a list as in lines 514-521 ?

OK

22. L. 552: opimization -> optimization

OK

23. Figure 7: "dasehd" -> "dashed"

OK

24. Figure 7: "arounr" -> "around"

OK

---

## Author Comment (AC3)

**Response to Reviewer #3**

First of all, we would like to thank the reviewers for the comments on our manuscript. We appreciate all valuable suggestions, which helped us improve the quality of the paper. Following the reviewers' suggestions, we have improved the method to select optimal and diverse candidates. It is now based on a clustering analysis, and sensitivity tests regarding the number of clusters to consider is presented in an appendix of the paper. Given this new method, we found the need to consider around 10 clusters to explore the diversity of error patterns in the ensemble. Therefore, 10 candidate calibrations have been tested in the climate model and are presented in the paper. Most of the figures related to the candidate selection and their associated comments will be updated based on these modifications. Another appendix will be added regarding the choice and evaluation of the statistical models, and the paper will be slightly restructured to present all the methodological details in the Method section. Note also that, as requested by the reviewers, the title will be modified to : "Exploration of diverse solutions for the calibration of imperfect climate models".

Our responses to the reviewer #3 comments are described below. In the following, the comments from reviewers are in black, our answers are in blue.

**General comments**

1) The Introduction is rather terse and could be more welcoming for a broader audience. It would also be an improvement if the authors would discuss which fields are generally used in the calibration of climate models. Is 'calibration' the same process as what is often referred to as 'tuning'?

We will add a first paragraph in the Introduction that should help the paper to be more welcoming to a broader audience, as well as better explain the concept of model calibration (or 'tuning', as it is often referred to).

New paragraph in the Introduction : "General Circulation Models (GCM) and Earth System Models (ESM) are the primary tools for making projections about the future state of the climate system. It is an important goal of climate science to continually improve these models and to better quantify their uncertainties. Constraints on computational resources limit the ability to resolve small-scales mechanisms, and sub-grid parametrizations are used to represent processes such as atmospheric convection or clouds. These parametrizations are based on numerous poorly constrained parameters that introduce uncertainty in climate simulations. Therefore, climate models are subject to a challenging calibration (or 'tuning') problem. When used as tools of projection of future climate trajectories, they cannot be calibrated directly on their performance. Instead, assessment of performance and skill arises jointly from confidence in the understood realism of physical parametrizations of relevant physical processes, along with the fidelity of model representation of historical climate change.

Practical approaches to model calibration are subject to both data, time and computational constraints."

It would be nice if the authors would define the concept 'emulator'. It seems from the examples that this is here understood as statistical tools to interpolate between the parameters of which you have climate model experiments. However, it is my understanding that emulators also can be simple physical models.

We consider emulators to be a computationally efficient approximation of a more complex model. In the literature, these can be either purely statistical (as is the case here), or quasi-physical (as for simple climate models). In either case, there are degrees of freedom which can be adjusted to reproduce certain aspects of complex model response. The term 'emulator' has long been used in PPE analysis (Sanderson et al. (2008)), where mapping is from the perturbed parameter values to a selection of outputs of the climate model .

However, given the increased use of the term 'emulator' to represent simple climate models in, for example, IPCC applications, we propose to replace the term 'emulator' in the Introduction with 'statistical model' or 'statistical predictions'. Moreover, the Section **2.4 Emulator design and optimization** will be re-named **2.4 Statistical model and optimization** and will clearly define the concept of "emulations", that refers to the statistical model predictions.

**Reference :** Sanderson, B. M., Knutti, R., Aina, T., Christensen, C., Faull, N., Frame, D. J., … & Allen, M. R. (2008). Constraints on model response to greenhouse gas forcing and the role of subgrid-scale processes. *Journal of Climate*, *21*(11), 2384-2400.

Where does the word 'spatial' in the title come from? I guess the method is rather general although you apply it here to spatial fields.

We agree that the title of the paper was not the most appropriate and we changed it to : "Exploration of diverse solutions for the calibration of imperfect climate models".

2) It seems to be assumed that observations are perfect. This is not the case and often there are also errors originating from the finite sampling: the estimated climatology in both observations and models depends on the length of the time-series. How will these errors impact your results?

Indeed, in this study, we did not consider the uncertainty associated with observations, but additional analyses suggest that our results are sensitive to the observational dataset used. As a rough illustration, we present here (Figure 1), the projection of different surface temperature observations and reanalyses (BEST, ERA5, NCEP, NOAA) on the PPE EOF basis. We also selected candidates minimizing the error associated with each one of the projected observations. It appears that the distances between candidates in the EOF space is as large as the spread of the 5% best performing

models when considering a single observation dataset (BEST here), suggesting that considering a different observational dataset would affect our results to some degree. But the uncertainty range represented in the top 5% of models is representative of what we should expect from observational uncertainty.

[Figure]

**Fig 1**. Standardised principal components associated with the 5 first modes of the surface temperature EOF basis computed on the 102 members of the PPE. The Figure presents the projections of 1 000 000 emulated members (light gray), the 5% best emulated members compared to the BEST observations (dark grey), the projections of different surface temperature observations (colored crosses) and the 4 candidates CNRM-CM minimizing the error compared to a given observational dataset (colored triangles). Note that this analysis uses a different and larger LH sampling of the parameter space than what is presented in the paper, therefore the grey dots will not match the Figures of the paper.

Finding a formal way to include observational uncertainty in our method for candidate selection is beyond the scope of this paper. But we will add a paragraph in the conclusion to highlight this limitation and suggest the consideration of observational uncertainty as a perspective to improve the method.

3) The emulator used in this study seems to be linear multiple regression. Furthermore, the methodology of the analysis is based on EOF/PC analyses. But other emulators are non-linear. The output of such emulators are therefore not limited to the linear space spanned by the EOFs. So how would your analyses change with a non-linear emulator and how will it change your conclusions?

The emulators used in this study are Multi Linear Regressions (MLR), which take the model parameters as input and predict the Principal Components of the different climatic fields, used to reconstruct the 3D fields and the parametric component of the model error compared to observations. This choice was made because, given the modest sample size of perturbed ESM runs available, the linear model outperformed nonlinear techniques - implying that higher order parameter effects are not robustly sampled.

As a better discussion about the choice and performance of the emulators was requested by the Reviewers #1 and #2, we have decided to add an appendix showing the performance of our statistical model and comparing it to linear regression with penalization (LASSO model) and a non-linear emulator (a Random Forest).

In 10 random selections of out-of-sample test sets, we obtain an average correlation of 0.7 between the predictions and the true values of total error (Figure 1 (c)), with a RMSE between predictions and true values representing 8% of the total parametric error (Figure 1 (f)), which is sufficient to validate the use of this model for our study. However, results suggest that there is room for improvement, especially in the prediction of the LW error, and that another model could improve the predictions, as it is the case of the Random Forest model. The error bars associated with the prediction of the total error suggests that the MLR performance is sensitive to the test set selected and that the model will perform unevenly across the parameter space. Thanks to variable selection and regularization, the Lasso model seems a bit less sensitive to the test set selection for the prediction of total error, but the prediction of LW error is still a limitation. Figure 1 will be added in an appendix and we will add sentences in the main paper clearly stating the limitations of the emulators.

[Figure]

**Fig 1.** Correlations and RMSE (in % compared to the true values) between emulated and true parametric components of the errors within a test set representing 10% of the dataset. The evaluation is repeated 10 times with random sampling of training and test sets and the mean and standard deviation among these 10 evaluations are represented by the bars and the dashed lines, respectively. Performances are shown for (a), (d) a Random Forest, (b), (e) a LASSO regression and (c), (f) the Multi Linear Regression used in this analysis and. The EOF truncation lengths used to compute the parametric error are presented in Figure 2 and 5 of the paper.

In conclusion, it seems that using a non-linear emulator could improve certain aspects of the predictions, though our capacity to train such emulators is fundamentally limited by the sample size available in the dataset, and enhancing the size of the ensemble would be a necessary prerequisite to try to improve our statistical predictions. Gaussian Processes are non-linear models often used in PPE analysis and even though we did not test them in this study, we will add sentences in the conclusion suggesting it as a potential perspective to this work.

4) The analytical deviations can in places be hard to follow. I would suggest that a notation is used that differentiates between scalars, vectors, and matrices.

Thank you for the suggestion. After taking into account the feedback from all the reviewers, the point that seems the most confusing when trying to follow the equations is to identify whether we are talking about emulated scores or outputs of the climate models. To improve this aspect of the manuscript, we have decided to keep the notations as they are, but to use $p_{em,j}(\theta_i)$ for the emulated scores and $p_j(\theta_i)$ for the actual scores, in order to differentiate whether we discuss statistical predictions or climate model outputs.

5) I am also confused about the optimisation described in section 2.4 and 2.5. As far as I can see you don't apply a minimizing method but generates a lot of emulations with different parameters. Then you find a set of parameters with error smaller than the error from a reference model (what is this). And then you prune that set to a much smaller set (2.5). Is this correct? So the set you find are then not local minima?

Reviewer #1 also requested that we clarify the notion of 'optimality' in the paper, so we will be using the term "near-optimal" to refer the vectors of parameter values $\hat{\theta}_i$ associated with emulated parametric errors lower than the reference model configuration CNRM-CM6-1.

The reference model configuration CNRM-CM6-1 results from a tuning by the developers for the CMIP6 exercise. This tuning was done following the historical common practices for tuning a climate model (Hourdin et al. (2017), Schmidt et al. (2017)) and has been validated by model developpers. This reference model will be better defined and cited in the paper : "The reference model will be the model CNRM-CM6-1, tuned by the model developers for the CMIP6 exercise (Roehrig et al.

(2020)). This reference model has been validated by the experts and can serve as a threshold to define whether a model calibration is near-optimal."

**References :**

Hourdin, F., Mauritsen, T., Gettelman, A., Golaz, J. C., Balaji, V., Duan, Q., ... & Williamson, D. (2017). The art and science of climate model tuning. *Bulletin of the American Meteorological Society*, *98*(3), 589-602.

Schmidt, G. A., Bader, D., Donner, L. J., Elsaesser, G. S., Golaz, J. C., Hannay, C., ... & Saha, S. (2017). Practice and philosophy of climate model tuning across six US modeling centers. *Geoscientific Model Development*, *10*(9), 3207-3223.

**Specific comments**

43: This is the first time PPE is used in the main text. It should be spelled out here.

Agreed

64: Salter et al. should not be in ().

Agreed

l85: So n=102 and each \Theta_i is a vector of length 30? As mentioned above it would be helpful if the notation separated vectors from scalars.

Yes - $\theta_i$ is a vector of length 30, which is an input of the climate model $f(\theta_i)$ and the PPE $F = (f(\theta_1), ... , f(\theta_n))$ contains $n = 102$ simulations $f(\theta_i)$ that can have $l$ grid points (finite elements).

l106: \mu seems to be on the left-hand side in Eq. 105, so does r_f include \mu here?

Agreed that, as said in the text, r_f should include \mu, which is not obvious looking at the equation. We will remove \mu from equations (2) and (3) and explain in the text that f_f and r_y include \mu.

Eqs. 6 and 7:  c_f → c_y ?

Yes, this should be c_y

Eq. 14: So this is multi-linear regression. I am again confused about notation. Should \Theta_i just be the vector \Theta? Theta_i is the  vector of parameters used in the i'th climate model?

With \theta = (\theta_1,...,\theta_n), \theta_i is a particular vector of 30 parameter values, that is used as input of the climate model f(\theta_i) or the emulator : c(\theta_i) = \beta \theta_i + c0 with \beta a vector of 30 coefficients of the multi linear regression.

l174: What is a reference model? Is there any reason to assume that this is better than any random selected model from the ensemble?

The reference model configuration CNRM-CM6-1 results from a tuning by the developers for the CMIP6 exercise. This tuning was done following the historical common practices for tuning a climate model (Hourdin et al. (2017), Schmidt et al. (2017)) and has been validated by model developpers. This reference model will be better defined and cited in the paper : "The reference model will be the model CNRM-CM6-1, tuned by the model developers for the CMIP6 exercise (Roehrig et al. (2020)). This reference model has been validated by the experts and can serve as a threshold to define whether a model calibration is near-optimal."

**References :**

Hourdin, F., Mauritsen, T., Gettelman, A., Golaz, J. C., Balaji, V., Duan, Q., ... & Williamson, D. (2017). The art and science of climate model tuning. *Bulletin of the American Meteorological Society*, *98*(3), 589-602.

Schmidt, G. A., Bader, D., Donner, L. J., Elsaesser, G. S., Golaz, J. C., Hannay, C., ... & Saha, S. (2017). Practice and philosophy of climate model tuning across six US modeling centers. *Geoscientific Model Development*, *10*(9), 3207-3223.

Section 3, beginning: You consider the SAT from 3 years but what is more precisely used here? The annual means?

We use the annual mean averaged over the 3 years - this will be added in the paper

l211: Section ??

Ok

In Fig. 1 left: It is not clear to me what the hatched regions show. Is the non-parametric error only shown for q=102?

We agree that Figure 1 included too many hatched regions, making the legend a little bit confusing. We removed most of the hatched regions to simplify the figure. Here, we highlight the fact that a non-parametric component of the error exists for a perfect reconstruction of q=102 and that this non-parametric component does not evolve much when considering a lower truncation of q=50.

**Fig 2.** Figure showing the full model error $e_{tas}(\theta_i)$ and its parametric component $p_{tas}(\theta_i)$ for different truncation lengths : q=5 (red dots), q=20 (blue dots), q=50 (pink dots), q=102 (orange dots). The grey hatched region shows the non-parametric components of model errors for a perfect reconstruction of q=102.

l232: We follow section 14? Eq. 14?

Yes, it should be Eq. 14

l239: It is not clear to me what in-sample refers to here.

"In-sample" refers to the sample used for the training of the emulator and "out-of-sample" refers to a test set that was not used for the training. The out-of-sample test set is used to evaluate the prediction skill of the emulator.

l257: I think this is the first time in the paper GMMIP is mentioned. What is it?

Agreed - new sentences here : "In the context of the Global Monsoons Model Inter-comparison Project (GMMIP) (Zhou et al. (2016)), an ensemble of 10 atmospheric-only simulations of the CNRM-CM6-1 was run. In this ensemble, the reference model calibration was used, the SST was forced with the same observations as the PPE and the members differ by their initial conditions only. This dataset can be used to consider the effect of internal variability on the error decomposition, and will be referred to as the GMMIP dataset."

**Reference :** Zhou, T., Turner, A. G., Kinter, J. L., Wang, B., Qian, Y., Chen, X., Wu, B., Wang, B., Liu, B., Zou, L., and He, B.: GMMIP (v1.0) contribution to CMIP6: Global Monsoons Model Inter-comparison Project, Geosci. Model Dev., 9, 3589–3604, https://doi.org/10.5194/gmd-9-3589-2016, 2016.

I283: What does LHS mean?

"Latin Hypercube Sampling (LHS)" will be added line 82, when the term first appears

Fig. 3: The caption should also describe the plots along the diagonal.

Agreed

---

## Referee Report (RR1)

**Review of "Exploration of diverse solutions for the calibration of imperfect climate models" by Peatier et al. (2023)**

**1 Summary**

This is the second review of the manuscript by Peatier et al. 2023.

In this study, the authors propose a method to calibrate a climate model, taking into account the model uncertainty due to input parameters. For this purpose, they propose the use of emulator (statistical model) to sample the input parameters space and select sets of parameters values that lead to similar results than a reference. In order to build the emulator, they reduce the dimensionnality of the model output by using Empirical Orthogonal Function analysis. To select the candidates, a k-medians clustering method is employed. Furthermore, to better analyse the model error, a decomposition according to the parametric and non-parametric error is analysed spatially.

The manuscript has been improved from the last version. The last version really suffered of its structure. But, this new version is better structured and the main major comments have been answered. Furthermore, the methodology used is more convincing. For this reason, I accept the publication but only, if the following minors comments are taking into account. They mainly concern the structure or the explanation, not the scientific results.

**2 Minor comments**

**Title**

I find that the title of the manuscript is still very vague. It's not about diverse solutions for calibration, but rather about a single method allowing for diverse calibrations of the same model, taking into account the inherent uncertainty of the model. I suggest a title such as 'Addressing discrepancy in the calibration of a climate model'.

**Abstract**

I would add few sentences to describe the method succinctly. For example, 'A meta-model, simulating the outputs of a climate model, reduced through principal component analysis, is used to sample the degrees of freedom of the model. Thus, a subset of input parameter values yielding results similar to a reference simulation is identified.'

**Introduction**

- L. 14: Maybe add references for your sentences 'It is an important [...] quantify their uncertainties'

- L. 15: 'atmospheric convection or clouds' is quite redundant as, generally, atmospheric convection refer to convective clouds. I suggest 'radiation or turbulence or clouds'.

- L. 29-30: 'In particular, the potential [...] future climate evolution'. I think this sentence is not completed.

- L.50: I don't agree. There are different studies that have conduct sensitivity analyses to quantify uncertainty even for atmospheric model. See for example (for limited area model):

  Di, Z., Q. Duan, W. Gong, C. Wang, Y. Gan, J. Quan, J. Li, C. Miao, A. Ye, and C. Tong (2015), Assessing WRF model parameter sensitivity: A case study with 5 day summer precipitation forecasting in the Greater Beijing Area, Geophys. Res. Lett., 42, 579–587, doi:10.1002/2014GL061623.

  Wimmer, M., Raynaud, L., Descamps, L., Berre, L. and Seity, Y.(2022) Sensitivity analysis of the convective-scale AROME model to physical and dynamical parameters. Quarterly Journal of the Royal Meteorological Society, 148, 743, 920– 942, doi:10.1002/qj.4239

  And even for ARPEGE-Climat, see works done by Laurent Descamps (not published, personal documentation).

- L. 89: Add a reference to section 2: "The approach, presented in Section 2, is used as a practical..."

**Methods**

- L. 97: I would suggest to create a new paragraph to present the PPE.

- L. 99-102: This is a long sentence to describe the LHS. Please reformulate to make it clearer.

- L. 102: I suggest: 'we consider the annual means averaged over the whole 1979-1981 period as model outputs'

- L. 106: 'the simulated spatial climatology' -> 'ARPEGE-Climat' or 'GCM'

- L. 191: Maybe define $\hat{\theta} = (\hat{\theta_1}, ..., \hat{\theta_m})$

- L. 182: 'objective function' -> 'multi-variate spatial error'

- L. 183: Maybe define here $\theta_0$.

- L. 183: It is not necessary to define again the reference simulation here.

- L. 198-202: It is redundant to write twice 'The selection of candidate [...]  are shown in Section'. Maybe write only: 'The selection of candidate calibrations is detailed in Section 2.6, the results for the application to surface temperature are shown in Section 3 and for the multi-variation application, in Section 4.'

- L. 204: I suggest to keep a general view and to consider $\hat{\theta}$ instead of $\hat{\theta}_{tas}$ and $\hat{\theta}_{tot}$ for the whole section.

- L. 204: To make it clearer, I suggest: 'we aim to identify $k$ solutions among $m$ configuration, which explore...'

- L.204-226: I think the explanation in Appendix is quite good and should be put in the manuscript. Furthermore, there are redundancy in this section. This section must be reformulated.

**First application: surface temperature error**

- Figure 1: As previously, I still suggest to switch the left and right panel, as you firstly describe the right panels in the manuscript.

- L. 261: I would add: 'averaged over the PPE members, according to Figure 1b.'

- L. 263: I suggest to better clarify that $q = 5$ is an example and add 'For example, for a truncation of $q = 5$, ...'

- All along the Section 3.2 section, you refer to $\theta$, $c$, ... and not $\theta_{tas}$, $c_{tas}$, ... I suggest to use the 'tas' abbreviation in the whole section.

- L. 300: delete 'includes 10 atmospheric simulations [...] but different initial conditions'

- L. 301: I suggest this: 'to compute their associated parametric errors (yellow in Figure 2.b)'

- L. 308: 'This emulator is then optimized to find an example' -> 'This emulator is then used to optimized and thus find an example ...'

- L. 343: 'have to have' -> 'must have' ?

- L. 376: Can you justify these 4 candidates ? Why these 4 ones and not the others ?

**Second application: multi-variate error**

- Section 4.3: Why do you present candidate 6 in Figure 8 and not candidate 9 that you discuss more in the section? I would suggest to discuss and present only the best and worst candidate (1 and 5). This is sufficient. You can still discuss to the other candidates, but I think it is better to put candidates 6 and 10 in appendix.

**Conclusion**

- I would suggest to review the structure of the conclusion. I mean it is better to summarise the study firstly (paragraph L. 540-547), and then present the main results (paragraph L.530-539).

- L. 578: A missing word ? 'we did not [consider?] the observational uncertainty' ?

**3 Technical corrections**

- L.73: "perturbed parameter ensemble" -> " PPE"

- L.84: order in the references: "Peatier et al. 2022, Hourdin et al. 2023"

- L. 97: 'A PPE of this model f is created', correct the typology of '$f$'.

- L. 98: 'amip' -> 'AMIP'

- L.146: add a space between 'Rougier (2007)' and 'and'

- L. 233: '(Salter et al. 2019)' -> 'Salter et al. (2019)

- L. 234: '$y_j$' -> '$z_j$'

- L. 243: '(Salter et al. 2019)' -> 'Salter et al. (2019)

- L. 270: 'Section 14' ?

- L.309: 'for' repeated twice

- L. 350; add space in '2,4 and 5'.

- L. 505: 'an d' -> 'and'

- L. 540: 'Rougier et al. (2007)' -> '(Rougier et al., 2007)'

- L. 570: 'Williamson et al. (2013), Hourdin et al (2023)' -> '('Williamson et al. (2013), Hourdin et al (2023))'

- L. 575: '(Salter et al. 2019)' -> 'Salter et al. (2019)

- L. 577: '(Howland et al, 2022)' -> 'Howland et al. (2022)'

---

## Author Response (AR2)

**Responses to Reviewer #2**

We would like to express our gratitude once more to Reviewer #2 for their comprehensive and precise review, as well as to the editor for their dedication to this manuscript. We agreed to most of the minor comments and technical corrections and ensured that the colour schemes used in our maps and charts allowed readers with colour vision deficiencies to correctly interpret the findings.

Please, find our responses to Reviewer #2 comments below. The comments from Reviewer #2 are in black, our answers are in blue and the line numbers correspond to the track-change document.

**Summary**

This is the second review of the manuscript by Peatier et al. 2023.
In this study, the authors propose a method to calibrate a climate model, taking into account the model uncertainty due to input parameters. For this purpose, they propose the use of emulator (statistical model) to sample the input parameters space and select sets of parameters values that lead to similar results than a reference. In order to build the emulator, they reduce the dimensionnality of the model output by using Empirical Orthogonal Function analysis. To select the candidates, a k-medians clustering method is employed. Furthermore, to better analyse the model error, a decomposition according to the parametric and non-parametric error is analysed spatially.

The manuscript has been improved from the last version. The last version really suffered of its structure. But, this new version is better structured and the main major comments have been answered. Furthermore, the methodology used is more convincing. For this reason, I accept the publication but only, if the following minors comments are taking into account. They mainly concern the structure or the explanation, not the scientific results.

**Minor comments**

**Title**

I find that the title of the manuscript is still very vague. It's not about diverse solutions for calibration, but rather about a single method allowing for diverse calibrations of the same model, taking into account the inherent uncertainty of the model. I suggest a title such as 'Addressing discrepancy in the calibration of a climate model'.

The title has been changed following reviewer's suggestion.

**Abstract**

I would add few sentences to describe the method succinctly. For example, 'A meta-model, simulating the outputs of a climate model, reduced through principal component analysis, is used to sample the degrees of freedom of the model. Thus, a

subset of input parameter values yielding results similar to a reference simulation is identified.'

Agreed

**Introduction**

- L. 14: Maybe add references for your sentences 'It is an important [...] quantify their uncertainties'
  - Agreed, see L. 16
- L. 15: 'atmospheric convection or clouds' is quite redundant as, generally, atmospheric convection refers to convective clouds. I suggest 'radiation or turbulence or clouds'.
  - Agreed, see L.17
- L. 29-30: 'In particular, the potential [...] future climate evolution'. I think this sentence is not completed.
  - Agreed, "... is rarely considered" L.32
- L.50: I don't agree. There are different studies that have conduct sensitivity analyses to quantify uncertainty even for atmospheric model. See for example (for limited area model):
  Di, Z., Q. Duan, W. Gong, C. Wang, Y. Gan, J. Quan, J. Li, C. Miao, A. Ye, and C. Tong (2015), Assessing WRF model parameter sensitivity: A case study with 5 day summer precipitation forecasting in the Greater Beijing Area, Geophys. Res. Lett., 42, 579–587, doi:10.1002/2014GL061623.
  Wimmer, M., Raynaud, L., Descamps, L., Berre, L. and Seity, Y.(2022) Sensitivity analysis of the convective-scale AROME model to physical and dynamical parameters. Quarterly Journal of the Royal Meteorological Society, 148, 743, 920– 942, doi:10.1002/qj.4239
  And even for ARPEGE-Climat, see works done by Laurent Descamps (not published, personal documentation).
  - We agree tat "impossibility" was strong word and we agree that work has been done to quantify uncertainty with PPE, we removed that part of the sentence.
- L. 89: Add a reference to section 2: "The approach, presented in Section 2, is used as a practical..."
  - Agreed, L.91

**Methods**

- L. 97: I would suggest to create a new paragraph to present the PPE.
  - Agreed, L.101
- L. 99-102: This is a long sentence to describe the LHS. Please reformulate to make it clearer.
  - Agreed, L.103-106
- L. 102: I suggest: 'we consider the annual means averaged over the whole 1979-1981 period as model outputs'
  - Agreed, L.107
- L. 106: 'the simulated spatial climatology' → 'ARPEGE-Climat' or 'GCM'
  - Agreed, L.110
- L. 191: Maybe define $\hat{\theta} = (\hat{\theta}, ..., \hat{\theta})_{1m}$
  - We removed the this part of sentences, as the vectors $\hat{\theta}$ are defined L.185
- L. 182: 'objective function' → 'multi-variate spatial error'
  - Agreed, 187

- L. 183: Maybe define here $\theta_0$.
  - Agreed, L.189
- L. 183: It is not necessary to define again the reference simulation here.
  - Agreed, L.186-187
- L. 198-202: It is redundant to write twice 'The selection of candidate […] are shown in Section'. Maybe write only: 'The selection of candidate calibrations is detailed in Section 2.6, the results for the application to surface temperature are shown in Section 3 and for the multi-variation application, in Section 4.'
  - Agreed, L.205-208
- L. 204: I suggest to keep a general view and to consider $\hat{\theta}$ instead of $\hat{\theta}$ and $\hat{\theta}$ for the whole section. To make it clearer, I suggest: 'we aim to identify k solutions among m configuration, which explore…'
  - Agreed to keep a general view
- L.204-226: I think the explanation in Appendix is quite good and should be put in the manuscript. Furthermore, there are redundancy in this section. This section must be reformulated.
  - Agreed, L.210-216 and L.231-239

**First application: surface temperature error**

- Figure 1: As previously, I still suggest to switch the left and right panel, as you firstly describe the right panels in the manuscript.
  - Agreed, see Figure 1
- L. 261: I would add: 'averaged over the PPE members, according to Figure 1b.'
  - Agreed, L.277
- L. 263: I suggest to better clarify that q = 5 is an example and add 'For example, for a truncation of q = 5, …'
  - Agreed, L.279
- All along the Section 3.2 section, you refer to $\theta$, c, … and not $\theta_{tas}$, $c_{tas}$, … I suggest to use the 'tas' abbreviation in the whole section.
  - Agreed
- L. 300: delete 'includes 10 atmospheric simulations […] but different initial conditions'
  - Agreed
- L. 301: I suggest this: 'to compute their associated parametric errors (yellow in Figure 2.b)'
  - Agreed, L.318-319
- L. 308: 'This emulator is then optimized to find an example' → 'This emulator is then used to optimized and thus find an example …'
  - Agreed
- L. 343: 'have to have' → 'must have' ?
  - Agreed
- L. 376: Can you justify these 4 candidates ? Why these 4 ones and not the others ?
  - This sub-set include the best (candidate 3) and the worst (candidate 7) model, as well as some models with an interesting error pattern or emulated parametric error (candidate 10). Presenting all the 12 candidates on the same figure would make it impossible to read, but all of the candidates are visible in the appendices of the paper.

**Second application: multi-variate error**

- Section 4.3: Why do you present candidate 6 in Figure 8 and not candidate 9 that you discuss more in the section? I would suggest to discuss and present only the best and worst candidate (1 and 5). This is sufficient. You can still discuss to the other candidates, but I think it is better to put candidates 6 and 10 in appendix.
    - We think it is useful to be able to see a bit of diversity in the spatial patterns of the model errors : candidate 10 has a very different SW biases than the other candidates, for example. Being able to compare these 4 candidates on the same figure helps to convey the message of the paper.

**Conclusion**

- I would suggest to review the structure of the conclusion. I mean it is better to summarise the study firstly (paragraph L. 540-547), and then present the main results (paragraph L.530-539).
    - Agreed
- L. 578: A missing word ? 'we did not [consider?] the observational uncertainty' ?
    - Agreed

**Technical corrections**

- We have made all of the technical corrections.

• L.73: "perturbed parameter ensemble" → " PPE"
• L.84: order in the references: "Peatier et al. 2022, Hourdin et al. 2023"
• L. 97: 'A PPE of this model f is created', correct the typology of 'f'.
• L. 98: 'amip' → 'AMIP'
• L.146: add a space between 'Rougier (2007)' and 'and'
• L. 233: '(Salter et al. 2019)' → 'Salter et al. (2019)
• L. 234: '$y_j$' → '$z_j$'
• L. 243: '(Salter et al. 2019)' → 'Salter et al. (2019)
• L. 270: 'Section 14' ?
• L.309: 'for' repeated twice
• L. 350; add space in '2,4 and 5'.
• L. 505: 'an d' → 'and'
• L. 540: 'Rougier et al. (2007)' → '(Rougier et al., 2007)'
• L. 570: 'Williamson et al. (2013), Hourdin et al (2023)' → '('Williamson et al. (2013), Hourdin et al (2023))'

• L. 575: '(Salter et al. 2019)' → 'Salter et al. (2019)
• L. 577: '(Howland et al, 2022)' → 'Howland et al. (2022)'

[revised manuscript text omitted]